# The phosphorylation of carboxyl-terminal eIF2α by SPA kinases contributes to enhanced translation efficiency during photomorphogenesis

Hui-Hsien Chang [1], Lin-Chen Huang [1], Karen S. Browning [2], Enamul Huq [2] & Mei-Chun Cheng [1] ✉

Light triggers an enhancement of global translation during photomorphogenesis in Arabidopsis, but little is known about the underlying mechanisms. The phosphorylation of the α-subunit of eukaryotic initiation factor 2 (eIF2α) at a conserved serine residue in the N-terminus has been shown as an important mechanism for the regulation of protein synthesis in mammalian and yeast cells. However, whether the phosphorylation of this residue in plant eIF2α plays a role in regulation of translation remains elusive. Here, we show that the quadruple mutant of SUPPRESSOR OF PHYA-105 family members (SPA1-SPA4) display repressed translation efficiency after light illumination. Moreover, SPA1 directly phosphorylates the eIF2α C-terminus under light conditions. The C-term-phosphorylated eIF2α promotes translation efficiency and photomorphogenesis, whereas the C-term-unphosphorylated eIF2α results in a decreased translation efficiency. We also demonstrate that the phosphorylated eIF2α enhances ternary complex assembly by promoting its affinity to eIF2β and eIF2γ. This study reveals a unique mechanism by which light promotes translation via SPA1-mediated phosphorylation of the C-terminus of eIF2α in plants.

The translational landscape of any organism is influenced by environmental stimuli to promote or inhibit synthesis of proteins in response to the stimuli for survival. Many studies have shown that environmental stimuli cause greater impact on translational control than on the transcriptional level, which results in stage transition or better adaptation in plants[1,2]. During the photomorphogenic process, this translational control prepares young seedlings to transform from heterotrophic to autotrophic growth[2,3]. Although there are a few studies reporting the identities and molecular signatures associated with mRNAs regulated at the translational level[2,3], little is known about the mechanisms of light-triggered regulation of cytoplasmic translation in plants.

Translation proceeds through three distinct phases: initiation, elongation, and termination. Among these, initiation is a well-studied and crucial phase that differs significantly between eukaryotes[4]. In cytosolic mRNA translation initiation, both the 5′-m7GpppN-cap and the 3′-poly(A) tail are essential, with initiation factors recognizing these features to initiate translation[5–7]. A key regulatory mechanism in yeast and mammals involves phosphorylation of a conserved serine residue on the α-subunit of eukaryotic initiation factor 2 (eIF2α)[8]. This phosphorylation prevents eIF2B from dissociating from phosphorylated eIF2α during GDP-to-GTP recycling, thus inhibiting the formation of a new ternary complex and shutting down initiation in the absence of ternary complex

[1]Department of Biochemical Science and Technology, National Taiwan University, Taipei 10617, Taiwan. [2]Department of Molecular Biosciences, University of Texas at Austin, Austin, TX 78712, USA. ✉e-mail: ninadscheng@ntu.edu.tw

formation[6,9,10]. There are four eIF2α kinases in mammals—heme-regulated inhibitor, protein kinase double-stranded RNA-dependent (PKR), PKR-like ER kinase, and general control non-derepressible-2 (GCN2)—all phosphorylating a conserved serine residue (Ser52) of eIF2α, inhibiting translation initiation in response to different stresses[9,11]. While the translational machinery of plants shares similarities with that of *Saccharomyces cerevisiae* and mammals, plants have unique translational machinery with specific factors due to their unique biological activities such as photomorphogenesis and stress response[6,10,12]. Despite initial observations of a "PKR-like" activity in plants infected with viruses[13–16], no specific kinase has been isolated, and the sequence of a potential PKR ortholog is notably absent from plant genomes[17]. Thus, GCN2 is the only recognizable plant eIF2α kinase targeting a similar serine residue in plant eIF2α[12,18–22]. In vitro studies have identified CK2 as a kinase targeting eIF2α and eIF2β subunits, although the in vivo significance of this phosphorylation remains uncertain[23,24]. CK2 also phosphorylates the N-terminal eIF3c, the C-terminal eIF5 and eIF2β, enhancing their interaction with each other and with eIF1 in vitro[24]. Additionally, the reinitiation supporting protein can be phosphorylated by the target of rapamycin (TOR) protein, influencing the interaction between 40S and 60S ribosomal subunits to restart the translation process[25]. Notably, a study on the translational apparatus phosphoproteome under the effects of light/dark did not report phosphorylation of any eIF2 subunits[26].

In the past, SUPPRESSOR OF PHYA-105 (SPA1-4) family members have been reported to be important components in the CONSTITUTIVE PHOTOMORPHOGENIC 1 (COP1) complex, serving as degradation machinery for light signaling regulators[27,28]. Although they have kinase-like domain at the N-terminus according to sequence identity[29], they had never been reported to have kinase activity until recently. Recent evidence supports that the COP1-SPA complex is necessary for the rapid light-induced degradation of PIF1. Moreover, the COP1-SPA complex preferentially recruits phosphorylated forms of PIF1 to the E3 ligase for ubiquitylation and subsequent degradation in response to light[30–33]. In 2019, Paik et al. demonstrated that SPA1 functions as a kinase and phosphorylates PIFs[34]. Moreover, the red/far-red light photoreceptor phyB interacts with SPA1 through its C-terminus and enhances the recruitment of PIF1 for phosphorylation[34]. In addition, PIF4 plays a role in responses to light, shade and high temperature and is also a substrate for SPA1 kinase[35]. Another recently found substrate of SPA1 kinase is ELONGATED HYPOCOTYL 5 (HY5) which is also involved in light signaling[36]. However, in both of these cases, SPA1 acts as a stabilizer whereas in high temperatures, phosphorylated PIF4 and HY5 are more stable and regulate downstream genes responsible for morphological changes in seedlings[37].

Recently, COP1 was reported to negatively regulate protein translation in the dark by repressing the TOR-ribosomal protein S6 (RPS6) pathway through auxin signal transduction in Arabidopsis[38]. Upon light exposure, COP1 activity is repressed by the far-red light photoreceptor, phyA, so the TOR-RPS6 pathway can activate de novo protein synthesis in deetiolated seedlings[38]. Whether SPA1 works together with COP1 as part of the E3 ligase complex to repress translation or functions as a kinase to regulate protein synthesis remains unclear. In this study, we discovered that SPA1 directly phosphorylates multiple Ser/Thr residues of eIF2α at its C-terminus, which is distinct from the more studied site of Ser52 phosphorylation in other eukaryotes. The C-term-phosphorylated eIF2α promotes protein translation and photomorphogenesis, whereas the C-term-unphosphorylated eIF2α results in a decreased translation efficiency. We also demonstrate that the phosphorylated eIF2α may enhance initiation complex assembly by interacting strongly with other translation initiation factors. Our study reveals a regulatory mechanism specific to plants in which SPA1 promotes translation in response to light by phosphorylation of the C-terminus of eIF2α.

## Results

### *spaQ* mutants show enhanced translation efficiencies in etiolated seedlings but decreased translation upon light treatment

To understand whether SPAs participate in the light-responsive translational regulation, we performed polysome profile analyses to examine the translational status of 4-day-old Col-0, *cop1-4* and *spaQ* (*spa* quadruple mutants) etiolated seedlings treated with or without 4 h white light illumination. Fractions with no greater than 2 ribosomes were designated as the non-polysomal (NP) fraction, whereas the polysomal (PL) fraction includes mRNAs associated with at least three ribosomes representing a more active translational status (Fig. 1a). Compared with Col-0, the etiolated *cop1-4* and *spaQ* mutants (Dark) showed an increase in the PL fraction (Fig. 1a). Four-hour white light illumination did not further increase the translation in *cop1-4* mutant. However, after light exposure, *spaQ* showed a decrease in the PL fraction compared with *cop1-4* and Col-0 (Fig. 1a). To obtain more quantitative analyses of the ribosome loading efficiency, we isolated RNAs in NP and PL fractions to quantify the translation efficiency. Consistent with our ribosome profiling, the proportion of RNAs in the PL fraction in the dark-grown *cop1-4* and *spaQ* mutants were comparable to that in Col-0 treated with 4 h light (L4h). This suggests that both COP1 and SPAs repress global translation in the dark. On the other hand, *spaQ* mutants had significantly lower PL RNA content after illumination compared to Col-0 (Fig. 1b). Besides the translational level, we performed a de novo protein synthesis analysis with three repeats which showed a similar pattern to the polysome profile (Supplementary Fig. 1). Collectively, these results suggest that SPAs might enhance global translation in response to the dark-to-light transition.

### SPAs-mediated translational changes target mRNAs functioning in light responses, growth development, and cell wall biogenesis

To understand the global translation regulated by SPAs, we performed transcriptomic analyses of steady-state mRNAs (mRNA$_{SS}$) and polysome-bound mRNAs (mRNA$_{PL}$) by RNA-sequencing using 4-d-old Col-0 and *spaQ* dark-grown seedlings treated with or without 4 h white light illumination. Three biological replicates were performed for 0 h (Dark) and 4 h (Light) treated-samples to measure changes of mRNA at the transcriptional level (mRNA$_{SS}$) and translational level (mRNA$_{PL}$) in both Col-0 and *spaQ* (Fig. 2a). Compared with Col-0, in dark-grown *spaQ* plants, 7383 genes were identified as SPAs-regulated differentially expressed genes (DEG) with statistically significant changes at the mRNA$_{SS}$ or mRNA$_{PL}$ level (Fig. 2b). In order to observe the changes in mRNA$_{PL}$ compared to mRNA$_{SS}$, we have calculated the translation efficiency (TE) of these DEG by normalizing the fold change in mRNA$_{PL}$ with the fold change in mRNA$_{SS}$ (TE = mRNA$_{PL}$ fold change/mRNA$_{SS}$ fold change) and performed clustering analysis as shown in Fig. 2b. Most DEG showed increased TE in the dark condition but decreased TE under light condition, suggesting that SPAs inhibit the translation but enhance the translation of these genes. To further classify the type and extent of these genes regulated by SPAs, we performed k-means clustering analysis. The list of SPA-regulated genes which are regulated under both dark and light condition in different clusters are listed in Supplementary Data 1–4. GO analysis of SPA-regulated genes on both dark and light condition in different clusters are listed in Supplementary Data 5–8. mRNAs in Cluster 1 (1620 genes) showed a concordant increase in mRNA$_{SS}$ and mRNA$_{PL}$ levels in *spaQ* under dark condition (Fig. 2c). The mRNAs in cluster 1 are likely targets of transcription factors degraded by SPAs and COP1 during photomorphogenesis. mRNAs in Cluster 2 (830 genes) showed an increase in mRNA$_{SS}$ or mRNA$_{PL}$ level in *spaQ* mainly under light condition, so SPAs might repress the steady-state level or translation efficiency of these mRNAs. mRNAs in Cluster 3 (927 genes) showed a general reduction in both mRNA$_{SS}$ and mRNA$_{PL}$ levels in *spaQ* under both dark and light conditions (Fig. 2c), suggesting that SPAs are required to maintain both transcription and translation level of these mRNAs. Cluster 4,

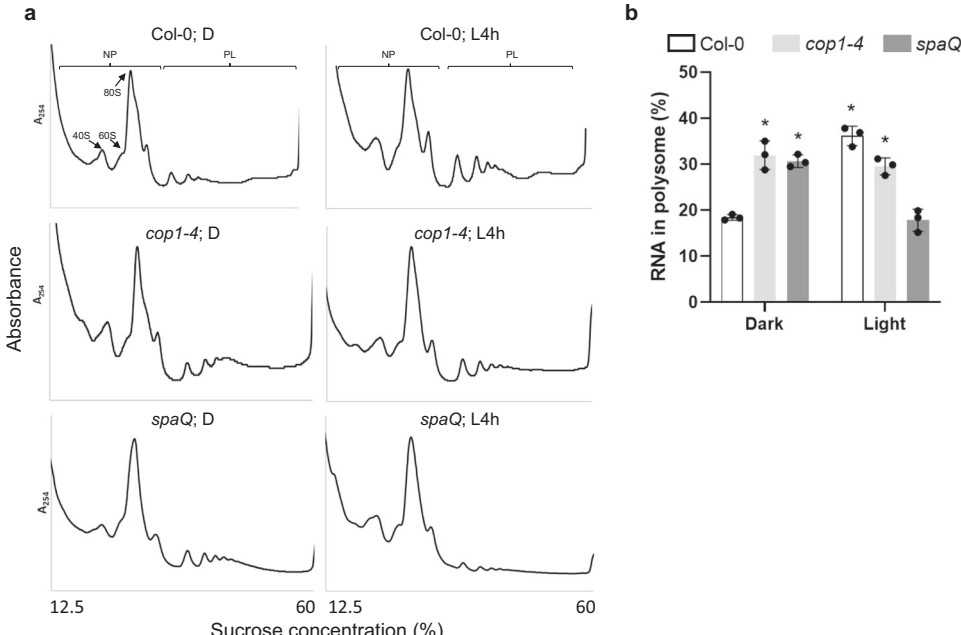

**Fig. 1 | *spaQ* and *cop1-4* mutants exhibited enhanced translation in the dark but repressed translation upon light exposure. a** The plots of polysome profiling showed different pattern of ribosomal RNA between Col-0, *cop1-4* and *spaQ*. Extracted ribosomal RNA were fractionated by 12.5 to 60% sucrose gradient. The positions of non-polysomal (NP) and polysomal (PL) RNAs are indicated in the profiles. **b** The bar charts show the percentage of polysomal RNA in total RNA in Col-0, *cop1-4* and *spaQ*. Four-day-old etiolated seedlings were incubated either under dark (D) or illuminated 4 h white light (L4h) before sampling. After gradient separation of NP and PL, the RNAs were extracted by PCI method. Error bars indicate the mean ± SD ($n = 3$ biological repeats). Spike-in RNA (Daptomycin) was used for data normalization. Asterisks indicate translation efficiency of conditions statistically different from that of 4-day-old etiolated Col-0 seedlings. \**p*-value < 0.01, two-sided Student's *t* test.

representing the largest group of mRNAs (4006 genes), showed a more decrease in translation but smaller changes in $\text{mRNA}_{SS}$ level in *spaQ* under the light condition compared to the levels under dark condition. More than 50% of SPAs-regulated genes are in Cluster 4. According to our gene ontology analyses, mRNAs function in photosynthesis, chloroplast organization, and chlorophyll metabolism are largely regulated in cluster 1, whereas mRNAs involved in cell wall metabolism, growth development, auxin response, and photosynthesis are regulated in cluster 4 (Fig. 2c).

Interestingly, our translatomic analysis showed that mRNAs involved in auxin response and auxin-activated signaling pathway are overrepresented in cluster 4, suggesting that SPAs might repress the expression of these mRNAs under dark condition but promote their expression upon light exposure. To test whether SPAs might also regulate TOR-RPS6 pathway through auxin signal transduction, we selected the genes that are involved in TOR-RPS6 pathway and examined their expression patterns at both transcriptional and translational levels. As shown in Supplementary Fig. 2a, genes related to auxin-activated signal transduction show enhanced expression under dark condition but repressed expression upon light exposure. Moreover, 21 selected genes involved in TOR-RPS6 pathway, including *TOR*, *S6K*, *FCS-LIKE ZINC FINGER* (*FLZ*), *LETHAL WITH SEC13 PROTEIN 8* (*LST8*) and *RHO OF PLANTS 2* (*ROP2*), also show similar pattern (Supplementary Fig. 2b). These results indicate that SPAs might play contrasting role in regulating translation through repressing auxin-TOR signal transduction under dark condition but promoting auxin-TOR signal transduction upon light exposure.

## SPA1 interacts with both eIF2α homologs in a light-dependent manner through its kinase domain

Previous reports showed that SPAs not only play accessory roles in enhancing COP1 E3 ligase function, they are also important kinases that phosphorylate light signaling regulators, such as PIF1, PIF4 and HY5[34–36]. Since SPAs might participate in translational regulation during the dark-light transition, we postulated that SPAs might target and phosphorylate eIF2α to regulate its activity. To test this hypothesis, we first determined whether SPA1 was able to interact with eIF2α. Truncated forms of SPA1 domain constructions[34] were used to map the interacting domain by the yeast two-hybrid (Y2H) assay. SPA1 was separated into five fragments depending on its domain structure, including kinase domain, coiled-coil (CC) domain, and WD40 domain. These fragments were fused with the GAL4 activation domain (AD) and eIF2α was fused with the DNA binding (BD) domain. Both homologs of eIF2α, eIF2α.1 (AT5G05470) and eIF2α.2 (AT2G40290), interact with the full length and kinase domain of SPA1 (Fig. 3a). eIF2α.1 also interacts with CC and WD40 domains, whereas eIF2α.2 also interacts with CC domain. We also performed pull-down assay to validate the interaction between eIF2α homologs and SPA1. As shown in Supplementary Fig. 3, both homologs of eIF2α can interact with SPA1. To investigate their interaction activities under dark and light conditions, we performed co-immunoprecipitation (co-IP) assay to further test their interactions. The total proteins of eIF2α.1- and eIF2α.2-GFP transgenic seedlings were extracted and mixed together with total extract from tandem affinity purification cMyc-SPA1 (TAP-SPA1) transgenic seedlings. The mixture was incubated with GFP-Trap (Chromotek) to perform co-IP. The results show that the interaction between eIF2α homologs and SPA1 was exclusively detected in light-treated etiolated seedlings (Fig. 3b), indicating that light promotes the interaction between eIF2α homologs and SPA1. The previous reports have showed that SPA1 is localized in nucleus both under dark condition and upon red light exposure[29,39], but the function and dynamics of cytosolic SPA1 remain unclear. To confirm the light-dependent interaction between eIF2α, which is localized in the cytosol, and SPA1, we have fractionated the cytosolic and nuclear proteins using TAP-SPA1 etiolated seedlings and 4 h-light-treated etiolated seedlings to detect the localization of SPA1 under dark and light conditions. As shown in Supplementary

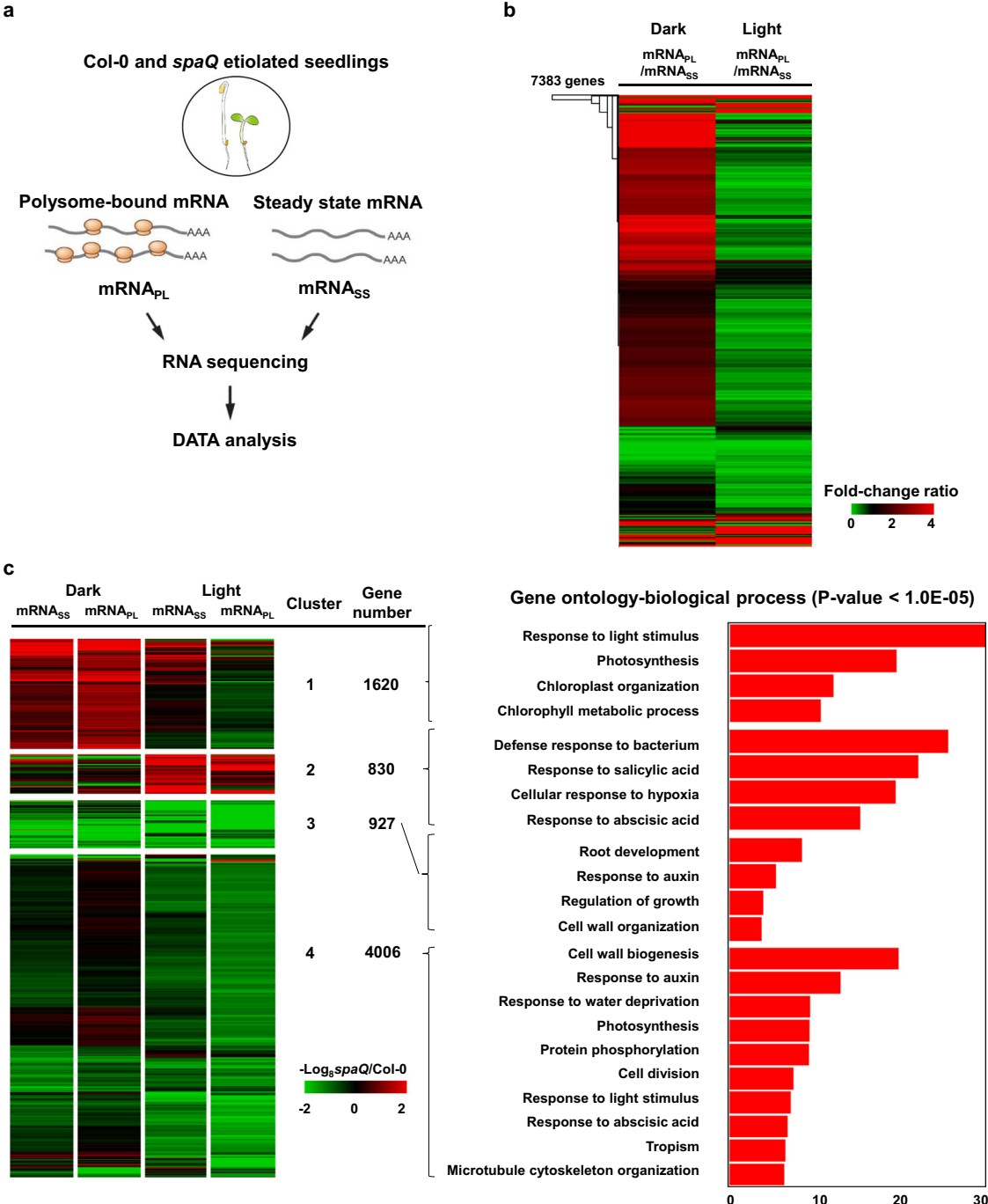

**Fig. 2 | SPAs enhance massive translation after light exposure. a** An illustration of the experimental design. Steady-state mRNAs (mRNA$_{SS}$) and polysome-bound (mRNA$_{PL}$) were isolated in parallel and used for library construction and sequencing for transcriptomic and translatomic analyses, respectively. **b** Translation efficiency (TE) analysis of 7383 SPA-regulated genes. TE was calculated by normalizing the fold change in mRNA$_{PL}$ with the fold change in mRNA$_{SS}$. Extreme red and green colors indicate ratio 4 and ratio 0 of the fold change, respectively. **c** Classification of SPA-regulated genes at mRNA$_{SS}$ or mRNA$_{PL}$ level in *spaQ* normalized to the WT. K-means clustering was used to classify the 7383 genes regulated by SPA. Extreme red and green colors indicate 64-fold up-regulation and down-regulation, respectively. Enrichment analysis (Gene Ontology) based on the Fisher Exact test was implemented using the Database for Annotation, Visualization and Integrated Discovery (DAVID) v6.8.

Fig. 4, SPA1 was more abundant in the nucleus under dark. However, after 4 h light treatment, SPA1 was mostly in the cytosolic fraction. To further confirm this result, we generated the construct of green fluorescence protein (GFP) fused to the C-terminus of SPA1 driven by *Cauliflower mosaic* virus 35 S promoter and transformed it into Col-0 protoplasts. As shown in Supplementary Fig. 5, SPA1-GFP localized in the nucleus in 91.3% of the transformed protoplasts incubated under dark for 2 h before observation. However, under light condition, 37.5%

of the transformed protoplasts have SPA1-GFP forming speckles in the cytosol. To further confirm the interaction between SPA1 and eIF2α in the cytosol, we isolated cytosolic protein of TAP-SPA1 transgenic seedlings and incubated with anti-cMyc (MERCK, Cat#op10) to perform co-IP (Fig. 3c). The results showed that the cytosolic TAP-SPA1 has higher interaction intensity with native eIF2α after light treatment. We also performed bimolecular fluorescence complementation (BiFC) to further validate the interaction between eIF2α homologs and SPA1 in

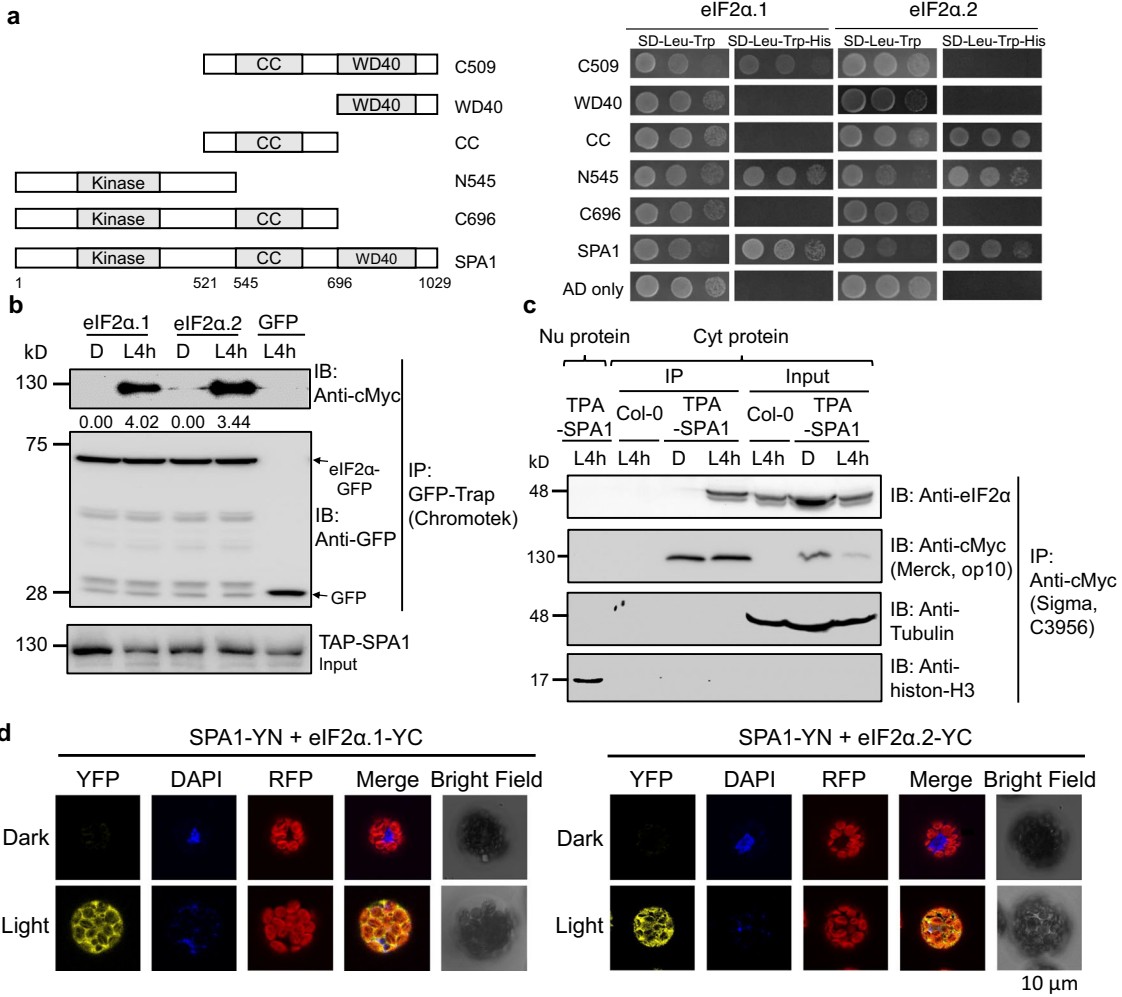

**Fig. 3 | SPA1 interacts with eIF2α in the cytosol in a light-dependent manner through its kinase domain. a** Mapping of the interaction domains for SPA1 using the yeast-two-hybrid assays. Left panel shows the full-length and various domains of SPA1. Right, interactions between SPA1 and eIF2α proteins in GAL4 yeast two-hybrid assays. The full-length of eIF2α homologs and different forms of SPA1 were fused with the GAL4 binding domain (BD) and activation domain (AD), respectively. The AD domain without SPA1 conjugation (AD only) is used as negative control. **b** Co-immunoprecipitation (Co-IP) assay shows that two homologs of eIF2α interact with SPA1 in response to light. Four-day-old etiolated seedlings were incubated either under dark (D) or illuminated under white light for 4 h (L4h) before sampling. The total proteins were extracted and incubated with GFP-Trap. The precipitated proteins were examined by immunoblotting using antibodies against cMyc and GFP. The numbers below the blot of TAP-SPA1 indicate the relative band with three biological repeats intensities of co-precipitated TAP-SPA1 normalized to precipitated eIF2α-GFP. The GFP only was used as negative control. **c** Co-IP assay shows

that the cytosolic TAP-SPA1 interacts with native form of eIF2α after light treatment. Four-day-old etiolated seedlings were incubated either under dark (D) or illuminated under white light for 4 h (L4h) before sampling. The total proteins were extracted and separated into nuclear (Nu) or cytosolic (Cyt) proteins. The cytosolic proteins were incubated with anti-cMyc. The precipitated proteins were examined by immunoblotting using antibodies against cMyc and eIF2α. The nuclear proteins in L4h TAP-SPA1 and cytosolic proteins in L4h Col-0 were used as negative control. Anti-tubulin and anti-histone-H3 antibodies were used to detect cytosolic and nuclear marker protein, respectively. Three biological repeats of data showed the same results. **d** BiFC assay shows the interaction of SPA1 with eIF2α homologs under light condition with three biological repeats. The dark-treated samples were incubated under dark for 2 h before observation. The YFP, Chl, 4′, 6′-diaminophenylindole (DAPI; nucleus staining), red fluorescence protein (RFP; chlorophyll fluorescence), merge (merged image) and bright field were shown for each type of transformation combination. Bar = 10 μm.

plant. We generated the constructs of N- and C-terminal yellow fluoresce protein fused to the C-terminus of SPA1 and eIF2α homologs, respectively. The generated constructs were transformed into Col-0 protoplasts. The BiFC results showed that both homologs of eIF2α can interact with SPA1 in the cytosol under light condition (Fig. 3d; Supplementary Fig. 6). Collectively, these results suggest that the interactions between SPA1 and eIF2α homologs are light-dependent and localized in the cytosol.

## SPA1 directly phosphorylates the C-terminus of eIF2α but not the Ser56

The serine regulatory phosphorylation site in the N-terminus of eIF2α is highly conserved among eukaryotes. An alignment of the amino acid

residues around the target serine in budding yeast (*Saccharomyces cerevisiae*), human (*Homo sapiens*), Arabidopsis, rice (*Oryza sativa*), and wheat (*Triticum aestivum*) eIF2α is shown in Supplementary Fig. 7[40]. After confirming the interaction between SPA1 and eIF2α, we tested whether SPA1 phosphorylates eIF2α at Ser56. To test this, we purified strep-tagged full-length SPA1 protein from eukaryotic host (*P. pastoris*) and eIF2α homologs with GST tagged to their N-terminus and strep tagged to their C-terminus (GST- eIF2α.1/GST- eIF2α.2) from bacteria (*E. coli*) to perform in vitro kinase assay. To examine whether SPA1 phosphorylates eIF2α at Ser56, we also purified a mutated version of eIF2α (S56A) where the Ser56 was replaced with alanine to generate a form of eIF2α that cannot be phosphorylated. With increasing eIF2α protein, we observed stronger phosphorylation signals of both

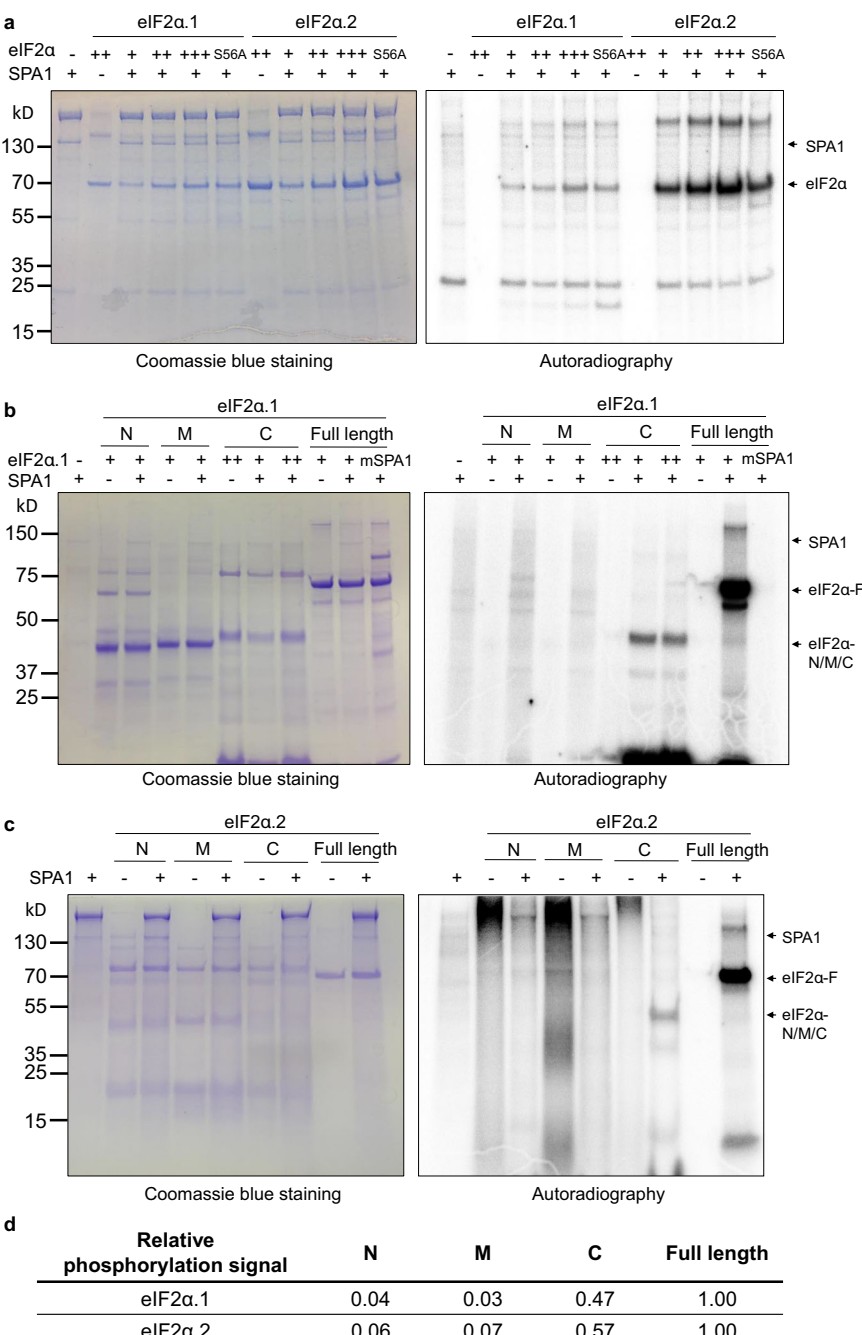

| Relative phosphorylation signal | N | M | C | Full length |
|---|---|---|---|---|
| eIF2α.1 | 0.04 | 0.03 | 0.47 | 1.00 |
| eIF2α.2 | 0.06 | 0.07 | 0.57 | 1.00 |

**Fig. 4 | SPA1 directly phosphorylates eIF2α in the C-terminal domain in vitro but not at Ser56. a** An in vitro kinase assay showed that SPA1 protein purified from *P. pastoris* phosphorylates eIF2α (autoradiogram on the right). The left figure showed the protein levels in a Coomassie blue staining gel. The conserved amino acid mutation Ser56 to alanine (S56A) did not reduce the phosphorylation signal of eIF2α. Three biological repeats of data showed the same results. **b**, **c** The in vitro kinase assay showed that SPA1 protein purified from *P. pastoris* phosphorylates C-terminal eIF2α.1 and eIF2α.2 (autoradiogram on the right), respectively. The left

figure showed the protein levels in a Coomassie blue staining gel. N-terminal (N), middle (M), and C-terminal (C) domains of eIF2α were shown. A conserved amino acid mutation on the SPA1 kinase domain (mSPA1) reduced the phosphorylation activity of SPA1 on eIF2α. Three biological repeats of data showed the same results. **d** A table showing the relative phosphorylation signal intensities of autoradiogram normalized to protein contents detected by immunoblotting as shown in Supplementary Fig. 8. The ratio of the phosphorylation of full-length eIF2α was set to 1 for analysis.

wild-type eIF2α homologs (Fig. 4a). The phosphorylation signal in eIF2α.2 is stronger than in eIF2α.1. Strikingly, the phosphorylation signals of the S56A mutants in both eIF2α homologs were not eliminated, suggesting Ser56 is not be a major target of SPA1 in Arabidopsis eIF2α.

To analyze the phosphorylation sites, we again performed in vitro kinase assay using three different fragments of eIF2α homologs, including N-terminal (N), middle (M) and C-terminal (C) fragments. The results show that only the C-terminal portion is phosphorylated by

SPA1 for both eIF2α homologs (Fig. 4b–d). Due to the low amount of C-terminal eIF2α.2, a western blot detecting the C-terminal eIF2α.2 was provided (Supplementary Fig. 8). To further provide evidence that SPA1 can interact with the C-terminal eIF2α, we also performed semi-in vivo and in vivo co-IP analysis using eIF2α fragments. Results showed that SPA1 interacts with both N- and C-terminal fragments of eIF2α homologs (Supplementary Fig. 9). However, only C-terminal of eIF2α can be phosphorylated by SPA1. We also used a mutated form of SPA1

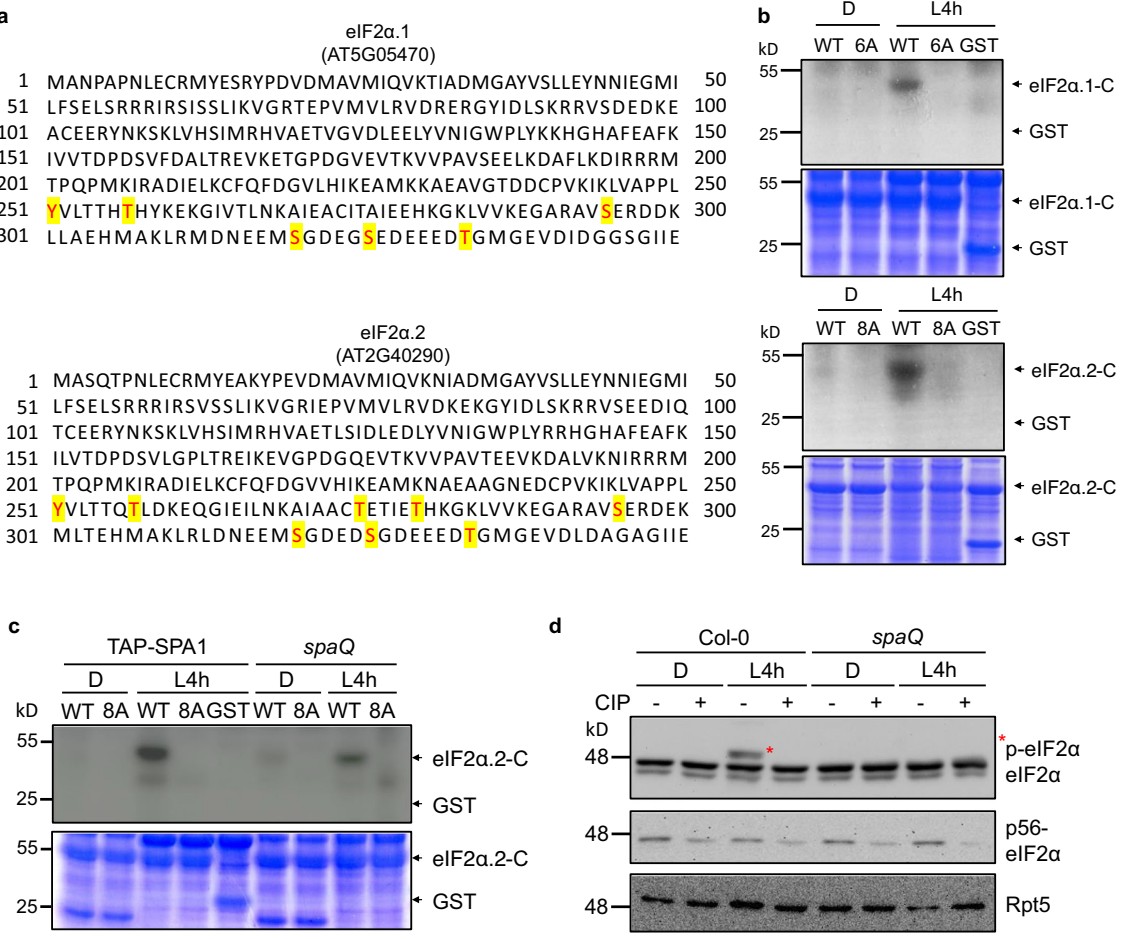

**Fig. 5 | SPA1 phosphorylates C-terminus of eIF2α.1 and eIF2α.2 in a light-dependent manner. a** Predicted phosphorylation sites located at C-terminal eIF2α are indicated in red. Tyr, Ser or Thr were then replaced with alanine (A) and aspartic acid (D) to generate a phospho-null form and a phospho-mimicking form of eIF2α, respectively. **b** A semi-in vivo kinase assay showed that SPA1 protein extracted from TAP-SPA1 overexpression line phosphorylates wild-type (WT) eIF2α but not the phospho-null form (6A/8A) of eIF2α (autoradiogram in upper panel). The plant extracts were derived from 4-day-old TAP-SPA1 seedlings incubated either under dark (D) or illuminated under white light for 4 h (L4h) before sampling. Three biological repeats of data showed the same results. **c** A semi-in vivo kinase assay showed that WT version of eIF2α has higher phosphorylation intensity in TAP-SPA1 than in *spaQ* under light condition (autoradiogram in upper panel). The plant extracts were derived from 4-day-old TAP-SPA1 and *spaQ* incubated either under

dark (D) or illuminated with 4 h white light (L4h) before sampling. The GST only was used as negative control. The lower panel shows the protein levels in a Coomassie blue staining gel. Three biological repeats of data showed the same results. **d** Immunoblots showed phosphorylation of eIF2α under both dark and light condition in Col-0 and *spaQ*. Four-day-old Col-0 and *spaQ* etiolated seedlings incubated either under dark (D) or illuminated with 4 h white light (L4h) before sampling, and extracted proteins were then separated on 6.5% sodium dodecyl sulfatepolyacrylamide gel electrophoresis (SDS-PAGE) gels containing 15 μM Phostag. The slow-migrating band (*) is the phosphorylated form of eIF2α. Three biological repeats of data showed the same results. Treatment of calf intestinal alkaline phosphatase (CIP, +) and inactivated boiled CIP (-) were shown. Rpt5 proteins were used as loading control.

(mSPA1) that has the R517E mutation in Ser/Thr kinase domain in the kinase assay and no phosphorylation activity was observed[34] (Fig. 4b; Supplementary Fig. 10). These data suggest that SPA1 acts as a protein kinase and directly phosphorylates the C-terminal domain of eIF2α rather than the conserved phosphorylated site in the N-terminal domain.

## SPAs are necessary for the light-induced phosphorylation of eIF2α in vivo

To further map the phosphorylation sites in the C-terminus of eIF2α homologs, we used GST-eIF2α to conduct in vitro phosphorylation assay using SPA1 as a kinase and performed mass-spectrometric analyses. However, no phosphopeptides were detected during the mass-spectrometric analyses of eIF2α tryptic fragments. This is likely because the C-terminal peptide lacks any basic amino acids, has a high mass, and contains multiple methionine residues that may be partially oxidized. These factors all likely contribute to the difficulty in detecting phosphorylated residues by mass spectrometry. Despite the

absence of any unambiguous identification, computational prediction analysis (NetPhosK 3.0) suggested six amino acids might serve as potential phosphorylation sites within the C-terminus of Arabidopsis eIF2α.1, including Y251, T257, S295, S317, S322 and T329. In addition, eight amino acids within the C-terminus of Arabidopsis eIF2α.2, including Y251, T257, T275, T280, S295, S317, S322 and T329, were predicted to be potential phosphorylation sites (Fig. 5a). To understand how conserved these sites are among different eukaryotes, we performed an alignment analyses of both N- and C-terminal sequences of the two Arabidopsis eIF2α homologs together with the sequences of eIF2α proteins from other eukaryotic species (Supplementary Figs. 11, 12). As shown in Supplementary Fig. 12, Y251 and T257 in Arabidopsis eIF2α are highly conserved among different eukaryotic species. S295, S317, S322, and T329 in Arabidopsis eIF2α are only present in plants and are absent in animals and insects.

To address the significance of the phosphorylation sites and their regulatory function under different light conditions, we replaced these sites with alanine (6A/8A) or aspartic acid (6D/8D) to generate

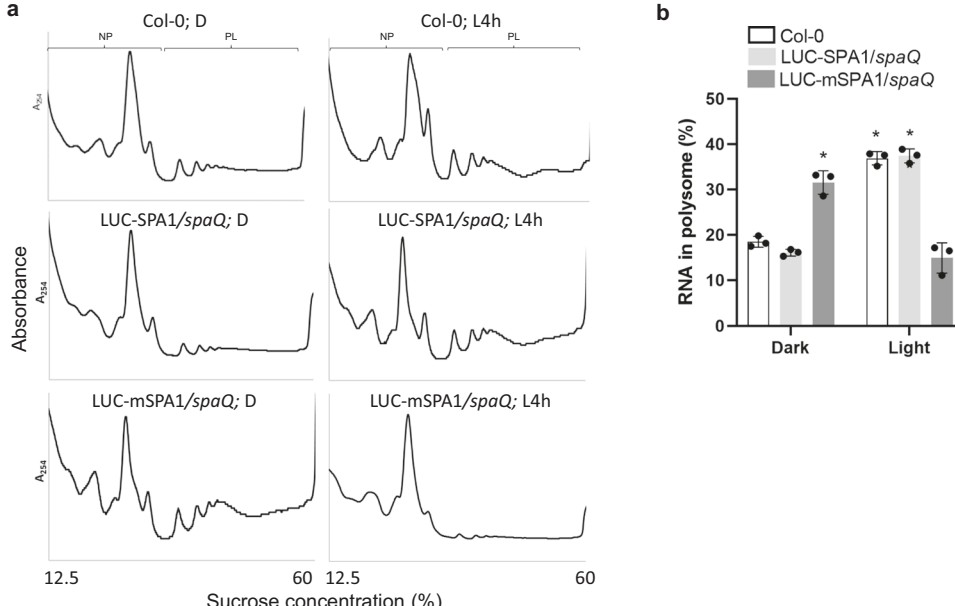

**Fig. 6 | Mutation in SPA1 kinase domain could not rescue the repressed translation efficiency in *spaQ* after light treatment. a** The plots of polysome profiling showed different pattern of ribosomal RNA between Col-0, LUC-SPA1/*spaQ* and LUC-mSPA1/*spaQ*. Extracted ribosomal RNA were fractionated by 12.5–60% sucrose gradient. The positions of non-polysomal (NP) and polysomal (PL) RNA are indicated on the profiles. **b** The bar charts showed the percentage of polysomal RNA in total RNA in Col-0, LUC-SPA1/*spaQ* and LUC-mSPA1/*spaQ*. Four-day-old etiolated seedlings were incubated either under dark (D) or illuminated 4 h white light (L4h) before sampling. After gradient separation of NP and PL, the RNA was extracted by PCI method. Error bars indicate the mean ± SD ($n = 3$ biological repeats). The expression of spike-in RNA (Daptomycin) was used for data normalization. Asterisks indicate translation efficiency of conditions statistically different from that of 4-day-old etiolated Col-0 seedlings. *$p$-value < 0.01, two-sided Student's $t$ test.

phospho-null mutant and phospho-mimicking mutative form, respectively. To eliminate any potential interference signals from Ser56 phosphorylation by GCN2, we generated the variant eIF2α homologs containing only the C-terminal domain. We then performed semi-in vivo kinase assay using the wild-type form (WT) or phospho-null form (6A/8A) of GST-tagged C-terminal eIF2α.2, which purified from *E. coli* as the substrates, with plant crude extracts prepared from TAP-SPA1 etiolated seedlings incubated either under dark (D) or illuminated under white light for 4 h (L4h) as the kinase source. The results show that mutant recombinant protein, eIF2α.1-6A and eIF2α.2-8A, cannot be phosphorylated in the presence of TAP-SPA1 (Fig. 5b). Moreover, eIF2α homologs can be phosphorylated only in the light-treated sample, supporting that at least some of these predicted residues might be the phosphorylation sites of eIF2α homologs in light conditions. Of the predicted sites that we performed mutagenesis, S317 and S322 were predicted to be the possible CK2 phosphorylation sites in Arabidopsis[24]. Further studies are needed to clarify whether mutation of these two sites alone can result in similar effect as in 8A and 8D mutations.

To further confirm that SPAs contribute to the C-terminal eIF2α phosphorylation, we performed semi-in vivo kinase assay using the C-terminal eIF2α.2-WT and 8A purified from *E. coli* as the substrates, with plant crude extracts prepared from TAP-SPA1 and *spaQ* etiolated seedlings incubated either under dark (D) or illuminated under white light for 4 h (L4h) as the kinase source (Fig. 5c). The results showed that eIF2α.2-WT has higher phosphorylation signal in TAP-SPA1 than in *spaQ* background after light treatment. The weak signal observed in *spaQ* is probably due to the presence of other kinases, such as CK2, that might still phosphorylate the C-terminal eIF2α.2[24]. The in vivo phosphorylation status of eIF2α was further examined by utilizing 15 μM Phos-tag containing SDS-PAGE gels. In wild type plants treated with white light, only the inactivated boiled calf intestinal alkaline phosphatase (CIP)-treated eIF2α showed mobility shift. However, the *spaQ* mutant did not exhibit mobility shift under light conditions

(Fig. 5d). The reduction of the light-induced phosphorylation of eIF2α in the *spaQ* mutant suggests that SPAs are likely responsible for the light-induced C-terminal eIF2α phosphorylation in vivo.

## Mutation in SPA1 kinase domain could not rescue the repressed translation efficiency in *spaQ* under light treatment

To examine the biological significance of the SPA1-mediated phosphorylation of eIF2α, we used homozygous transgenic plants expressing either the wild-type Luciferase (LUC)-SPA1 or the LUC-mSPA1 in the *spaQ* background to examine their translation efficiencies. Polysome profile results showed that the wild-type LUC-SPA1 rescued the altered translation in *spaQ* to levels comparable to Col-0 (Fig. 6a). However, the kinase mutant LUC-mSPA1 failed to complement the light-mediated translational changes in *spaQ*, suggesting that the kinase activity is crucial for the translational regulation. To further confirm this result, we also isolated NP and PL RNA to quantify the translation efficiency. Consistent with our ribosome profiling data, similar to *spaQ* mutant, LUC-mSPA1/*spaQ* seedlings have higher PL RNA in dark-grown seedlings and lower PL RNA content after illumination compared to Col-0 and LUC-SPA1/*spaQ* (Fig. 6b). Both ribosome profiling and RNA content results support a role of the SPA1 kinase activity in regulating the light-induced translational changes through eIF2α.

## eIF2α.2 might play a more important role in embryogenesis compared with eIF2α.1

To identify the difference between eIF2α homologs, we ordered two different T-DNA insertion lines for each of the eIF2α homologs from Arabidopsis Biological Research Center. Mutants were screened as described in Materials and Methods section. The T-DNA inserted sites were identified as shown in Supplementary Fig. 13a. One of the T-DNA inserted site of *eif2α.1* mutant is at 5' UTR and another one is at the fourth intron. Both T-DNA inserted sites of *eif2α.2* mutants are at exons. Next, we tried to generate their homozygous mutants for

phenotypical analyses. However, we could not obtain homozygous lines for both *eif2α.2* mutants (SAIL_864_B04 and SAIL_1156_D08), and the genotypic analyses showed that the ratio of heterozygous: wild type is about 2:1 (Supplementary Fig. 13b). In the siliques of *eif2α.2* heterozygous mutants, we also observed about 25% white seeds among green seeds, suggesting that *eif2α.2* homozygous knockout mutants might be embryogenic lethal. We used quantitative RT-PCR and western blot to check the gene and protein expressions in the *eif2α.1* mutants. The results suggested that the line SALK_144009 mutant might be knockout mutant (Supplementary Fig. 13c); however, our western blot results of protein expression showed that *eif2α.1* mutants had similar eIF2α levels in both T-DNA insertion lines compared with Col-0 (Supplementary Fig. 13d). This might be because eIF2α.2 is able to compensate for the loss of eIF2α.1. Both results suggest that eIF2α.2 might be involved in embryogenic development and seems to play a more important role in that process compared with eIF2α.1.

### Phosphorylation of the C-terminal domain of eIF2α.2 promotes translation efficiency and photomorphogenesis

To investigate the function of C-term-phosphorylated eIF2α.2, we generated transgenic plants overexpressing wild-type form (WT), phospho-null form (8A) and phospho-mimicking form (8D) eIF2α.2 in Col-0 background (Supplementary Fig. 14a). Interestingly, the native form of eIF2α in these transgenic lines are slightly less abundant than in Col-0, showing that the transgenic eIF2α.2 might account for a partial function in the transgenic lines (Supplementary Fig. 14a). To examine the functionality of GFP conjugated eIF2α.2, we generated complementary lines of eIF2α.2-WT, 8A, 8D in *eif2α.2* mutants. These complementary lines produced viable seeds (Supplementary Fig 14b), in contrast to the embryogenic lethal seeds observed in *eif2α.2* heterozygous knockout mutants (Supplementary Fig. 13b), suggesting that the GFP-fused eIF2α.2 proteins are functional. To examine the translation efficiencies, four-day-old etiolated seedlings of eIF2α.2-WT, 8A and 8D transgenic lines treated with or without 4 h white light illumination were used to perform polysome profile analyses (Fig. 7a). As shown in Fig. 7a, eIF2α.2-WT overexpressing line displays similar pattern compared with Col-0, which is shown in Fig. 1a. The PL fraction significantly increased after illumination. However, seedlings overexpressing eIF2α.2-8A has much lower PL peaks even after light treatment compared with eIF2α.2-WT. Conversely, seedlings overexpressing eIF2α.2-8D have much higher PL peaks compared with eIF2α.2-WT and even higher after illumination. Consistent with our ribosome profiling data, the quantification results by isolating NP and PL RNA also show that the phospho-null eIF2α.2-8A has a significantly lower translation efficiency, whereas the phospho-mimicking eIF2α.2-8D displays much higher translation efficiency compared with wild-type form (Fig. 7b). To further verify the protein production in these transgenic lines, we measured their fresh and dry weight, as well as relative water content. The results showed that both eIF2α.2-WT and 8D had higher fresh weight and dry weight than eIF2α.2-8A (Supplementary Fig. 15a, b), whereas the relative water content in these lines did not exhibit significant difference (Supplementary Fig. 15c). We also performed de novo protein synthesis analysis and obtained similar pattern compared to the result of polysome profiles (Supplementary Fig. 15d). To further verify that the translation changes in *spaQ* mutant are due to defective phosphorylation of C-terminal eIF2α in vivo, we have transiently expressed eIF2α.2-WT, 8A, and 8D using AGROBEST transfection technique and performed polysome profile analyses to see whether ectopic expression of phospho-mimicking eIF2α.2 could rescue the repressed translation in *spaQ* mutants under light condition. As shown in Fig. 7c, both eIF2α.2-WT and 8A overexpressing lines display similar pattern with *spaQ* (Fig. 1a), where the translation efficiency cannot be highly induced upon light exposure. However, seedlings overexpressing eIF2α.2-8D showed high translation

efficiencies in both dark and light treatments. Consistent with our ribosome profiling data, the quantification results by isolating NP and PL RNA also show the same patterns with polysome profiles (Fig. 7d). The equal expression of different eIF2α.2 variants transiently expressed in *spaQ* mutants were verified using immunoblotting as shown in Supplementary Fig. 16. These data suggest that the SPAs-mediated phosphorylation of the C-terminal domain of eIF2α.2 promotes translation efficiency under dark-light transition.

To check whether the light-induced phosphorylation of eIF2α.2 is involved in the regulation of photomorphogenesis, we observed the light-responsive phenotypes in the mutant forms of eIF2α.2 overexpression lines under dark and different light conditions. The hypocotyl elongation was observed in 3-d-old etiolated seedlings (Fig. 8a). The eIF2α.2-8A showed longer hypocotyl elongation than eIF2α.2-WT and eIF2α.2-8D. In addition to dark treatment, we also cultured plants under long day (16 h-light/day) and red light (50 μmol m$^{-2}$ s$^{-1}$ red light exposure; 24 h-red light/day) conditions for three days (Fig. 8b, c). Under these two light conditions, we observed that eIF2α.2-8A also exhibited the longest hypocotyl elongation length. Furthermore, we found that eIF2α.2-8A had the smallest cotyledon opening angle under these two conditions. In particular, under red light conditions, eIF2α.2-8D exhibited the shortest hypocotyl length and the widest cotyledon opening angle (Fig. 8c). We also examined their cotyledon greening and chlorophyll biosynthesis phenotypes. Interestingly, 3-day-old etiolated eIF2α.2-8A seedlings showed delayed cotyledon greening (Fig. 8d, Supplementary Fig. 17). To determine whether these overexpression lines have altered levels of protochlorophyllide (Pchlide), we performed spectrofluorometric analyses of acetone extracts of 3-day-old dark-grown seedlings of each genotype. The results show that eIF2α.2-8D seedlings have relatively higher fluorescence peaks at 632 nm, indicative of Pchlide, and eIF2α.2-8A has relatively lower fluorescence compared with wild-type form seedlings (Fig. 8e). These data suggest that the phosphorylation of C-terminal eIF2α.2 also promotes photomorphogenesis.

### Phosphorylated form of eIF2α.2 promotes eIF2 ternary complex formation

Previous studies have suggested that the assembly of a multiple factor complex (MFC) is critical for translation initiation. This complex might include eIF1, eIF1A, the eIF2-GTP-Met-tRNAi ternary complex, eIF3, eIF5 and other factors. Phosphorylation of factors within this complex by protein kinase CK2 promote complex formation and subsequent enhancement in translation initiation[5,24,41]. This also implies that the assembly of eIF2 ternary complex is a crucial factor in controlling translation initiation[42]. To understand the molecular mechanism underlying the SPAs-mediated phosphorylation of eIF2α, we asked whether the phosphorylated form of C-terminal eIF2α.2 might affect eIF2 ternary complex assembly through mediating the interacting efficiencies with eIF2β and eIF2γ. To prove this hypothesis, we performed both semi-in vivo co-IP and in vitro pull-down assays. For semi-in vivo co-IP assay, we used eIF2α.2-GFP immunoprecipitated from the overexpression lines described above (Fig. 9a, b). For in vitro pull-down assay, we used purified fusion proteins, GST-eIF2α.2 (Supplementary Fig. 18a, b). When performing co-IP using anti-GFP antibody, more eIF2β protein was co-immunoprecipitated in the eIF2α.2-8D line than that in the eIF2α.2-8A line (Fig. 9a, e). Interestingly, in our semi-in vivo co-IP assay with eIF2γ, almost no eIF2γ-His were precipitated by eIF2α.2-8A, whereas similar amount of eIF2γ was precipitated by eIF2α.2-8D compared with wild-type form of eIF2α.2-GFP (Fig. 9b, e). In our in vitro pull-down assay using purified fusion proteins, eIF2β-His and eIF2γ-His, each of the recombinant GST-fused eIF2α.2-WT, eIF2α.2-8A, and eIF2α.2-8D proteins was precipitated by eIF2β-His and eIF2γ-His. The in vitro pull-down assay showed a similar pattern to semi-in vivo co-IP assay. The phospho-mimicking form of eIF2α.2-8D exhibits a stronger interaction with eIF2β compared with the phospho-null form,

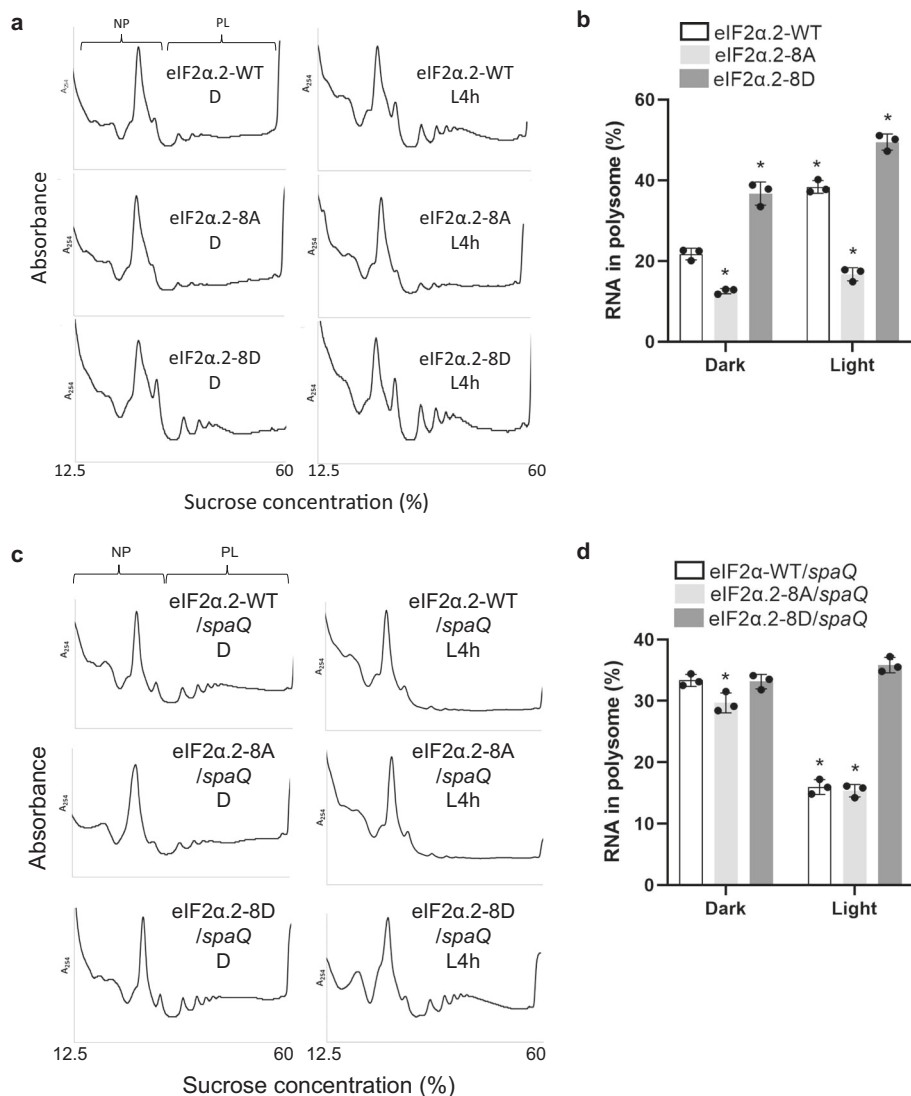

**Fig. 7 | The phosphorylation of C-terminal eIF2α by SPAs promotes translation.** The plots of polysome profiling showed different patterns of ribosomal RNA between eIF2α.2-WT, eIF2α.2-8A, and eIF2α.2-8D transgenic lines in Col-0 (**a**) and *spaQ* (**c**) backgrounds. Extracted ribosomal RNA were fractionated by 12.5–60% sucrose gradient. The positions of non-polysomal (NP) and polysomal (PL) RNA are indicated on the profiles. Four-day-old etiolated seedlings in Col-0 background (**a**) and five-day-old infected etiolated seedlings in *spaQ* background that transiently express eIF2α.2 variants (**c**) were incubated either under dark (D) or illuminated with 4 h white light (L4h) before sampling. **b**, **d** The bar charts show the percentage of polysomal RNA in total RNA in eIF2α.2-WT, eIF2α.2-8A, and eIF2α.2-8D transgenic lines in Col-0 (**a**) and *spaQ* (**c**) backgrounds. Error bars indicate the mean ± SD (*n* = 3 biological repeats). The expression of spike-in RNA (Daptomycin) was used for data normalization. Asterisks indicate translation efficiency of conditions statistically different from that of eIF2α.2-WT etiolated seedlings under dark condition. **p*-value < 0.01, two-sided Student's *t* test.

whereas the phospho-null form of eIF2α.2-8A has a much weaker interaction with eIF2γ compared with the phospho-mimicking form (Supplementary Fig. 18a–c). To provide evidence that SPA kinases regulate translation via affecting eIF2 complex formation, we checked the eIF2α-eIF2β interaction in *spaQ* background by performing semi-in vivo pull-down assay using GST-eIF2α.2 purified from *E. coli* mixed with plant total extract from Col-0 or *spaQ* seedlings. The results showed that GST-eIF2α.2 has stronger interaction with native eIF2β in the light-treated Col-0 sample, whereas the interaction intensity between GST-eIF2α.2 and eIF2β in *spaQ* was too low to be detected (Supplementary Fig. 18d). To further verify the complex formation in planta, we also checked the eIF2α-eIF2β and eIF2α-eIF2γ interaction in Col-0 and *spaQ* background by performing in vivo co-IP assay using AGROBEST technique. Five-day-old etiolated seedlings of Col-0 and *spaQ* were transfected with GFP-conjugated eIF2β and eIF2γ for transient expression. The results showed that in the light-treated Col-0 sample, eIF2β and eIF2γ exhibited a stronger interaction with native eIF2α. In contrast,

the interaction intensity between native eIF2α and eIF2β or eIF2γ in light-treated *spaQ* significantly decreased. (Fig. 9c, d, f). These results suggest that SPAs participate in the C-term-phosphorylation of eIF2α.2 to promote the formation of eIF2 ternary complex.

## Discussion

The mechanism of how reversible phosphorylation of eIF2α serves as an important regulatory pathway for the inhibition of protein synthesis has been established in mammalian and yeast cells. However, plants seem to constitute a significant exception. Recently, Zhigailov et al. reported that the conserved Ser52 residue of wheat eIF2α could not be phosphorylated under salt, oxidative, or heat stress conditions[43]. Moreover, phosphorylation of eIF2α-Ser52 by heterologous recombinant human protein kinase, HsPKR, or by endogenous protein kinase TaGCN2 does not essentially inhibit mRNA translation in wheat germ cell-free system[43]. Although eIF2α was reported to be a target of GCN2, the results largely rely on the phospho-eIF2α (Ser51) specific

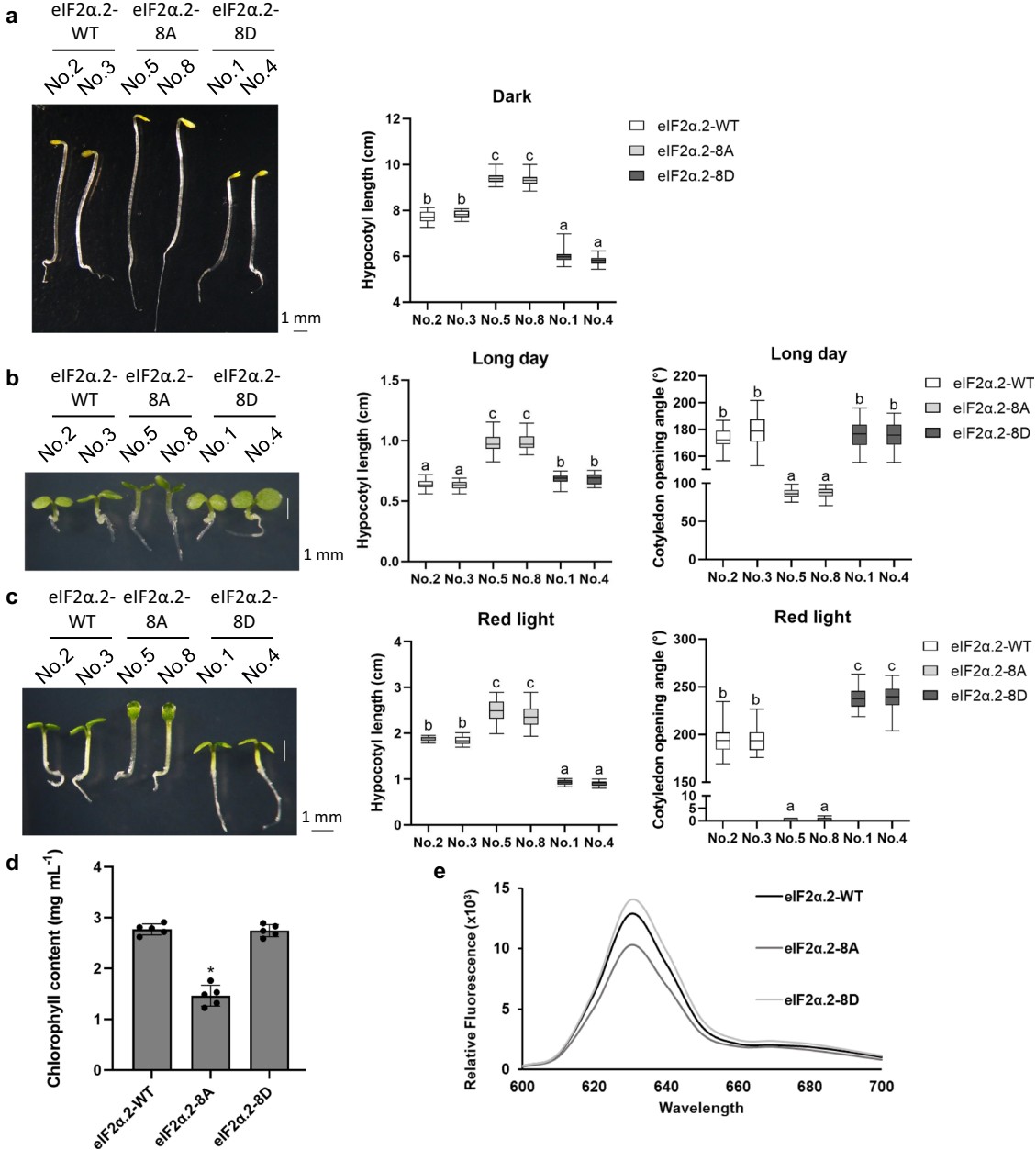

**Fig. 8 | Phosphorylation of the C-terminal domain of eIF2α promotes photomorphogenesis. a** Phenotypes of 3-day-old eIF2α.2-WT, eIF2α.2-8A, and eIF2α.2-8D transgenic seedlings under dark condition. **b** Phenotypes of 3-day-old eIF2α.2-WT, eIF2α.2-8A, and eIF2α.2-8D transgenic seedlings under long day condition (16 h-light/day). **c** Phenotypes of 3-day-old eIF2α.2-WT, eIF2α.2-8A, and eIF2α.2-8D transgenic seedlings with 50 μmol m²s⁻¹ red light exposure (24 h-red light/day). Scale bar represents 1 cm. Bar graph showing the hypocotyl lengths or cotyledon opening angle of eIF2α.2-WT, eIF2α.2-8A, and eIF2α.2-8D transgenic lines grown under different illumination conditions. The data represent means ± SD ($n = 30$) of biological replicates. Letters above the bars indicate significant differences ($p$-value < 0.01), as determined by one-way ANOVA with Tukey's post hoc analysis. For the box and whisker plots (**a**–**c**), the boxes represent from the 25th to the 75th percentile and the bars equal the median values. **d** Chlorophyll contents in 3-day-old dark-grown eIF2α.2-WT, eIF2α.2-8A, and eIF2α.2-8D transgenic seedlings sampling after 4 h white light illumination. Error bars indicate the mean ± SD ($n = 5$ biological repeats). *$p$-value < 0.01, two-sided Student's $t$ test. **e** Relative fluorescence of protochlorophyllide in 3-day-old dark-grown eIF2α.2-WT, eIF2α.2-8A, and eIF2α.2-8D transgenic seedlings.

antibodies[20,44,45] and the role of GCN2-mediated phosphorylation in the promotion of translation during photomorphogenesis is still unclear. These observations raise the question of whether the phosphorylation of this conversed residue (Ser56 for Arabidopsis) in plant eIF2α is indeed related to translational repression like it is in yeast and mammals. Previous studies have also shown that the residues other than Ser52 in eIF2α along with other initiation factors can be phosphorylated by protein kinase CK2 and thus promote complex formation and subsequent enhancement in translation initiation[5,24,41]. Here, we provide the first in vivo evidence that the non-canonical C-term-

phosphorylation of eIF2α mediated by SPA1 protein kinases play significant positive role in light-induced translational regulation (Fig. 7a, b).

We show that the light-triggered enhancement of global translation is defective in *spaQ* mutants, and the kinase activity of SPAs is crucial for the translational regulation (Figs. 1 and 6). The transcriptome and translatome analyses reveal that SPAs-mediated translational changes target mRNAs function in light responses, photosynthesis, and chlorophyll biosynthesis (Fig. 2). We have confirmed that SPA1 can interact with both eIF2α homologs with its kinase

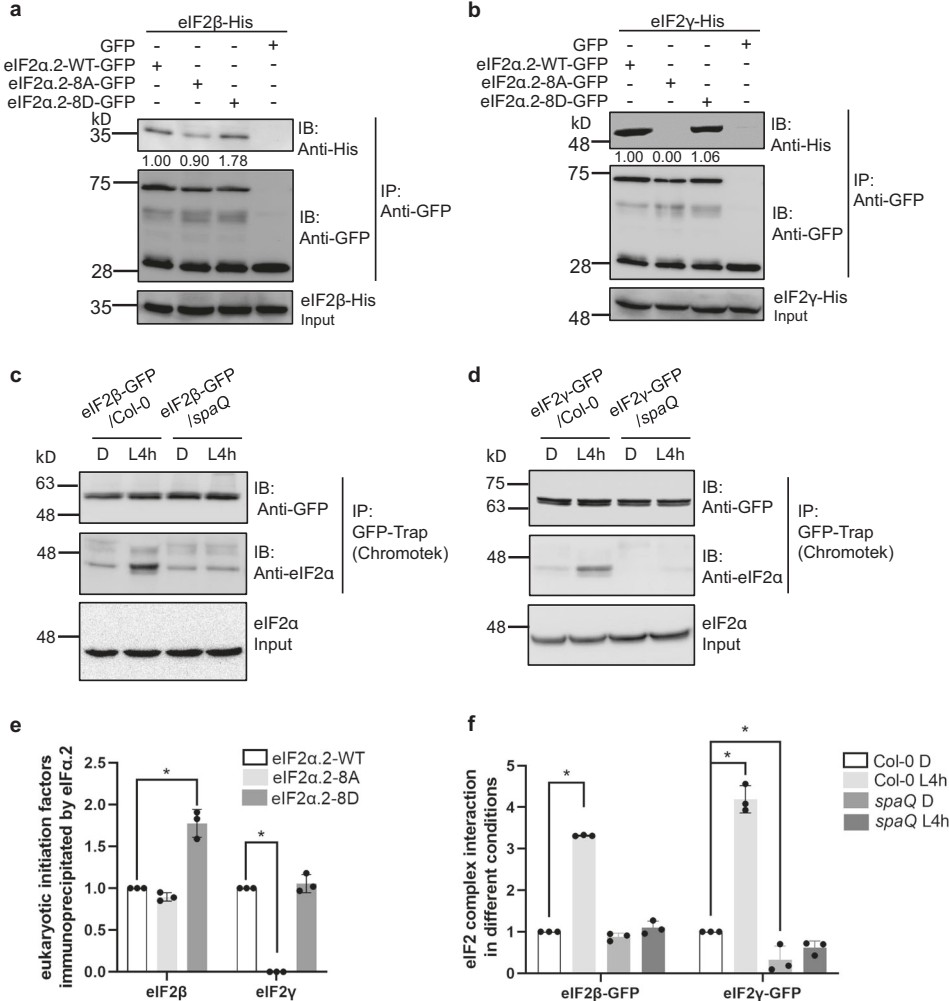

**Fig. 9 | Light-induced phosphorylation of eIF2α.2 promotes eIF2 complex formation.** Semi-in vivo co-immunoprecipitation assay showed eIF2β (**a**) and eIF2γ (**b**) have different interaction intensities with different mutative types of eIF2α.2 in planta. The different mutative types of eIF2α.2, phospho-null form (8A) and phospho-mimicking form (8D), were extracted from overexpression line in Col-0 background. Both of the eIF2β and eIF2γ proteins were expressed and purified from *E. coli*. Co-immunoprecipitation assay showed that eIF2β (**c**) and eIF2γ (**d**) have different interaction intensities with eIF2α in different backgrounds in vitro. Five-day-old infected etiolated seedlings which transiently express eIF2β (**c**) and eIF2γ

(**d**) in Col-0 and *spaQ* background were incubated either under dark (D) or illuminated with 4 h white light (L4h) before sampling. Both eIF2β and eIF2γ proteins were conjugated with GFP tag. **e** Bar graph showing the interactions between different mutative types of eIF2α.2, eIF2β and eIF2γ in vivo. **f** Bar graph showing the interactions between the subunits of eIF2 complex in Col-0 and *spaQ* backgrounds. Error bars indicate the mean ± SD (*n* = 3). Asterisks indicate the interaction intensities statistically different from eIF2α.2-WT. *$p$-value < 0.01, two-sided Student's $t$ test.

domain and directly phosphorylates the C-terminal eIF2α (Figs. 3–5). We also show that both the interaction intensity between eIF2α and SPA1 and the phosphorylation activity could be enhanced by light (Figs. 3 and 5). To clarify the function of C-term-phosphorylated eIF2α.2, we generated the mutative form of eIF2α.2, phospho-null form (8A) and a phospho-mimicking form (8D). In contrast with eIF2α.2-8D, eIF2α.2-8A seedlings show lower translation efficiency and were less sensitive to light during de-etiolation (Figs. 7 and 8). Finally, we provide evidence that the phosphorylation status of eIF2α.2 modulates its interaction strength with eIF2β and eIF2γ, ultimately influencing the assembly capacity of the eIF2 ternary complex. Taken together, we proposed a model showing that SPAs kinases phosphorylate the C-terminal eIF2α to regulate translation efficiency during photomorphogenesis (Fig. 10).

The two homologs of eIF2α have different transcript levels in *Arabidopsis* Col-0 ecotype. *eIF2α.2* mRNA level is about 7 times higher than *eIF2α.1*[46]. The two T-DNA insertion lines of eIF2α.2 studied here are both embryogenic lethal, whereas the two T-DNA insertion *eif2α.1* mutants can grow normally (Supplementary Fig. 13), suggesting that

eIF2α.1 could not complement the function of eIF2α.2. According to Cho et al.[46], another T-DNA insertion mutant of each eIF2α homolog can also grow and fertilize normally[46]. This might be because that the T-DNA insertion line they use for eIF2α.2 still has some leaky expression. However, this eIF2α.2 mutant line is smaller and has much less total eIF2α protein level than eIF2α.1 mutant. These observations imply that eIF2α.2 might play a more important role in embryogenic development or in vegetative growth. Future investigation is needed to identify the specific roles of these two homologs in their biochemical activities and the biological functions.

A recent study has shown that *cop1* mutants display a significant increase in RNAs in the PL fraction in darkness compared with Col-0. Moreover, the light treatment did not further increase the translation in the *cop1* mutant[38]. They suggested that in the dark, COP1 represses the TOR- RPS6 pathway and thus inhibit global translation in the dark. In our polysome profile analyses, *spaQ* mutant also shows a significant increase in PL-RNA level in darkness similar to *cop1* mutant (Fig. 1). COP1 might need SPAs as co-factor to co-inhibit the auxin and TOR-S6K-RPS6 signaling pathways in the dark. However, when exposed to

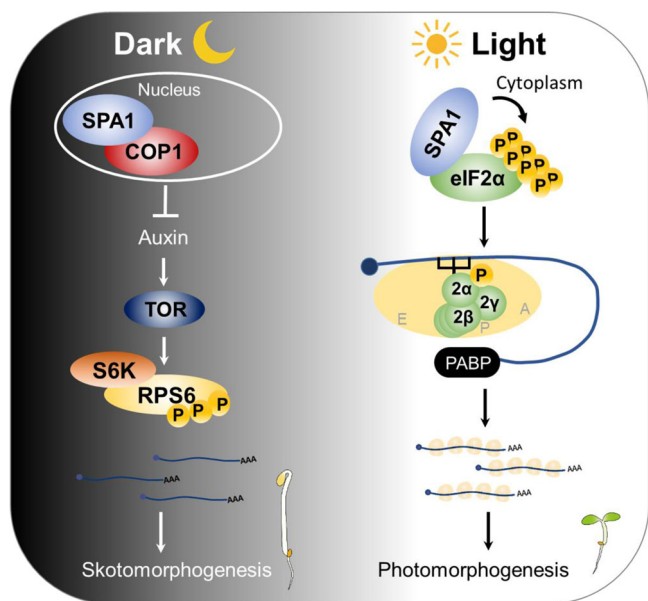

**Fig. 10 | A model showing the translation mechanism regulated by SPAs during photomorphogenesis.** Left, in the dark, SPA1 forms the E3 ligase complex with COP1 to prevent photomorphogenesis. COP1-SPA1 complex negatively regulates protein translation by repressing TOR-RPS6 module through auxin signal transduction. Right, upon light exposure, SPA1 interacts with eIF2α and triggers the light-induced phosphorylation of C-terminal eIF2α in cytoplasm. The phosphorylated form of eIF2α promotes the formation of eIF2 complex and initiates global translation and thereby promotes photomorphogenesis.

light, *spaQ* seedlings showed much lower translation efficiency than both Col-0 and *cop1* mutant (Fig. 1), suggesting that SPAs might play different roles under dark and light conditions. Upon exposure to light, SPAs not only induce eIF2α phosphorylation but are also likely essential for promoting the TOR-RPS6 pathway by phosphorylating PIFs, which subsequently leads to their degradation. PIFs can act as cofactors, enhancing the E3 ligase activity of the COP1-SPA complex[47,48], thereby inhibiting the TOR-RPS6 pathway. This dual roles of SPAs may explain the substantial reduction in translation observed in *spaQ* when exposed to light. COP1-SPA complex has been shown to display dual (both positive and negative) functions in various processes including photomorphogenesis and thermomorphogenesis[34–36]. Our data suggest that, upon light exposure, SPAs could interact with and phosphorylate eIF2α to enhance the translation efficiency. Previous studies have reported that SPA-mediated phosphorylation of PIF1 and PIF4 are also light-induced[34,35], whereas the phosphorylation of HY5 by SPAs is light-independent[36]. This indicates that the C-term-phosphorylation of eIF2α might resemble that of PIF1 and PIF4, and SPAs might have other interactors that determine its activity upon light exposure. Whether SPAs could also regulate translation through other mechanisms, such as repressing the TOR-S6K-RPS6 pathway, interacting with other eIFs to form the MFC, or regulating the assembly of eIF2 as suggested in effector-triggered immunity[42], awaits further investigation.

## Methods
### Plant materials and growth conditions
The ecotype of *Arabidopsis thaliana* we used as wild-type is Columbia (Col-0). TAP-SPA1, LUC-SPA1/*spaQ* and LUC-mSPA1/*spaQ* transgenic plants were reported previously[33,34,36] and were kindly provided by Dr. Enamul Huq. The T-DNA insertion mutants, SALK_144009 and SAIL_532_H01 for eIF2α.1 (AT5G05470), SAIL_864_B04 and SAIL_1156_D08 for eIF2α.2 (AT2G40290), were obtained from Arabidopsis Biological Resource Center. The genotypes of mutants were

screened by PCR of genomic DNA with gene-specific primers to map the T-DNA insertion sites. For analysis of these T-DNA insertion mutants, the primers we used are described in Supplementary Table 1. The gene-specific primers were performed with LBb1.3 primer for SALK line and LB3 primer for SAIL lines. The resulting PCR fragments were sequenced and used to map the T-DNA insertion site. The plants were grown in 16: 8 h, light: dark photoperiod at 22 °C.

### Vectors construction and generation of transgenic plants
To generate the eIF2α.1/eIF2α.2-WT-GFP transgenic lines, eIF2α.1/eIF2α.2 full length, truncated forms (N-terminal, middle fragment and C-terminal), eIF2β and eIF2γ transient expression lines, coding sequence of eIF2α.1/eIF2α.2, eIF2β and eIF2γ were first cloned into pENTR/D-TOPO vector and recombined into pEarleyGate103 with LR reaction using Gateway™ LR Clonase™ II Enzyme mix (Invitrogen™, Catalog #11791020). The mutant versions of eIF2α.2-8A-GFP and eIF2α.2-8D-GFP were generated using QuickChange Lightning Site-Directed Mutagenesis Kit (Agilent, Catalog #210518) in pEarleyGate103-eIF2α.2-WT. Eight amino acids were changed into alanine (eIF2α.2-8A) or aspartic acid (eIF2α.2-8D), including Y251, T257, T275, T280, S295, S317, S322, and T329. The primers we used are described in Supplementary Table 1. The constructs of pEarleyGate103-eIF2α.2-WT/8A/8D were transformed into the *Agrobacterium* strain GV3101 and then transformed into Col-0 by floral dipping.

### Isolation of total RNA and polysomal RNA
Polysomal RNA was isolated as described in Liu et al.[1]. Four-day-old etiolated seedlings were incubated either under dark or illuminated under white light for 4 h before sampling with liquid nitrogen. The 500 μL of ground sample powders were resuspended by 500 μL of polysome extraction buffer (200 mM Tris-HCl, pH 8.0, 50 mM KCl, 25 mM MgCl$_2$, 50 μg/mL cycloheximide, 100 μg/mL heparin, 2% PTE, 10% DOC, 400 U/mL RNasin) and centrifuged at $12,000 \times g$ at 4 °C for 10 min. The supernatants were filtered with 100 μm cell strainers and loaded to 11 mL of sucrose gradient (12.5–60%) and then were ultra-centrifuged at $210,000 \times g$ at 4 °C for 3.5 h. The OD$_{254}$ of gradients were detected by UV detector to illustrate the pattern of ribosomal RNA profiling. After the centrifugation, the NP and PL fractions were separately collected and extracted with PCI method[1] after equal amount of spike-in RNA (GeneChip™ Poly-A RNA Control Kit, Applied Biosystems, Catalog #900433) was added to each fraction. The extracted RNAs of NP and PL fractions were precipitated by LiCl, and resuspended with Ultra-Pure H$_2$O. The expression of spike-in RNA, Daptomycin (DAP) was used for normalization. The PL% is the percentage of the polysomal RNA relative to the sum NP and PL RNA.

### RNA sequencing and data analysis
For transcriptome and translatome analysis, three independent biological repeats of 4-day-old dark-grown seedlings of Col-0 and *spaQ* were harvested after 0 min and 4 h of light treatment and immediately frozen in liquid nitrogen for polysome fractionation and RNA extraction. The total RNAs were extracted using the pine tree method[49]. TruSeq Stranded mRNA Library Prep Kit (Illumina, San Diego, CA, USA) was used to generate mRNA sequencing library according to the manufacturer's instructions. The constructed library was quality checked by Agilent 2100 Bioanalyzer (Waldbroon, Germany) and sequenced on Illumina NovaSeq 6000 platform. For the RNA-seq data analysis, the quality of raw reads (FASTQ) was initially checked using bcl2fastp (Illumina) (v2.20) before aligned to the Arabidopsis genome using HISAT2 calling Bowtie2 (v2.1.0)[50]. Arabidopsis genome was acquired from TAIR10.1 (https://www.arabidopsis.org/). Feature-Counts (subread v2.0.1) was used to generate counts of reads mapped to annotated Arabidopsis genes. The raw reads in each batch were analyzed to give the expression ratio of genes between Col-0 and *spaQ*

mutants under dark and light conditions for the total RNA and polysome RNA samples using the StringTie (StringTir v2.1.4) and DEseq (DEseq v1.39.0). For comparison between light and dark conditions, the FPKM values were normalized by the 75th percentile relative FPKM value for total RNA and polysome RNA within the treatment and control. The genes that showed significant difference ($P < 0.05$, |FC|≥8 for mRNA$_{SS}$ or $P < 0.05$, |FC|≥8 for mRNA$_{PL}$) between Col-0 and *spaQ* mutants were further clustered. Enrichment analysis (Gene Ontology) was implemented using the Database for Annotation, Visualization and Integrated Discovery (DAVID) v6.8.

## Yeast two-hybrid

For the bait proteins, the full-length of eIF2α.1 was PCR and cloned into EcoRI and BamHI sites in pGBT9 vector, which contains the DNA-binding domain (BD). Likewise, the full-length of eIF2α.2 was PCR amplified and cloned into EcoRI and NcoI sites in pGBT9 vector. For the prey proteins, the full-length and truncated forms of SPA1 were PCR amplified and cloned into SmaI and SalI sites in pGAD424 vector, which contains the transcriptional activation domain (AD). The constructs were co-transformed into AH109 using the EZ-Yeast transformation kit (MP, Catalog #112100200) and selected on SD-Leu-Trp media at 30 °C for 3 days. Then, the selected single colony was inoculated into liquid SD-Leu-Trp media and incubated at 30 °C overnight. Finally, the overnight cultures were serial diluted into 3 concentrations, $10^{-1}, 10^{-2}$ and $10^{-3}$, and dropped on SD-Leu-Trp and SD-Leu-Trp-His medium and incubated at 30 °C for 2 and 3 days.

## Co-Immunoprecipitation (Co-IP) and semi-in vivo co-IP

For analyzing the interactions of eIF2α homologs with TAP-SPA1 under dark and light-treated conditions, four-day-old eIF2α.1/eIF2α.2-GFP and TAP-SPA1 overexpressed etiolated seedlings were used for IP. For cytosolic TAP-SPA1 interacting with native form of eIF2α after light treatment, four-day-old Col-0 and TAP-SPA1 overexpressed etiolated seedlings were used for separation of nuclear and cytosolic proteins and IP. For semi-co-IP, four-day-old eIF2α.1/eIF2α.2-GFP overexpressed etiolated seedlings and the eIF2β-His-MBP and eIF2γ-His fusion proteins purified from *E. coli* BL21 strain were used for IP. Four-day-old etiolated seedlings were incubated either under dark or illuminated under white light for 4 h before sampling with liquid nitrogen. The collected etiolated seedlings were homogenized into powder. Total proteins were extracted by adding equal amount of IP buffer (50 mM Tris-HCl, pH 8.0, 150 mM NaCl, 0.1% Tergitol NP-40, 10% glycerol, 250 mM DTT, 1x protease inhibitor cocktail, 1x phosphatase inhibitor cocktail) and centrifuged at 12,000 × *g* at 4 °C for 10 min. To IP the GFP-conjugated proteins, the lysate was incubated with 25 μL of GFP-Trap magnetic agarose (Chromotek, Catalog #gtma-20) or anti-GFP antibody (Abcam, Catlog #ab6556) bound to dynabeads (15 μL/μg antibody; Life Technologies) at 4 °C for 2 h. To IP the TAP-conjugated proteins, the lysate was incubated with anti-cMyc (Sigma, Catlog #C3956) bound to dynabeads at 4 °C for 2 h. After 2 h incubation, the beads with immunoprecipitated proteins were washed three times with IP buffer and collected on a magnetic rack. Then, 2X sample buffer was added to mix with immunoprecipitated proteins and boiled at 98 °C for 10 min. The boiled samples were separated on 10% SDS-PAGE gel and blotted with corresponding antibodies.

## Fractionation of nuclear and cytosolic proteins

Four-day-old TAP-SPA1 or Col-0 etiolated seedlings were incubated either under dark or illuminated under white light for 4 h before sampling with liquid nitrogen. The collected etiolated seedlings were homogenized into powder. About 0.1 g of sample powder were extracted by adding 150 μL nuclei isolation buffer (250 mM sucrose, 15 mM PIPES, pH 6.8, 5 mM MgCl$_2$, 60 mM KCl, 15 mM NaCl, 1 mM CaCl$_2$, 0.9% Triton X-100, 250 mM DTT, 1x protease inhibitor cocktail, 1x phosphatase inhibitor cocktail) and filtered with 100 μm cell

strainers. To separate the nuclear and cytosolic proteins, the filtered extracted sample was centrifuged at 12,000 × *g* at 4 °C for 10 min. The cytosolic proteins are in the supernatant of the centrifuged sample. And the pellet was washed three times with nuclei isolation buffer. The pellet was resuspended by adding equal amount of nuclei lysis buffer (50 mM HEPES, pH 7.5, 150 mM NaCl, 1 mM EDTA, 1% SDS, 0.1% DOC, 1% Triton X-100, 250 mM DTT, 1x protease inhibitor cocktail, 1x phosphatase inhibitor cocktail) to obtain nuclear proteins. The nuclear and cytosolic proteins were separated on 10% SDS-PAGE gel and blotted with corresponding antibodies. Anti-tubulin (Sigma, Catlog #T5168) and anti-histone H3 (Abcam, Catlog #ab1791) antibodies were used to detect marker proteins for cytosolic and nuclear fractions, respectively.

## Microscopy of GFP-conjugated protein and BiFC

For SPA1-GFP construct, full-length SPA1 coding sequence was cloned into pEarleyGate103 vector. Fifteen μg of resulted plasmid was transfected into Arabidopsis protoplasts for transient expression. The method of polyethylene glycol (PEG)-mediated transfection of protoplast was described by Yoo et al.[51]. For the BiFC assay verifying the interaction between SPA1 and eIF2α homologs, full-length coding sequence of SPA1 was cloned into modified pEarleyGate201 vector to generate SPA1-YN (N-terminal yellow fluorescence protein). Full-length eIF2α.1/eIF2α.2 coding sequence was cloned into modified pEarleyGate202 vector to generate eIF2α.1/eIF2α.2-YC. The vectors were modified by Lu et al.[52]. Ten μg of each of the two resulted plasmids were co-transfected into Arabidopsis protoplasts for transient expression. The same polyethylene glycol (PEG)-mediated transfection method was used. The green and yellow fluorescence were observed with a laser scanning confocal microscope (Leica TCS SP5, Taipei, Taiwan).

## Protein purification from *E. coli*

The full-length coding sequences of eIF2α.1 and eIF2α.2 and their truncated fragments were subcloned into pASK75 vector, then their full-length and truncated sequences were subcloned into pGEX-4T-1 with Strep sequence. The mutative versions of eIF2α.2-8A and eIF2α.2-8D C-terminus were generated using QuickChange Lightning Site-Directed Mutagenesis Kit (Agilent, Catalog #210518) in pGEX-4T-1 with Strep sequence. Eight amino acids were changed into alanine (eIF2α.2-8A) or aspartic acid (eIF2α.2-8D), including Y251, T257, T275, T280, S295, S317, S322, and T329. The primers we used are described in Supplementary Table 1. The plasmids were transformed into BL21 (DE3) cells. Protein expression was induced under 16 °C overnight with 0.1 mM IPTG and then collected the cells by centrifugation. The collected cells were sonicated in binding buffer (100 mM Tris-HCl, pH 7.5, 150 mM NaCl, 1 mM EDTA, 0.1% Tergitol NP-40, 1 mM PMSF, and 1x protease inhibitor cocktail) and purified by Strep-Tactin column (IBA, Catalog #2-1209-001). Columns were washed thoroughly with 10X resin volume of binding buffer. Target proteins were eluted with elution buffer (100 mM Tris-HCl, pH 7.5, 150 mM NaCl, 1 mM EDTA, 2.5 mM d-Desthiobiotin) into different fractions. The eluted proteins were used in kinase assays.

## SPA1 purification from *Pichia pastoris*

The pPIC3.5K-SPA1-strep[34] was transformed into *Pichia pastoris* strain GS115 using the Pichia expression kit (Life Technology, Pichia Expression Kit, Cat. #K1710-01). The transformants were selected on MM medium (1x YNB, 1x biotin, 1.8% agar) incubated at 30 °C for two days. The selected colony was inoculated in 10 mL of liquid MM medium for overnight, and then amplified into 200 mL of liquid MGY medium (1x YNB, 1x biotin, 2% glycerol) and incubated at 30 °C for 24 h. By cell resuspended and diluted into 1 L of liquid MM medium with 5% methanol, SPA1 expression was induced for 24 h at 30 °C. The cells were collected and resuspended in lysis buffer (100 mM Tris-HCl, pH

7.5, 150 mM NaCl, 0.1% Tergitol NP-40, 1 mM EDTA, 1x Protease inhibitor cocktail, 1 mM PMSF), and dropped into liquid nitrogen to make ice droplets. The ice droplets were ground thoroughly using pestle and mortar. The fine powder was collected and thawed in a 1.5 mL tube and centrifuged at $20,200 \times g$ for 20 min at 4 °C. The SPA1-strep proteins were purified by Strep-Tactin column (IBA, Catalog #2-1209-001). Columns were washed three times thoroughly with 10 mL of binding buffer. Target proteins were eluted with the elution buffer (100 mM Tris-HCl, pH 7.5, 150 mM NaCl, 1 mM EDTA, 2.5 mM d-Desthiobiotin) into different fractions. The eluted protein was used in kinase assays.

### In vitro kinase assay
About 500 ng of SPA1 and the purified GST-eIF2α.1-Strep or GST-eIF2α.2-Strep and their truncated form proteins were used in the kinase assay. All reactions were performed in kinase buffer (50 mM Tris-HCl, pH 7.5, 10 mM $MgCl_2$, 4 mM β-mercaptoethanol, 1 mM EDTA) with 10 μCi $^{32}$P radio-labeled gamma-ATP (Perkin Elmer Cat#BLU502A) and incubated at 30 °C for one hour. Then, 6X sample buffer was added to stop the reactions, and all the reactions were loaded and separated in the 4–12% SDS-PAGE gradient gel. Equal loading amounts of the substrates and kinase were added in the kinase assay. Coomassie blue staining was performed after the electrophoresis, and the dyed gel was used in Phospho-mapping.

### Sequence alignment
The amino acid sequences of different eIF2α were collected from National Centre for Biotechnology Information Nucleotide Database (NCBI). The full sequences were aligned through MUltiple Sequence Comparison by Log-Expectation (MUSCLE) using the Jalview[53].

### Semi-in vivo kinase assay
GST recombinant eIF2α.2-WT, eIF2α.2-8A, and eIF2α.2-8D C-terminus proteins were expressed in *E. coli* strain BL21 and purified with the method mentioned above. Four-day-old TAP-SPA1 and *spaQ* seedlings were incubated either under dark or illuminated under white light for 4 h before harvesting with liquid nitrogen. The collected etiolated seedlings were homogenized into powder. Total proteins of etiolated seedlings were extracted with kinase/phosphatase buffer (25 mM Tris-HCl, pH 7.5, 1 mM DTT, and 5 mM $MgCl_2$), plus protease inhibitor, 250 mM DTT, and 1 mM PMSF. The protein extracts were used in kinase assay to incubate with the purified GST and recombinant eIF2α.2, eIF2α.2-8A, and eIF2α.2-8D C-terminus proteins. Semi-in vitro kinase assay with plant extracts were performed and modified from previous study[54,55]. For Fig. 5b, c, 2 μg of GST only, or mutative eIF2α.2 proteins and 25 μg of plant seedling extracts were mixed in kinase/phosphatase buffer with protease inhibitor, 1 mM PMSF, and 10 μCi 32 P radio-labeled gamma-ATP (Perkin Elmer, Cat#BLU502A) in a total volume of 40 μL. The reaction was incubated at 30 °C for 1 h, and the reactions were stopped by adding sample buffer and boiling at 65 °C for 5 min. The products were separated by electrophoresis using 4–12% gradient gels. The gels were stained, dried, and then visualized by exposure to chemiluminescent detection films.

### eIF2α mobility shift assay
To observe eIF2α mobility shift in Col-0 and *spaQ*, the total proteins of five-day-old etiolated Col-0 and *spaQ* seedlings were extracted with extraction buffer (100 mM Tris-HCl, pH 6.8, 20% glycerol, 5% SDS, 20 mM dithiothreitol, 1 mM PMSF, 250 mM DTT, 1x protease inhibitor cocktail, 1x phosphatase inhibitor cocktail). The extracts were cleared by centrifugation and then added the activated calf intestinal alkaline phosphatase (CIP) or inactivated boiled CIP as control. The mixtures were incubated at 37 °C for 1 h. The reaction was terminated by adding 6x SDS sample buffer and boiling at 65 °C for 10 min. The total proteins were separated in 6.5% SDS-PAGE containing 15 μL Phos-tag acrylamide (Wako Pure Chemical Industries, Cat# AAL-107). Immunoblotting

analyses were performed with anti-eIF2α antibody, which was kindly provided by Dr. Browning as reported by Dennis and Browning, 2009[23], anti-p56-eIF2α antibody (Cell Signaling, Cat#9721) and anti-Rpt5 antibody (ENZO, Cat# BML-PW8770).

### Agrobacterium infection in Arabidopsis seedlings
AGROBEST infection assays were conducted as described in Wu et al.[56] and as following. The disarmed strain of *Agrobacterium tumefaciens* C58C1 with the construct containing the target protein for infection, pEarleyGate103- was freshly streaked out from a −80 °C glycerol stock onto a LB agar plate for a 2 days incubation at 28 °C. A fresh single colony from the plate was used to inoculate 5 mL of LB liquid medium containing appropriate antibiotics for shaking (220 rpm) at 28 °C for 20–24 h. For pre-induction of gene expression, *A. tumefaciens* cells were pelleted and resuspended at an $OD_{600}$ of 0.2 in AB-MES liquid medium (17.2 mM $K_2HPO_4$, 8.3 mM $NaH_2PO_4$, 18.7 mM $NH_4Cl$, 2 mM KCl, 1.25 mM $MgSO_4$, 100 μM $CaCl_2$, 10 μM $FeSO_4$, 50 mM MES, 2% glucose (w/v), pH 5.5) with freshly added 200 μM acetosyringone without antibiotics and then shaken (220 rpm) at 28 °C for 12–16 h. Before infection, *A. tumefaciens* cells were pelleted and resuspended in infection liquid medium, which is one-to-one ratio of MS and AB-MES liquid medium with freshly added 200 μM acetosyringone, to an $OD_{600}$ of 0.02. Col-0, TAP-SPA1 and *spaQ* were grown in continuous dark condition at 22 °C. Three-day-old etiolated seedlings were transplanted into 1 mL *A. tumefaciens* cells in infection liquid medium and incubated in the same growth condition for 2 days. Five-day-old infected etiolated seedlings were then used for polysome profiling, ribosomal RNA extraction and Co-IP.

### Measurement of hypocotyl length
For measurement of hypocotyl length, seedlings were grown on MS agar plates. Seeds were first stratified at 4 °C in darkness for 4–7 days, then germination induced by incubation in white light (70 μmol m$^{-2}$ s$^{-1}$) at 22 °C for 4 h. Seedlings were then transferred to either all dark condition or dark condition with 30 min light treatment each day. The hypocotyl length of at least 30 individual seedlings was measured at the fourth day following germination induction.

### Cotyledon greening and protochlorophyllide measurement
Surface-sterilized seeds were plated on MS agar (1% phytoagar, pH 5.7) and imbibed for 3 days at 4 °C in the dark before germination was induced by incubation in white light (70 μmol m$^{-2}$ s$^{-1}$) at 22 °C for 4 h. Seeds were then transferred to darkness for 3 days. After dark incubation, seedlings were transferred to white light (70 μmol m$^{-2}$ s$^{-1}$) at 22 °C. The phenotype of cotyledon greening was captured after 4 h illumination.

Three-day-old Arabidopsis etiolated seedlings were weighed and ground in liquid nitrogen. And the protochlorophyllide extraction was described by Cheng et al. and Runge et al.[57,58]. Three-day-old dark-grown seedlings for each overexpression line were used. Spectro-fluorometer was performed at an excitation wavelength of 440 nm and an emission wavelength of 600–700 nm.

### In vitro pull-down assay
For in vitro pull-down assay, both of the mutative GST-eIF2α.2 and eukaryotic initiation factors, eIF2β-His and eIF2γ-His fusion proteins, were expressed in *E. coli* BL21 strain. The vector construction of the mutative GST-eIF2α.2 was mentioned above. For eIF2β and eIF2γ conjugated with His tag fusion proteins, the coding sequences of eIF2β and eIF2γ were cloned into the pET-21a (+) vector. The eIF2β-His or eIF2γ-His combine with GST-eIF2α.2 were incubated with 20 μL of HisPur™ Ni-NTA Resin (Thermo, Catalog #88221) in the binding buffer (50 mM Tris-HCl, pH 7.5, 150 mM NaCl, 0.6% Tween 20, and 1 mM DTT) for 2 h, respectively. The beads were collected and washed three times with binding buffer. The bound wild type and mutative GST-eIF2α.2 proteins were detected by anti-GST-HRP conjugate (cytiva, Catalog

#RPN1236V). The blot was quantified using IMAGEJ software and normalized to added eIF2β-His and eIF2γ-His proteins. The ratio of the band of GST-eIF2α.2-WT was set to 1 for each.

## de novo protein synthesis assay

Etiolated seedlings were incubated with 100 μM L-azidohomoalanine (AHA) (Thermo Fisher Scientific, Catalog #C10102) and treated under dark or light conditions for 4 h before sampling. The proteins were extracted with extraction buffer (50 mM Tris-HCl, pH 7.0, 1% SDS, 1× protease inhibitor cocktail, 1 mM PMSF). The AHA-labeled proteins were bound with biotin-alkyne (Thermo Fisher Scientific, Catalog #B10185) using a Click-iT reaction kit (Thermo Fisher Scientific, Catalog #C10176) as the manufacturer's protocol. The proteins were pelleted using chloroform-methanol precipitation. Then, sample buffer was added to resolve the protein pellets, separated on SDS-PAGE gel, and transferred to PVDF membrane. HRP-conjugated Streptavidin (Thermo Fisher Scientific, Catalog #89880D) was used for immunoblot analyses to detect the AHA-biotin-labeled proteins.

## Fresh weight, dry weight and relative water content measurement

Surface-sterilized seeds were plated on MS agar (1% phytoagar, pH 5.7) and imbibed for 3 days at 4 °C in the dark before germination was induced by incubation under white light (70 μmol m$^{-2}$ s$^{-1}$) at 22 °C for 4 h. Seeds were then transferred to darkness for 4 days. The fresh weight of 4-day-old etiolated seedlings was measured. For dry weight measurement, etiolated seedlings were subjected to overnight incubation at 65 °C for dehydration. The dry weight of the desiccated etiolated seedlings was subsequently measured. The relative water content is calculated as the percentage difference between the fresh weight and dry weight, relative to the fresh weight.

## Antibodies

A complete list of all primary antibodies (with manufacturers, catalog numbers, and dilution factors) used throughout this study can be found in Supplementary Table 2. Polyclonal goat anti-mouse and rabbit immunoglobulin conjugated to horseradish peroxidase were used for western blot analysis at a 1:20,000 dilution (Cell signaling).

## Statistics & reproducibility

The samples of extracted ribosomal RNA, RNA sequencing and signal quantifications of western blots were measured and analyzed as biological triplicates. The phenotype quantifications, including hypocotyl length and cotyledon opening angle, were measured and analyzed as biological thirty repeats. No statistical method was used to predetermine sample size. No data were excluded from the analysis.

## Reporting summary

Further information on research design is available in the Nature Portfolio Reporting Summary linked to this article.

## Data availability

The authors declare that all data in this study are available in the figures and accompanying Supplementary Information file. The RNA sequencing raw data have been deposited in the Gene Expression Omnibus database under accession number GSE233765. Arabidopsis mutants and transgenic lines, as well as plasmids and antibodies generated in this study are available from the corresponding author upon request. Source data are provided with this paper.

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

## Acknowledgements

The authors acknowledge the mass spectrometry technical research services from NTU Consortia of Key Technologies and NTU Instrumentation Center. We acknowledge the confocal microscopy technical research services from Technology Commons in College of Life Science and the Instrumentation Center sponsored by Ministry of Science and Technology, National Taiwan University (Taiwan). We also thank Dr. Shih-Shun Lin and Hsiu-Chu Cheng for the technical support, and the Cheng lab members for critical reading of the manuscript. This work was supported by the Young Scholar Fellowship Einstein Program from the National Science and Technology Council in Taiwan under grant nos. NSTC 112-2636-B-002-011, NSTC 113-2636-B-002-007 to M.-C.C, and the National Science Foundation (MCB-2014408) to EH. M.-C.C. acknowledges the financial support from National Taiwan University.

## Author contributions

H.-H.C., K.S.B., E.H. and M.-C.C. initiated the study and designed the experiments. H.-H.C., L.-C.H. and M.-C.C. performed the experiments. H.-H.C and M.-C.C. analyzed the data. H.-H.C., K.S.B, E.H. and M.-C.C interpreted the results. H.-H.C. and M.-C.C. wrote the initial draft. K.S.B., E.H. and M.-C.C. reviewed and edited the manuscript.

## Competing interests

The authors declare no competing interests.
