## [Peer Review File · Nature Communications]

REVIEWER COMMENTS

Reviewer #1 (Remarks to the Author):

This manuscript describes SPAs as direct interactors of eIF2 alpha. In the presence of light, SPA would phosphorylate eIF2alpha in its C-ter. This phosphorylation would promote the assembly of eIF2 complex in the light and increase global translation.

The paper could potentially be very nice and of high impact showing a still missing molecular mechanism to explain the global increase in translation that plants experiment in light compared to dark. However, in my honest opinion, strong conclusions are taken from weak results, which invalidate them. Some results are too poor to be able to conclude anything from them, others experiment require better controls. Methods and constructs are poorly described, or not described at all. Some paragraphs of the method section do not even correspond to constructs or experiments employed in this manuscript. Many figures are mislabeled or have labeling issues. In my opinion, this story, potentially very nice, needs a lot more work, both experimentally and to improve the actual text.

Questions that would need discussion:

In the spaQ mutant, in the dark, translational activity seems high, and it decreases in the light. Why is that? if the proposed role of increasing translation in the light by phosphorylating eIF2 were true, why would translation decrease so much in the light compared to that in the dark? Should it not be the same as in the dark, as what happens in the cop1 mutant?

I find that the entire hypothesis built out of the interaction between SPA1 and eIF2 alpha is a bit strange. From what is stated in the manuscript, SPA1 interacts with eIF2 through its kinase domain (when SPA1 holds domains for protein-protein interactions) and the N-ter of eIF2, but it is the C-ter of eIF2 which is phosphorylated. Interactions through the kinase domain are normally very transient... But then, in the subsequent experiments, the authors use only the C-ter to check for the SPA-mediated phosphorylation to avoid the interference of the possible phosphorylation on the N-ter of eIF2. How would SPA1 interact then with eIF2 alpha if the results of the Y2H experiments are real?

The quality of the polysome profiles is very poor. No conclusion can or should be taken from them. In addition, can the authors explain the nature of the peak that appears between the peaks that they identify as 60S and 80S in Figure 1?

L113 and Figure 1A. The manuscript describes the “non-polysomal fractions” as the fractions “with no greater than 2 ribosomes”. Based on this, the disomes would be part of it. However, in the figure, the disomes fractions are considered polysomal.

Figure 1B: From the legend, data is normalized to an RNA spike. How did they do this experiment? It is not described in the methods section.

About Supp Figure 1 and the metabolic labeling: It seems that only one experiment has been performed. Also, the manuscript considers that its results parallel that of the profiles. I don't think it does. In any case, it is meaningless without, at least, two additional biological replicates with their respective quantifications.

Figure 2 and L143: How were the transcriptomic experiments performed?? Please, clarify, as in the text they appear as “RNA-sequencing” but from the legend, they are microarray experiments.

About the experiments performed to check for the interaction between eIF2 and SPA1:

a) Does the TAP-SPA1 construct also carry the Myc tag? If so, it should be stated to understand why in the co-IP experiments, anti-myc is used to detect TAP-SPA1.

b) Supp Figure 2: The figure is mislabeled. It is an anti-GST pull down, and not an MBP pull down.

Also Given that eIF2-GST is interacting with all the bands of MBP-SPA1 that also appear in the input, I believe that a very important control is missing, that is to check that MBP-SPA1 is not binding the anti-GST resin/beads. MBP alone may not bind it, but SPA1 has not been tested.

c) Also, are the tagged version of SPA and eIF2 functional? Do they complement mutant phenotypes?

d) Figure 3C and Supp fig 3: Please, describe the eIF2 construct a bit. Under which promoter is it expressed? And also the position of the tag, as in the same figure (Supp 3), on one side appears to be in the N-ter and on the other side, in the C-ter.

e) I have a problem with the co-IPs done using Agrobast. Maybe I am biased as I don't know anyone for whom this system works. Given that, in my opinion, the pull down experiment is lacking an important control, and that trusting the interaction between SPA1 and eIF2 is key to trust the entire manuscript, I would request other more “standard” method to prove the interaction. Ideally, co-IP using spa1 and eif2alpha mutants expressing the tagged versions that complement the phenotype.

Figure 4: Please, indicate the protein sizes in the panels.

4a: please, indicate the amounts of proteins used, both for SPA1 and also which are the increasing amounts of eIF2alpha.

Please, describe in the method section the part corresponding to this experiment. In the main text, SPA1 is tagged with strep and eIF2 is tagged with GST. In methods, eIF2 is tagged with both strep and GST...

4b and c: please, discuss why phosphorylation on the C-ter does not explain the amount of phosphorylation of the full-length.

Also, how many biological replicates have been performed of these experiments? Why use for eIF2alpha.2 a experiment in which the different versions of eIF2alpha are barely expressed? In addition, instead of the Coomassie gels, I would like to see WB to be able to quantify phosphorylation relative to the amount of protein produced.

Figure 5B: how was this “semi in vivo kinase” experiment done? In the legend, it says that TAP-SPA1 protein was extracted. But in the methods section, L652, it says that for figure 5b, eIF2 alpha proteins were incubated with seedling extracts. If this is so, this is no proof that SPA1 is phosphorylating eIF2 alpha. This shows that the phospho-null versions of the C-ter of eIF2 do not get phosphorylated, and that the WT version gets phosphorylated in the light, but does not show a role of SPA1 on it.

Figure 5C: I do not understand the figure... why is the amount of eIF2alpha so low spaQ in L compared to the other lanes when the Rpt5 signal is stronger? Is that super faint phosphorylation signal in Col-0 in L significant?

L281: the “significant” reduction?

L285: please, explain the mSPA1 construct that has a mutation in the kinase domain. What is this mutation? Does it lack the entire domain? Is it a nucleotide substitution?

And as in Figure 1, no conclusions can be made from these profiles.

I don't think the data allows to think that eIF2alpha 2 is “specifically” involved in embryogenesis... it can play other roles plant-wide ...

The WT, Phospho-null, and phospho-mimetic versions of eIF2 alpha should be transformed in the eif2 alpha mutants (transforming the hets) to see if they complement and so, understand the biological significance of the phosphorylation in the C-ter. In Figure 7A, if all transformants are in Col-0 background, why are the ones expressing the 2-8A version not responding normally to light.

Figure 8A. Col-0 is not etiolated. It does not show a hook and the cotyledons are opening.

Also, please, discuss why a Col-0 background would not be responding normally to light when expressing ectopically the eIF2alpha mutant versions. It can be because the mutant versions sequester the TC, but if so, eIF2 alpha het mutants should also show a similar phenotype?

Fig 9C and D: Please, draw the panels as in A and B, because it seems that GST was pulled-down and that eIF2b interacted with GST alone with equal intensity.

Methods:

Please, revise all methods as important information is missing from them and it would not be possible to replicate some experiments using the information provided.

In vitro pull-down assays: the methods refer to eIF5A and eIF4E that do not appear in the manuscript. Please, rewrite the section using the actual constructs employed in this manuscript.

Please, describe the Agrobacterium method employed here better. It is just a copy of the original paper which also doesn't describe the method very clearly. Which is the "desire cocultivation liquid media" employed in these experiments? How do you cultivate Arabidopsis seedlings in 1 ml of Agrobacterium? Which are the conditions in which you incubate Arabidopsis "in the same growth room for 2 -3 days"?

The manuscript uses "ribosome-RNA profiling" as the technique to separate polysomes along a sucrose gradient based on the number of ribosomes when its actual name is "polysome profiling".

L121: "ribosome profiling" is a synonym of "Ribo-seq", which is not what has been done. The manuscript should refer here to the polysome profiles.

Figure 3B: the Y2H panel is mislabeled. SD-Leu should be SD-Leu-Trp- and the SD-Leu-Trp- should be SD-Leu-Trp-His.

L46-47: I would not dare to say that "there is a substantial knowledge of the identities and molecular signatures associated with mRNAs regulated at the translational level". Knowledge in this field is far from substantial. The great majority of the molecular mechanisms governing translation regulation are still unknown.

Other issues:

SPA appears as SAP in many places in the manuscript.

SUPPRESSOR OF PHYA-105 (SPA1-4) also appears as SUPPRESSOR OF PHYA-105 1 (SPA1-4).

L63: I'd put and "However," before "Because". And they don't only have plant-specific translation control mechanisms. They also have a plant-specific translational machinery with plant-specific translation factors.

L137: what does "(DAP)" stands for?

Reviewer #2 (Remarks to the Author):

The study presented a novel plant-specific regulatory mechanism of eIF2alpha phosphorylation in enhancing protein translation, which is distinct from the previously established role of eIF2alpha phosphorylation in repressing global protein translation in yeast and mammals. The authors demonstrated that the SPA-mediated phosphorylation of the eIF2alpha C-terminus enhances the eIF2 complex formation, resulting in increased translation during photomorphogenesis, which provides a new molecular mechanism for the eIF2alpha phosphorylation in regulating protein translation in eukaryotes. The manuscript was well-written and logically structured. However, concerns remain outstanding regarding the mechanisms proposed in the working model and need to be addressed to improve clarity and provide convincing evidence supporting the authors' main claims before publication in Nature Communications.

#1: The author claimed that the SPA regulated protein translation via phosphorylating eIF2alpha C-terminus. However, no genetic evidence (epistasis analysis) supports this claim. Instead, they provided separate data by showing that spaQ mutant regulates protein translation (Figure 1 and 2), SPA1 could phosphorylate eIF2alpha (Figure 4 and 5), and eIF2alpha phosphorylation regulate protein translation (Figure 7) without closing the regulatory loop.

Ectopic expression of eIF2alpha phospho-variants in spaQ mutant should be tested to close the loop by checking if eIF2alpha-8D or 8A could restore spaQ's translation, enhancing translation in darkness while

inhibiting translation in light. Importantly, the transgene spaQ/eIF2alpha will also benefit the eIF2 complex assembly assay in Figure 9 and the phos-tag gel assay in Figure 5, which will be stated below.

In the proposed model (Figure 10), the SPA1 represses protein translation through eIF2 complex assembly in dark conditions. However, the translational outputs from spaQ mutant (Figure 1) and eIF2a.2-8A transgene (Figure 7) under dark conditions are opposite. Furthermore, no protein interaction was detected between SPA1 and eIF2alpha in the dark, as shown in Figure 3c. These data raised concerns that SPA-regulated translation is not through eIF2alpha phosphorylation, at least in darkness. As the author wrote in the introduction, SPA kinases repress translation in the dark might through TOR/RPS6 pathway, like COP1. It will greatly benefit the proposed model to test the spaQ/eIF2alpha phospho-variant transgenes (spaQ/eIF2alpha-WT, spaQ/eIF2alpha-8A, and spaQ/eIF2alpha-8D) to check the epistasis.

#2: The authors need to present more convincing data for the phosphorylation of eIF2alpha dependency on SPA kinases in vivo.

In Figure 5b, the semi in vivo phosphorylation of eIF2alpha should include spaQ mutant because this figure did not provide SPA1 dependency in eIF2alpha phosphorylation, and the plant extracts contain massive other kinases, which may lead to false positive results.

In Figure 5c, because the native antibodies (Please provide cat no.) can detect more than one band in the hypo-phosphorylated conditions (Dark), making it not a perfect match for the assay. Moreover, the third line, which has the slightly increased p-eIF2alpha, has more protein loading than the rest. I suggest generating the spaQ/eIF2alpha-GFP transgenes, which were proven to show a unique band in WT, to repeat the phos-tag assay. According to the multiple phosphor-sites predicted in eIF2alpha, I expect strong shifted phosphorylation bands in light and weaker ones in spaQ mutant.

#3: The claim that SPA kinases regulate translation via affecting eIF2 complex formation could greatly benefit from additional experiments to check eIF2 assembly in spaQ mutant background. For example, check the eIF2alpha.2 and eIF2beta interaction in spaQ/eIF2alpha.2-GFP transgenes. The availability of the eIF2beta native antibody from Dr. Browning's lab, one of the authors in this work, makes the experiment easy to perform.

#4: Because one of the main focuses in the manuscript is to elucidate how SPA kinases regulate translation efficiency (TE), I suggest calculating TE ($TE = \text{mRNApl fold change} / \text{mRNAss fold change}$) instead of showing both the transcriptional and translational changes to simplify the data interpretation in Figure 2, and ranking the GO analysis base on the TE values. The current data in Figure 2 is distracting because we must focus on mRNAss and mRNApl changes.

#5: Typically, the ratio of polysome (PL)/monosome (NP) in ribosomal profiling is dynamically changing, with more polysome companies with less monosome or the other way around, like the paper (38) cited in this manuscript. However, this dynamic is hard to observe in the profiling figures (Figures 1 and 6), instead showing a similar trend in the manuscript. Could the authors comment on this?

#6: The AHA labeling assay in Figure S1, spaQ mutant showing more newly translated proteins than Col-0 under light conditions, indicating SPAs negatively regulate protein translation in light conditions. Does it contradict the paper's claim that SPAs positively control protein translation under light conditions?

#7: Protein translation occurs in the cytoplasm, and SPA1 kinase was reported localized in the nucleus. How does SPA1 regulate protein translation? Do SPA kinases have cytosol localization in light conditions? Please check the interaction between SPA1 and eIF2alpha in vivo using BiFC assay.

#8: eIF2alpha-8A transgene repressed protein translation in both dark and light conditions (Figure 7); however, the transgene displayed a more extended hypocotyl elongation. Do the authors comment on this counterintuitive phenotype and how the phenotype is linked to translation?

#9: Please indicate the SPA1 protein position in Figure 4.

#10: In Figure S2, is it GST or MBP pull-down? Please confirm.

In summary, the findings described in this manuscript are potentially very interesting and impactful. However, multiple points need clarification.

Reviewer #3 (Remarks to the Author):

This manuscript by Chang et al. describes that the phosphorylation of C-terminal eIF2 α by SPA kinases promotes translation efficiency and photomorphogenesis in Arabidopsis. The authors demonstrate that

SPA1 directly interacts and phosphorylates the eIF2 α C-terminus under light conditions. They also reveal that the phosphorylated eIF2 α enhances the formation of ternary complex by promoting its affinity to eIF2 β and eIF2 γ . The C-term-phosphorylated eIF2 α promotes translation efficiency and photomorphogenesis. These findings are very interesting and provide new evidence for SPA1 kinase activity and eIF2 α phosphorylation in the regulation of translation during photomorphogenesis. The manuscript was clearly written. Most of the data provided in the current version were logically presented. This study would be considered for extensive improvement in the following points.

Major points:

- The C-term-phosphorylated eIF2 α (8D) promotes translation efficiency and photomorphogenesis in the dark, whereas the C-term-unphosphorylated eIF2 α (8A) results in decreased translation efficiency. spaQ is defective in the C-term-phosphorylation of eIF2 α and thus in its translation efficiency, but develops enhanced photomorphogenesis with short hypocotyls like 8D under light conditions. The photomorphogenic phenotypes of 8A and spaQ seem contradictory at the first sight. One possible explanation is that strong inhibition of the degradation of photomorphogenesis-promoting proteins in spaQ is a major cause of constitutive photomorphogenesis since SPAs are crucial components of COP1 E3 ligase complex. Therefore, it is not clear here concerning the contribution of SPA1-mediated eIF2 α phosphorylation in SPA regulation of photomorphogenesis. The reviewer suggests the authors provide necessary genetic data to specifically evaluate the function of SPA1-mediated eIF2 α phosphorylation (apart from COP1-SPAs E3 activity). For example, compare the phenotypes of Col and LUC-mSPA1/spaQ (not shown here) and discuss. Will expressing 8D rescue LUC-mSPA1/spaQ?

- spaQ exhibited enhanced translation in the dark but repressed translation upon light exposure, suggesting that SPAs play opposite roles in translation regulation under dark and light conditions. How does light signal trigger the SPA1-eIF2 α interaction and eIF2 α phosphorylation to promote translation in plants? Based on in vitro systems, the SPA1-eIF2 α interaction and eIF2 α phosphorylation is independent on light conditions. However, these two events were detected specifically under light conditions in planta. In addition, from the model figure, the in vivo association between SPA1 and eIF2 α appears to be regulated at a subcellular level by light, and the assembly of COP1-SPA complex is also affected? The reviewer suggests the authors provide necessary analyses to clarify these points which are important for understanding the molecular basis of SPA1-mediated translation regulation.

- Another concern of the reviewer associates with the functional redundancy and possible differentiated roles of SPA1-4 in eIF2 α phosphorylation and translation regulation. Do SPA1-4 play conserved roles in eIF2 α interaction and eIF2 α phosphorylation? Considering the functional redundancy of SPA1-4 and the predominant role of SPA1 in repressing photomorphogenesis, spa1 single mutant is potentially a good material to test translation defects and eIF2 α phosphorylation defects. It is a little bit difficult to understand why these two experiments were performed only using spaQ but not including spa1 single mutant.

Minor points:

- Line 158-159, Cluster 4, representing the largest group of mRNAs (4,006 genes), showed decreased translation but negligible changes in mRNASS level in spaQ under the light condition. Why the decreased translation in spaQ mRNASS was regarded as negligible? The changes are much clearer than that in the dark.

- Fig.4, SPA1 band was very vague in Fig.4b, and its migration pattern seemed different from that in Fig.4a. Because there were many unspecific or degraded fragments, it is very difficult to tell the full-length and truncated eIF2 α .1 and eIF2 α .2 proteins. The authors should indicate these proteins on Coomassie blue staining gels. mSPA1 was not included as a negative control in Fig.4c.

- Line 254-255, T295 should be S295. Please check throughout the manuscript.

- Line 314, "+, inactivated boiled CIP" should be "-, inactivated boiled CIP"?

The quantification of Fig.9c and Fig.9d was exactly the same (1.00, 0.80, 1.91). Please check and make the data convincing.

Reviewer #1 (Remarks to the Author):

Questions that would need discussion:

In the *spaQ* mutant, in the dark, translational activity seems high, and it decreases in the light. Why is that? if the proposed role of increasing translation in the light by phosphorylating eIF2 were true, why would translation decrease so much in the light compared to that in the dark? Should it not be the same as in the dark, as what happens in the *cop1* mutant?

We thank the reviewer for pointing this out. Our hypothesis is that SPAs play negative roles under dark but play positive roles in the light condition as has been shown in a recent study (Paik et al., 2019). In the dark, SPAs work together with COP1 to inhibit translation through TOR-RPS6 pathway. Thus, in *spaQ* mutant, the translation activity is high. Upon light exposure, beside the effect of light-induced eIF2 α phosphorylation by SPAs, it is possible that SPAs are also necessary to promote TOR-RPS6 pathway through phosphorylating PIFs and other transcription factors (Paik et al., 2019), thus the translation decreased dramatically in the light. We will try to elucidate this regulatory mechanism and discuss in more detail in our future studies.

We have added this postulation in the discussion part in line 580-585: "Upon exposure to light, SPAs not only induce eIF2 α phosphorylation but are also likely essential for promoting the TOR-RPS6 pathway by phosphorylating PIFs, which subsequently leads to their degradation. PIFs can act as cofactors, enhancing the E3 ligase activity of the COP1-SPA complex (Pham et al., 2018; Xu et al., 2015), thereby inhibiting the TOR-RPS6 pathway. This dual role of SPAs may explain the substantial reduction in translation observed in *spaQ* when exposed to light."

I find that the entire hypothesis built out of the interaction between SPA1 and eIF2 alpha is a bit strange. From what is stated in the manuscript, SPA1 interacts with eIF2 through its kinase domain (when SPA1 holds domains for protein-protein interactions) and the N-ter of eIF2, but it is the C-ter of eIF2 which is phosphorylated. Interactions through the kinase domain are normally very transient... But then, in the subsequent experiments, the authors use only the C-ter to check for the SPA-mediated phosphorylation to avoid the interference of the possible phosphorylation on the N-ter of eIF2. How would SPA1 interact then with eIF2 alpha if the results of the Y2H experiments are real?

We thank the reviewer for the comment. Our Y2H analysis provided a possible interaction pattern in eukaryotic cells; however, the results might not hold true among other experimental system due to possible folding effect, transient/weak interaction, or subcellular localization. In Y2H system, the proteins need to interact in the nucleus to

activate the expression of reporter gene. We have moved the Y2H data for eIF2 α fragments to Supplementary Fig. 2. To provide additional evidence that SPA1 might still interact with the C-terminus of eIF2 α , we used another system, semi-*in vivo* co-IP, using TAP-SPA1 seedlings and C-term-eIF2 α proteins purified from *E. coli*, which were the same samples used in our semi-*in vivo* kinase assay, to test their interaction. As shown in Supplementary Fig. 9 in our revised version, TAP-SPA1 could interact with the C-terminal eIF2 α when using light-treated sample, suggesting that the C-terminal eIF2 α alone can be the substrate in our kinase assay as shown in Fig. 4b and c.

The quality of the polysome profiles is very poor. No conclusion can or should be taken from them. In addition, can the authors explain the nature of the peak that appears between the peaks that they identify as 60S and 80S in Figure 1?

We thank the reviewer for pointing this out. We have replaced the previous profiles with better profile results that were performed previously but in a different batch. We have indicated the peaks of 40S, 60S and 80S in the first profile in Fig. 1.

L113 and Figure 1A. The manuscript describes the “non-polysomal fractions” as the fractions “with no greater than 2 ribosomes”. Based on this, the disome would be part of it. However, in the figure, the disomes fractions are considered polysomal.

We thank the reviewer for pointing this out. As mentioned above, we have replaced the previous profiles with better profile results that were performed previously but in a different batch. We have indicated the peaks of 40S, 60S and 80S in the first profile in Fig. 1. The small peak next to 80S on the right represents disome, so our polysomal fraction contains the part with ≥ 3 ribosomes.

Figure 1B: From the legend, data is normalized to an RNA spike. How did they do this experiment? It is not described in the methods section.

We have provided detailed information regarding the spike-in normalization method used in this study in the “Methods” section in the new version of manuscript. As described in “Isolation of total RNA and polysomal RNA” in line 631: “After the centrifugation, the NP and PL fractions were separately collected and extracted with PCI method after equal amount of spike-in RNA (GeneChipTM Poly-A RNA Control Kit, Applied Biosystems, Catalog #900433) was added to each fraction.... The expression of spike-in RNA, Daptomycin (DAP) was used for normalization.”

About Supp Figure 1 and the metabolic labeling: It seems that only one experiment has been performed. Also, the manuscript considers that its results parallel that of the

profiles. I don't think it does. In any case, it is meaningless without, at least, two additional biological replicates with their respective quantifications.

The AHA labeling experiments were repeated more than 3 times with similar results. We have replaced the previous figures with higher-quality visuals that more accurately depict results closest to the average value as shown in Supplementary Fig. 1 in the revised version.

Figure 2 and L143: How were the transcriptomic experiments performed?? Please, clarify, as in the text they appear as “RNA-sequencing” but from the legend, they are microarray experiments.

We thank the reviewer for pointing this out. We have corrected the mistake in the legend of Fig. 2. The transcriptomic and translational analyses were performed by RNA-sequencing using total RNA and polysomal RNA, respectively. The detailed methods for RNA isolation and RNA sequencing analysis are provided in the “Methods” section.

About the experiments performed to check for the interaction between eIF2 and SPA1:
a) Does the TAP-SPA1 construct also carry the Myc tag? If so, it should be stated to understand why in the co-IP experiments, anti-myc is used to detect TAP-SPA1.

Yes, it is cMyc-tagged. The full name of TAP tag is Tandem Affinity Purification tag, which has 9 c-Myc repeats as reported by Saijo et al., 2003 and Paik et al., 2019. We have stated this in the text in line 200 where TAP-SPA1 was first mentioned: “...total extract from tandem affinity purification c-Myc-SPA1 (TAP-SPA1) transgenic seedlings.”

b) Supp Figure 2: The figure is mislabeled. It is an anti-GST pull down, and not an MBP pull down. Also Given that eIF2-GST is interacting with all the bands of MBP-SPA1 that also appear in the input, I believe that a very important control is missing, that is to check that MBP-SPA1 is not binding the anti-GST resin/beads. MBP alone may not bind it, but SPA1 has not been tested.

We thank the reviewer for pointing this out. We have corrected the statement in the legend as in Supplementary Fig. 3 in the revised version. We have also repeated this pull-down experiment using GST only as negative control. As shown in Supplementary Fig. 3, MBP-SPA1 does not bind to GST only, suggesting that SPA1 was pulled down by eIF2 α .

c) Also, are the tagged version of SPA and eIF2 functional? Do they complement mutant phenotypes?

Yes, we believe that the tagged version of SPA1 and eIF2 α are functional. As reported

by Saijo et al. in 2003, expressing TAP-SPA1 in *spa1-3* mutant can rescue the phenotype in the mutant. We are using the same line as described in Saijo et al. in 2003. Our GFP-tagged eIF2 α transgenic lines also showed significant changes in translation efficiencies and different light responses suggesting that this C-term-GFP-tagged eIF2 α still function to affect these phenotypes (Fig. 8).

d) Figure 3C and Supp fig 3: Please, describe the eIF2 construct a bit. Under which promoter is it expressed? And also the position of the tag, as in the same figure (Supp 3), on one side appears to be in the N-ter and on the other side, in the C-ter.

The eIF2 α construct built in this study is a GFP-tagged protein fused to the C-terminus of eIF2 α , driven by *Cauliflower mosaic virus* 35S promoter. We have provided the detailed information for “vector construction and transgenic lines generation” in the “Method” section. We have also unified the name of eIF2 α construct as eIF2 α -GFP throughout the revised manuscript.

e) I have a problem with the co-IPs done using Agrobrest. Maybe I am biased as I don't know anyone for whom this system works. Given that, in my opinion, the pull down experiment is lacking an important control, and that trusting the interaction between SPA1 and eIF2 is key to trust the entire manuscript, I would request other more “standard” method to prove the interaction. Ideally, co-IP using SPA1 and *eif2alpha* mutants expressing the tagged versions that complement the phenotype.

We have performed a new set of co-IP experiments using TAP-SPA1 transgenic lines and eIF2 α -GFP transgenic lines as shown in Fig. 3b. We have also performed a new set of co-IP experiments using the cytosolic fraction of TAP-SPA1 to perform IP and use anti-eIF2 α antibody to detect their interaction as shown in Fig. 3c. We have also performed BiFC to provide additional evidence that SPA1 can interact with eIF2 α in the cytosol as shown in Fig. 3d.

Figure 4: Please, indicate the protein sizes in the panels.

We have added and indicate the protein sizes in the panel in Fig. 4.

4a: please, indicate the amounts of proteins used, both for SPA1 and also which are the increasing amounts of eIF2 α .

The amount of both SPA1 and eIF2 α substrates added in our kinase assay was 500 ng as stated in “*In vitro* kinase assay” method section. We have indicated the increasing amounts of eIF2 α in the figures using “+, ++, and +++” symbols in Fig. 4a and b.

Please, describe in the method section the part corresponding to this experiment. In the

main text, SPA1 is tagged with strep and eIF2 is tagged with GST. In methods, eIF2 is tagged with both strep and GST...

As stated in our "Method" section in line 726, eIF2 α was first subcloned into pASK75 for the strep tag and then cloned into pGEX-4T-1 for the GST tag. In line 755, SPA1 was fused with strep tag. We have added this information and modified the text in line 263: "...eIF2 α homologs with GST tagged to their N-terminus and strep tagged to their C-terminus (GST-eIF2 α .1/GST-eIF2 α .2) from bacteria (*E. coli*) to perform *in vitro* kinase assay."

4b and c: please, discuss why phosphorylation on the C-ter does not explain the amount of phosphorylation of the full-length. Also, how many biological replicates have been performed of these experiments? Why use for eIF2 α .2 experiment in which the different versions of eIF2 α are barely expressed?

To avoid over exposure of the fragment proteins, we have lowered the amount added for the fragments. According to the trace amount of C-terminus added in the kinase assay (Fig. 4c), the phosphorylation signal is actually very strong that is comparable to the full-length if normalized. The protein amount added for eIF2 α .2 can be observed in the western blot analysis shown in Supplementary Fig. 8.

In addition, instead of the Coomassie gels, I would like to see WB to be able to quantify phosphorylation relative to the amount of protein produced.

We thank the reviewer for the suggestion. We have provided a western blot result in Supplementary Fig. 8 showing the amount for N-, M-, and C-eIF2 α fragments added in our kinase assay in Fig. 4c.

Figure 5B: how was this "semi *in vivo* kinase" experiment done? In the legend, it says that TAP-SPA1 protein was extracted. But in the methods section, L652, it says that for figure 5b, eIF2 α proteins were incubated with seedling extracts. If this is so, this is no proof that SPA1 is phosphorylating eIF2 α . This shows that the phospho-null versions of the C-ter of eIF2 do not get phosphorylated, and that the WT version gets phosphorylated in the light, but does not show a role of SPA1 on it.

We have deleted the statement in the legend. The semi-*in vivo* kinase assay was performed using seedling extracts. To show that SPAs are necessary for the phosphorylation of eIF2 α , we have repeated this experiment and included *spaQ* seedling sample. As shown in Fig. 5c, in *spaQ* mutants, eIF2 α was less phosphorylated under both dark and light conditions.

Figure 5C: I do not understand the figure... why is the amount of eIF2 α so low

spaQ in L compared to the other lanes when the Rpt5 signal is stronger?

We have repeated this phos-tag mobility shift experiment and provided a more significant and higher-quality result as shown in Fig. 5d. The shifted band of phosphorylated eIF2 α was detected in Col-0 but not in *spaQ*. The loading was more equal compared to the previous result.

Is that super faint phosphorylation signal in Col-0 in L significant?

As mentioned above, we have repeated this phos-tag mobility shift experiment and provided a more significant and higher-quality result as shown in Fig. 5d. The shifted band of phosphorylated eIF2 α was detected in Col-0 but not in *spaQ*. Together with our new semi-*in vivo* kinase assay using P32 detecting phosphorylated C-eIF2 α in both Col-0 and *spaQ* background and *in vivo* kinase assay detecting native eIF2 α in both Col-0 and *spaQ* backgrounds, we hope these results are more convincing.

L281: the “significant” reduction?

We have deleted the word “significant”. We have also repeated the phos-tag mobility shift experiment and provided a more significant result.

L285: please, explain the mSPA1 construct that has a mutation in the kinase domain. What is this mutation? Does it lack the entire domain? Is it a nucleotide substitution? The LUC-mSPA1/*spaQ* was described previously (Paik et al, 2019; Wang et al., 2021) as stated in our “method” section in line 599. A mutant form of SPA1 (mSPA1) that has the R517E mutation in the Ser/Thr kinase domain was cloned into pCAMBIA C-Luc vector. This mutation was also described when first mentioned in line 280: “We also used a mutated form of SPA1 (mSPA1) that has the R517E mutation in Ser/Thr kinase domain in the kinase assay and no phosphorylation activity was observed (Fig. 4b; Supplementary Fig. 10).”

And as in Figure 1, no conclusions can be made from these profiles.

We have repeated these experiments and provided better profiles with similar results.

I don't think the data allows to think that eIF2 α 2 is “specifically” involved in embryogenesis... it can play other roles plant-wide ...

We have toned down this conclusion as stated in line 404: “Both results suggest that eIF2 α .2 might be involved in embryogenic development and seems to play a more important role in that process compared with eIF2 α .1.”

The WT, Phospho-null, and phospho-mimetic versions of eIF2 alpha should be

transformed in the eif2 alpha mutants (transforming the hets) to see if they complement and so, understand the biological significance of the phosphorylation in the C-ter.

We thank the reviewer for the suggestion. We have actually tried transforming the 3 variants into eIF2 α heterozygous mutants many times but failed. We are in the middle of crossing the 3 transgenic lines with the eIF2 α heterozygous mutants to obtain the complementation lines. We are also in the middle of generating the eIF2 α variants driven by its own promoter and transform them into eIF2 α mutants to obtain the complementation lines. However, it might take more than 6 months for us to obtain the F2 generation to be able to screen and analyze the genotypes. We hope we can report the results in our future studies.

In Figure 7A, if all transformants are in Col-0 background, why are the ones expressing the 2-8A version not responding normally to light.

We thank the reviewer for the comment. Our suspicion is that the expression of transgene has a dominant effect over the native eIF2 α . It might be because that the protein levels of transgenic eIF2 α variants are much higher than the native eIF2 α and the native eIF2 α weren't sufficient enough to be phosphorylated by SPAs upon light exposure. We have performed a western blotting showing the native eIF2 α amount in Col-0 and transgenic lines as shown in Supplementary Fig. 14. Compared to Col-0, the levels of native eIF2 α were reduced possibly due to the compensation of transgenic proteins. Once the ternary complex is formed together with 8A, it could cause dominant negative function of native eIF2 α . The smaller portion of native eIF2 α might result in the phenotypes of these transgenics.

Figure 8A. Col-0 is not etiolated. It does not show a hook and the cotyledons are opening.

We thank the reviewer for pointing this out. We might have accidentally stretched the hook and moved the cotyledon while taking pictures. We have repeated this experiment and carefully observed the photomorphogenic phenotypes. A better-quality image is provided as shown in Fig. 8A.

Also, please, discuss why a Col-0 background would not be responding normally to light when expressing ectopically the eIF2alpha mutant versions. It can be because the mutant versions sequester the TC, but if so, eIF2 alpha het mutants should also show a similar phenotype?

As mentioned above, we think that the expression of transgene has a dominant effect over the native eIF2 α . It might be because that the protein levels of transgenic eIF2 α variants are much higher than the native eIF2 α and might sequester and dominate the

ternary complex formation. We have performed a western blotting showing the native eIF2 α amount in Col-0 and transgenic lines as shown in Supplementary Fig. 14. Compared to Col-0, the levels of native eIF2 α were reduced possibly due to the compensation of transgenic proteins. Once the ternary complex is formed together with 8A, it could cause the dominant negative function of native eIF2 α . The smaller portion of native eIF2 α might result in the phenotypes of these transgenics. Also, the eIF2 α protein inside eIF2 α heterozygous mutant are still wild-type form of eIF2 α that can be phosphorylated and dephosphorylated by kinases and phosphatases. It might not show similar phenotype as the phosphor-null and phosphor-mimicking transgenic plants.

Fig 9C and D: Please, draw the panels as in A and B, because it seems that GST was pulled-down and that eIF2 β interacted with GST alone with equal intensity.

We have corrected this mistake as shown in Fig. 9c and d.

Methods:

Please, revise all methods as important information is missing from them and it would not be possible to replicate some experiments using the information provided.

We have added new sections including “Vectors construction and generation of transgenic plants”, “Fractionation of nuclear and cytosolic proteins”, “Microscopy of GFP-conjugated protein and BiFC”, “Mobility shift assay”, as well as adding more detailed information in the existing methods.

In vitro pull-down assays: the methods refer to eIF5A and eIF4E that do not appear in the manuscript. Please, rewrite the section using the actual constructs employed in this manuscript.

We have corrected and re-written this method section which should be the assays for eIF2 β and eIF2 γ interactions.

Please, describe the Agrobacterium method employed here better. It is just a copy of the original paper which also doesn't describe the method very clearly. Which is the “desire cocultivation liquid media” employed in these experiments? How do you cultivate Arabidopsis seedlings in 1 ml of Agrobacterium? Which are the conditions in which you incubate Arabidopsis “in the same growth room for 2 -3 days”?

We have deleted this method as the experiment was replaced with new sets of co-IP experiments without this technique.

The manuscript uses “ribosome-RNA profiling” as the technique to separate polysomes

along a sucrose gradient based on the number of ribosomes when its actual name is “polysome profiling”.

L121: “ribosome profiling” is a synonym of “Ribo-seq”, which is not what has been done. The manuscript should refer here to the polysome profiles.

We thank the reviewer for pointing this out. We have corrected this word to “polysome profile” throughout the manuscript.

Figure 3B: the Y2H panel is mislabeled. SD-Leu should be SD-Leu-Trp- and the SD-Leu-Trp- should be SD-Leu-Trp-His.

We have corrected this mistake and added the missing labels in Supplementary Fig. 2 in this new version of manuscript.

L46-47: I would not dare to say that “there is a substantial knowledge of the identities and molecular signatures associated with mRNAs regulated at the translational level”. Knowledge in this field is far from substantial. The great majority of the molecular mechanisms governing translation regulation are still unknown.

We thank the reviewer for the comment. We have modified this statement in line 46-48 in the revised manuscript: “Although there are a few studies reporting the identities and molecular signatures associated with mRNAs regulated at the translational level, little is known about the mechanisms of light-triggered regulation of cytoplasmic translation in plants.”

Other issues:

SPA appears as SAP in many places in the manuscript.

We have corrected these mistakes in the revised manuscript.

SUPPRESSOR OF PHYA-105 (SPA1-4) also appears as SUPPRESSOR OF PHYA-105 1 (SPA1-4).

We have corrected this mistake in the abstract.

L63: I’d put and “However, ” before “Because”. And they don’t only have plant-specific translation control mechanisms. They also have a plant-specific translational machinery with plant-specific translation factors.

We have modified this statement as in line 63: “...however, plants have specific translational machinery with specific translation factors because of their unique biological activities, such as photosynthesis and the capacity to respond to stresses.”

L137: what does “(DAP)” stands for?

DAP stands for Daptomycin gene from *Bacillus subtilis*, a spike-in RNA added for normalization. We have added the full names for DAP in the legend of Fig. 1.

Reviewer #2 (Remarks to the Author):

#1: The author claimed that the SPA regulated protein translation via phosphorylating eIF2alpha C-terminus. However, no genetic evidence (epistasis analysis) supports this claim. Instead, they provided separate data by showing that *spaQ* mutant regulates protein translation (Figure 1 and 2), SPA1 could phosphorylate eIF2alpha (Figure 4 and 5), and eIF2alpha phosphorylation regulate protein translation (Figure 7) without closing the regulatory loop.

Ectopic expression of eIF2alpha phospho-variants in *spaQ* mutant should be tested to close the loop by checking if eIF2alpha-8D or 8A could restore *spaQ*'s translation, enhancing translation in darkness while inhibiting translation in light. Importantly, the transgene *spaQ*/eIF2alpha will also benefit the eIF2 complex assembly assay in Figure 9 and the phos-tag gel assay in Figure 5, which will be stated below.

We thank the reviewer for the comment and suggestion. Actually, we have tried to ectopically express the phosphor-variant of eIF2 α , eIF2 α -8A and -8D, in *spaQ* background by transforming the variants into *spaQ* by floral dipping and crossing. However, we have failed many times to generate the eIF2 α variants in *spaQ* mutants. It might be because that over-expression of the variant in *spaQ* is somehow toxic. We are currently trying to generate the transgenics driven by eIF2 α 's native promoter. Nevertheless, to show that SPAs are necessary for the phosphorylation of eIF2 α , we have repeated this phos-tag mobility shift experiment and provided a more significant and higher-quality result as shown in Fig. 5d. The shifted band of phosphorylated eIF2 α was detected in Col-0 but not in *spaQ*. The loading was more equal compared to the previous result. We have also repeated the semi-*in vivo* kinase assay and included *spaQ* seedling sample. As shown in Fig. 5c, in *spaQ* mutants, eIF2 α was less phosphorylated under both dark and light conditions. To manifest that SPAs are necessary for the ternary complex formation, we have also performed semi-*in vivo* pull-down assay and showed that eIF2 α could not interact with eIF2 β in *spaQ* mutants (Supplementary Fig. 16).

In the proposed model (Figure 10), the SPA1 represses protein translation through eIF2 complex assembly in dark conditions. However, the translational outputs from *spaQ* mutant (Figure 1) and eIF2 α .2-8A transgene (Figure 7) under dark conditions are opposite. Furthermore, no protein interaction was detected between SPA1 and eIF2alpha in the dark, as shown in Figure 3c. These data raised concerns that SPA-

regulated translation is not through eIF2alpha phosphorylation, at least in darkness. As the author wrote in the introduction, SPA kinases repress translation in the dark might through TOR/RPS6 pathway, like COP1. It will greatly benefit the proposed model to test the *spaQ*/eIF2alpha phospho-variant transgenes (*spaQ*/eIF2alpha-WT, *spaQ*/eIF2alpha-8A, and *spaQ*/eIF2alpha-8D) to check the epistasis.

We thank the reviewer for the comment and suggestion. As mentioned above, we have failed many times generating the transgenic lines in *spaQ* background and we are in the middle of generating the ones driven by eIF2 α 's native promoter. For now, we can only hypothesize why this would happen. In fact, we do not think that SPA1 represses translation through eIF2 complex assembly in dark conditions. Under dark condition, SPA1 does not affect translation through eIF2 complex assembly simply because they don't interact with each other. However, SPA1 might repress translation through TOR-RPS6 pathway under dark condition by working together with COP1. Our hypothesis is that SPAs regulate both TOR-RPS6 and eIF2 α phosphorylation pathways but independently, and SPAs play negative roles under dark but play positive roles for TOR-RPS6 regulation in the light condition. In the dark, SPAs work together with COP1 to inhibit translation through TOR-RPS6 pathway. Thus, in *spaQ* mutant, the translation activity is high. Upon light exposure, beside the effect of light-induced eIF2 α phosphorylation by SPAs, it is likely that SPAs are also necessary to promote TOR-RPS6 pathway through phosphorylating PIFs and other transcription factors (Paik et al., 2019), thus the translation decreased dramatically in the light. We will try to elucidate this regulatory mechanism and discuss in more detail in our future studies.

#2: The authors need to present more convincing data for the phosphorylation of eIF2alpha dependency on SPA kinases *in vivo*.

In Figure 5b, the semi *in vivo* phosphorylation of eIF2alpha should include *spaQ* mutant because this figure did not provide SPA1 dependency in eIF2alpha phosphorylation, and the plant extracts contain massive other kinases, which may lead to false positive results.

In Figure 5c, because the native antibodies (Please provide cat no.) can detect more than one band in the hypo-phosphorylated conditions (Dark), making it not a perfect match for the assay. Moreover, the third line, which has the slightly increased p-eIF2alpha, has more protein loading than the rest. I suggest generating the *spaQ*/eIF2alpha-GFP transgenes, which were proven to show a unique band in WT, to repeat the phos-tag assay. According to the multiple phosphor-sites predicted in eIF2alpha, I expect strong shifted phosphorylation bands in light and weaker ones in *spaQ* mutant.

We thank the reviewer for the great comment and suggestion. We have actually tried

expressing eIF2 α -GFP in Col-0 and *spaQ* background by AGROBEST technique and perform the phos-tag assay. However, the protein size of eIF2 α -GFP was too large to distinguish the shifted band. We have repeated this phos-tag mobility shift experiment and provided a more significant and higher-quality result as shown in Fig. 5d. The shifted band of phosphorylated eIF2 α was detected in Col-0 but not in *spaQ*. The loading was more equal compared to the previous result. To show that SPAs are necessary for the phosphorylation of eIF2 α , we have repeated the semi-*in vivo* kinase assay and included *spaQ* seedling sample. As shown in Fig. 5c, in *spaQ* mutants, eIF2 α was less phosphorylated under both dark and light conditions. Together with our new semi-*in vivo* kinase assay using ^{32}P detecting phosphorylated C-eIF2 α in both Col-0 and *spaQ* background and *in vivo* kinase assay detecting native eIF2 α in both Col-0 and *spaQ* backgrounds, we hope these results are more convincing.

#3: The claim that SPA kinases regulate translation via affecting eIF2 complex formation could greatly benefit from additional experiments to check eIF2 assembly in *spaQ* mutant background. For example, check the eIF2 α .2 and eIF2 β interaction in *spaQ*/eIF2 α .2-GFP transgenes. The availability of the eIF2 β native antibody from Dr. Browning's lab, one of the authors in this work, makes the experiment easy to perform.

We thank the reviewer for the suggestion. Due to lack of eIF2 α -GFP/*spaQ* transgenic plants, we have performed a semi-*in vivo* pull-down experiment using GST-eIF2 α to pull down native eIF2 β in Col-0 and *spaQ* as shown in Supplementary Fig. 16. The results showed that almost no interaction between alpha and eIF2 β can be detected in both Col-0 and *spaQ*, whereas eIF2 α .2 interacts with eIF2 β in Col-0 after light treatment but no interaction was detected in *spaQ* background.

#4: Because one of the main focuses in the manuscript is to elucidate how SPA kinases regulate translation efficiency (TE), I suggest calculating TE (TE = mRNApl fold change/mRNAss fold change) instead of showing both the transcriptional and translational changes to simplify the data interpretation in Figure 2, and ranking the GO analysis base on the TE values. The current data in Figure 2 is distracting because we must focus on mRNAss and mRNApl changes.

We thank the reviewer for the suggestion. We have analyzed the TE and added this figure in Fig. 2b in the revised manuscript. As shown in Fig. 2b, most DEG showed increased TE (redder) under dark condition but decreased TE (greener) under light condition, suggesting that SPAs inhibit the translation in dark, but enhance the translation of these genes under light, which is more consistent with our polysomal profile data. We have also kept the classification analysis showing the GO terms

regulated at different clusters. We think this is more informative to support our hypothesis that the translational regulation light-responsive and development-related genes results in photomorphogenic phenotypes as well as other phenotypes.

#5: Typically, the ratio of polysome (PL)/monosome (NP) in ribosomal profiling is dynamically changing, with more polysome companies with less monosome or the other way around, like the paper (38) cited in this manuscript. However, this dynamic is hard to observe in the profiling figures (Figures 1 and 6), instead showing a similar trend in the manuscript. Could the authors comment on this?

We thank the reviewer for pointing this out. We have replaced the previous profiles with better profile results that were performed previously but in a different batch. We have indicated the peaks of 40S, 60S and 80S in the first profile in Fig. 1. The small peak next to 80S on the right represents disome, so our polysomal fraction contains the part with ≥ 3 ribosomes. We actually do see some dynamic changes in some profiles, especially the size changes in 80S peaks. However, the monosomal fraction sometimes looks a lot bigger because of the increase in disome after light treatment. Our profiles might be different from the paper cited because of the gradient we use (12.5%-60%) might cause different resolution observed and the sensitivity set in different labs. It also might be because that the ribosome biosynthesis is affected in *spaQ* and *cop1-4* backgrounds.

#6: The AHA labeling assay in Figure S1, *spaQ* mutant showing more newly translated proteins than Col-0 under light conditions, indicating SPAs negatively regulate protein translation in light conditions. Does it contradict the paper's claim that SPAs positively control protein translation under light conditions?

The AHA labeling experiments were repeated more than 3 times with similar results. We have replaced the previous figures with higher-quality visuals that more accurately depict results closest to the average value as shown in Supplementary Fig. 1 in the revised version. The average value of the 3 biological repeats is provided in the table in Supplementary Fig. 1.

#7: Protein translation occurs in the cytoplasm, and SPA1 kinase was reported localized in the nucleus. How does SPA1 regulate protein translation? Do SPA kinases have cytosol localization in light conditions? Please check the interaction between SPA1 and eIF2alpha *in vivo* using BiFC assay.

We thank the reviewer for the great suggestion. To first observe the light-dependent localization of SPA1, we generated the construct of GFP fused to the C-terminus of SPA1 driven by CaMV 35S promoter and transformed it into Col-0 protoplasts. As

shown in Supplementary Fig. 5, SPA1-GFP localized in the nucleus in 91.3% of the transformed protoplasts incubated under dark for 2 h before observation. However, under light condition, 37.5% the transformed protoplasts have SPA1-GFP forming speckles in the cytosol, suggesting that light does induce some translocation of SPA1 into cytoplasm. To further confirm the interaction between SPA1 and eIF2 α in the cytosol, we isolated cytosolic fraction of TAP-SPA1 from transgenic seedlings and incubated with anti-cMyc to perform co-IP (Fig. 3c). The results showed that the cytosolic TAP-SPA1 has higher interaction intensity with native eIF2 α after light treatment. We also performed BiFC to further validate the interaction between eIF2 α homologs and SPA1 in planta (Fig. 3d). The BiFC results showed that both homologs of eIF2 α can interact with SPA1 in the cytosol under light condition. Collectively, these results suggest that the interactions between SPA1 and eIF2 α homologs are light-dependent and localized in the cytosol. These results indicate that SPA1 might interact with eIF2 α under light condition, and when overexpressing SPA1 might cause them to aggregate when there are not enough eIF2 α to interact with them.

#8: eIF2 α -8A transgene repressed protein translation in both dark and light conditions (Figure 7); however, the transgene displayed a more extended hypocotyl elongation. Do the authors comment on this counterintuitive phenotype and how the phenotype is linked to translation?

In our hypothesis, we think that the higher translation efficiency results in photomorphogenic phenotype, whereas lower translation efficiency results in skotomorphogenic phenotype. It could be the low translation efficiency in eIF2 α -8A transgenic plants triggers a more severe skotomorphogenesis, leading to longer, extended hypocotyl length, a more closed cotyledon and less chlorophyll synthesis. The transcriptome and translome analyses in *spaQ* could reflect the light-responsive genes regulated at the translational level both under dark and under light condition. We will further investigate the genes regulated at the translational level in mutative eIF2 α transgenic plants. We hope to provide more evidence in our future studies.

#9: Please indicate the SPA1 protein position in Figure 4.

We have added the labels and indicate the protein sizes in the panel in Fig. 4.

#10: In Figure S2, is it GST or MBP pull-down? Please confirm.

We have corrected this mistake. It is GST pull-down assay as shown in Supplementary Fig. 3 in the revised manuscript.

Reviewer #3 (Remarks to the Author):

1. Major points:

a. The C-term-phosphorylated eIF2 α (8D) promotes translation efficiency and photomorphogenesis in the dark, whereas the C-term-unphosphorylated eIF2 α (8A) results in decreased translation efficiency. *spaQ* is defective in the C-term-phosphorylation of eIF2 α and thus in its translation efficiency, but develops enhanced photomorphogenesis with short hypocotyls like 8D under light conditions. The photomorphogenic phenotypes of 8A and *spaQ* seem contradictory at the first sight. One possible explanation is that strong inhibition of the degradation of photomorphogenesis-promoting proteins in *spaQ* is a major cause of constitutive photomorphogenesis since SPAs are crucial components of COP1 E3 ligase complex. Therefore, it is not clear here concerning the contribution of SPA1-mediated eIF2 α phosphorylation in SPA regulation of photomorphogenesis. The reviewer suggests the authors provide necessary genetic data to specifically evaluate the function of SPA1-mediated eIF2 α phosphorylation (apart from COP1-SPAs E3 activity). For example, compare the phenotypes of Col and LUC-mSPA1/*spaQ* (not shown here) and discuss. Will expressing 8D rescue LUC-mSPA1/*spaQ*?

We thank the reviewer for the comment and suggestion. It is very likely that the photomorphogenesis regulated by SPA1-mediated eIF2 α phosphorylation (as the phenotypes observed in eIF2 α -8A and 8D) and the SPA1-mediated photomorphogenic phenotypes as part of the E3 ligase complex with COP1 are very distinct and resulted from many aspects. The eIF2 α -8D overexpression results in a shorter hypocotyl length, but it is still far from the extremely short hypocotyl in *spaQ* mutant. The photomorphogenic phenotype of *spaQ* comes from the strong inhibition of the degradation of photomorphogenesis-promoting proteins in *spaQ* because of the E3 ligase activity of SPAs. This photomorphogenic phenotype did not reverse even though SPA1 actually promotes photomorphogenesis under light condition. It has been reported that SPA1, as a kinase, plays a major role in regulating photomorphogenesis through phosphorylating PIFs and HY5 under light condition (Paik et al., 2019; Lee et al., 2020; Wang et al., 2021). Thus, one cannot expect the similar effect in photomorphogenic phenotypes caused by ectopic expression of eIF2 α -8A and mutation in SPAs (*spaQ* or mSPA1). The LUC-mSPA1/*spaQ* phenotype was previously reported in Paik et al., 2019 and Wang et al., 2021. It showed similar phenotype as *spaQ* mutant, as mutation in SPA1 kinase domain was very critical for both its roles in E3 ligase component as well as in kinase activity, which result in the overall phenotype. We believe that expressing eIF2 α -8D might partially rescue the constitutive photomorphogenic phenotypes in LUC-mSPA1/*spaQ* and *spaQ*. However, we have failed many times to generate the eIF2 α variants in *spaQ* mutants. It might be because

that over-expression of the variant in *spaQ* is somehow toxic. We are currently trying to generate the transgenic lines driven by eIF2 α 's native promoter. Nevertheless, to show that SPAs are necessary for the phosphorylation of eIF2 α , we have repeated this phos-tag mobility shift experiment and provided a more significant and higher-quality result as shown in Fig. 5d. The shifted band of phosphorylated eIF2 α was detected in Col-0 but not in *spaQ*. The loading was more equal compared to the previous result. We have also repeated the semi-*in vivo* kinase assay and included *spaQ* seedling sample. As shown in Fig. 5c, in *spaQ* mutants, eIF2 α was less phosphorylated under both dark and light conditions. To manifest that SPAs are necessary for the ternary complex formation, we have also performed semi-*in vivo* pull-down assay and showed that eIF2 α could not interact with eIF2 β in *spaQ* mutants (Supplementary Fig. 16). We hope these additional experiments support our hypothesis and satisfy reviewer's concern.

b. *spaQ* exhibited enhanced translation in the dark but repressed translation upon light exposure, suggesting that SPAs play opposite roles in translation regulation under dark and light conditions. How does light signal trigger the SPA1-eIF2 α interaction and eIF2 α phosphorylation to promote translation in plants? Based on *in vitro* systems, the SPA1-eIF2 α interaction and eIF2 α phosphorylation is independent on light conditions. However, these two events were detected specifically under light conditions in planta. In addition, from the model figure, the *in vivo* association between SPA1 and eIF2 α appears to be regulated at a subcellular level by light, and the assembly of COP1-SPA complex is also affected? The reviewer suggests the authors provide necessary analyses to clarify these points which are important for understanding the molecular basis of SPA1-mediated translation regulation.

We thank the reviewer for the comment. We think that this might be related to the subcellular level of SPA1 regulated by light. To first observe the light-dependent localization of SPA1, we generated the construct of GFP fused to the C-terminus of SPA1 driven by CaMV 35S promoter and transformed it into Col-0 protoplasts. As shown in Supplementary Fig. 5, SPA1-GFP localized in the nucleus in 91.3% of the transformed protoplasts incubated under dark for 2 h before observation. However, under light condition, 37.5% the transformed protoplasts have SPA1-GFP forming speckles in the cytosol, suggesting that light induces either retention or translocation of some SPA1 into cytoplasm from nucleus. To further confirm the interaction between SPA1 and eIF2 α in the cytosol, we isolated cytosolic protein of TAP-SPA1 transgenic seedlings and incubated with anti-cMyc to perform co-IP (Fig. 3c). The results showed that the cytosolic TAP-SPA1 has higher interaction intensity with native eIF2 α after light treatment. We also performed BiFC to further validate the interaction between eIF2 α homologs and SPA1 in planta (Fig. 3d). The BiFC results showed that both

homologs of eIF2 α can interact with SPA1 in the cytosol under light condition. Collectively, these results suggest that the interactions between SPA1 and eIF2 α homologs are light-dependent and localized in the cytosol.

c. Another concern of the reviewer associates with the functional redundancy and possible differentiated roles of SPA1-4 in eIF2 α phosphorylation and translation regulation. Do SPA1-4 play conserved roles in eIF2 α interaction and eIF2 α phosphorylation? Considering the functional redundancy of SPA1-4 and the predominant role of SPA1 in repressing photomorphogenesis, *spa1* single mutant is potentially a good material to test translation defects and eIF2 α phosphorylation defects. It is a little bit difficult to understand why these two experiments were performed only using *spaQ* but not including *spa1* single mutant.

We thank the reviewer for the comment. Although SPA1 plays a predominant role in repressing photomorphogenesis, it is possible that all four SPAs play roles in the translational regulation. We think that the global translational changes observed in *spaQ* result in its tiny and severe phenotype that is not observed in *spa1* single mutant and lower order mutants, reflected by the changes in light-responsive, growth-related genes revealed in our transcriptome analyses in *spaQ*, suggesting the possible functional redundancy in the translation defects and eIF2 α phosphorylation defects. We will keep investigating the roles of SPA2-4 in this regulation in our future studies.

2. Minor points:

a. Line 158-159, Cluster 4, representing the largest group of mRNAs (4,006 genes), showed decreased translation but negligible changes in mRNA_{SS} level in *spaQ* under the light condition. Why the decreased translation in *spaQ* mRNA_{SS} was regarded as negligible? The changes are much clearer than that in the dark.

We thank the reviewer for pointing this out. We have modified this statement in line 163: “Cluster 4, representing the largest group of mRNAs (4,006 genes), showed a more decrease in translation but smaller changes in mRNA_{SS} level in *spaQ* under the light condition compared to the levels in dark condition.” Also, to manifest the changes in translation compared to the transcriptional level, we have analyzed the TE by normalizing the fold change in mRNA_{PL} with the fold change in mRNA_{SS} (TE = mRNA_{PL} fold change/mRNA_{SS} fold change) and performed clustering analysis as shown in Fig. 2b in the revised manuscript. As shown in Fig. 2b, most DEG showed increased TE (redder) under dark condition but decreased TE (greener) under light condition, suggesting that SPAs inhibit the translation in the dark but enhance the translation of these genes under light, which is more consistent with our polysomal profile data.

b. Fig.4, SPA1 band was very vague in Fig.4b, and its migration pattern seemed different from that in Fig.4a. Because there were many unspecific or degraded fragments, it is very difficult to tell the full-length and truncated eIF2 α .1 and eIF2 α .2 proteins. The authors should indicate these proteins on Coomassie blue staining gels. mSPA1 was not included as a negative control in Fig.4c.

We have added the labels for the proteins and indicate the protein sizes in the panel in Fig. 4. We have also repeated the kinase assay for mSPA1 as shown in Supplementary Fig. 10 to support the SPA1-mediated phosphorylation of eIF2 α .2 in Fig. 4c. The SPA1 migration in Fig. 4a and b was resulted from the different markers used in the experiment. We have added the size in the panel in Fig. 4.

c. Line 254-255, T295 should be S295. Please check throughout the manuscript.

We have corrected these errors throughout the manuscript.

d. Line 314, “+, inactivated boiled CIP” should be “-, inactivated boiled CIP”?

We have repeated this phos-tag mobility shift experiment and provided a more significant and higher-quality result without CIP treatment as shown in Fig. 5d. The shifted band of phosphorylated eIF2 α was detected in Col-0 but not in *spaQ*. The loading was more equal compared to the previous result.

The quantification of Fig.9c and Fig.9d was exactly the same (1.00, 0.80, 1.91). Please check and make the data convincing.

We have corrected this mistake as shown in Fig. 9c and d.

References

1. Paik, I. et al. A phyB-PIF1-SPA1 kinase regulatory complex promotes photomorphogenesis in Arabidopsis. *Nat. Commun.* **10**, 4216. (2019).
2. Pham, V. N., Kathare, P. K. & Huq, E. Phytochromes and phytochrome interacting factors. *Plant Physiol.* **176**, 1025-1038. (2018).
3. Xu, X. et al. Illuminating progress in phytochrome-mediated light signaling pathways. *Trends Plant Sci.* **20**, 641-650. (2015).
4. Saijo, Y. et al. The COP1–SPA1 interaction defines a critical step in phytochrome A-mediated regulation of HY5 activity. *Genes Dev.* **17**, 2642-2647. (2003).
5. Wang, W. et al. Direct phosphorylation of HY5 by SPA kinases to regulate photomorphogenesis in Arabidopsis. *New Phytol.* **230**, 2311-2326. (2021).

6. Lee, S., Paik, I. & Huq, E. SPAs promote thermomorphogenesis by regulating the phyB-PIF4 module in Arabidopsis. *Development* **147**, dev189233. (2020).

REVIEWER COMMENTS

Reviewer #2 (Remarks to the Author):

#1: The author claimed that the SPA regulated protein translation via phosphorylating eIF2alpha C-terminus. However, no genetic evidence (epistasis analysis) supports this claim. Instead, they provided separate data by showing that spaQ mutant regulates protein translation (Figure 1 and 2), SPA1 could phosphorylate eIF2alpha (Figure 4 and 5), and eIF2alpha phosphorylation regulate protein translation (Figure 7) without closing the regulatory loop.

Ectopic expression of eIF2alpha phospho-variants in spaQ mutant should be tested to close the loop by checking if eIF2alpha-8D or 8A could restore spaQ's translation, enhancing translation in darkness while inhibiting translation in light. Importantly, the transgene spaQ/eIF2alpha will also benefit the eIF2 complex assembly assay in Figure 9 and the phos-tag gel assay in Figure 5, which will be stated below.

We thank the reviewer for the comment and suggestion. Actually, we have tried to ectopically express the phosphor-variant of eIF2 α , eIF2 α -8A and -8D, in spaQ background by transforming the variants into spaQ by floral dipping and crossing. However, we have failed many times to generate the eIF2 α variants in spaQ mutants. It might be because that over-expression of the variant in spaQ is somehow toxic. We are currently trying to generate the transgenics driven by eIF2 α 's native promoter. Nevertheless, to show that SPAs are necessary for the phosphorylation of eIF2 α , we have repeated this phos-tag mobility shift experiment and provided a more significant and higher-quality result as shown in Fig. 5d. The shifted band of phosphorylated eIF2 α was detected in Col-0 but not in spaQ. The loading was more equal compared to the previous result. We have also repeated the semi-in vivo kinase assay and included spaQ seedling sample. As shown in Fig. 5c, in spaQ mutants, eIF2 α was less phosphorylated under both dark and light conditions. To manifest that SPAs are necessary for the ternary complex formation, we have also performed semi-in vivo pull-down assay and showed

that eIF2 α could not interact with eIF2 β in spaQ mutants (Supplementary Fig. 16).

The transgenes expressing eIF2 α variants in spaQ are critical for the study. Without checking the translational output using them, I am not fully convinced that the translation changes in spaQ mutant are due to directly phosphorylate eIF2 α in vivo. Meanwhile, it is difficult for me to imagine the toxic effect of eIF2 α -8D in spaQ mutant because, theoretically, it will recover the spaQ translation to the WT level in light conditions, leading to better plant growth (Fig. 1). Since the toxic effect is likely due to overexpression of the truncated version of eIF2 α (C-terminus), I look forward to the native promoter-driven eIF2 α C-terminus transgenes in the spaQ mutant or would suggest using full-length eIF2 α variants to do the transgenes or at least do some transient assays using spaQ protoplasts or AGROBEST established in the study to support further their discoveries that SPAs regulate protein translation in light is through phosphorylating eIF2 α C-terminus in vivo.

I realized that the authors may need to use p56-eIF2 α phospho-antibodies to check if SPAs could phosphorylate the site S56 in eIF2 α . The assays performed in Figure 4 could not rule out the possibility that SPAs were able to phosphorylate eIF2 α N-terminus (S56) and C-terminus simultaneously because the S56A protein used in Figure 4 ruled out the possibility, which will affect the rationality of the following experiments only using C-terminal eIF2 α to check the photomorphogenesis in Figure 8.

In the proposed model (Figure 10), the SPA1 represses protein translation through eIF2 complex assembly in dark conditions. However, the translational outputs from spaQ mutant (Figure 1) and eIF2 α .2-8A transgene (Figure 7) under dark conditions are opposite. Furthermore, no protein interaction was detected between SPA1 and eIF2 α in the dark, as shown in Figure 3c. These data raised concerns that SPA-regulated translation is not through eIF2 α phosphorylation, at least in darkness. As

the author wrote in the introduction, SPA kinases repress translation in the dark might through TOR/RPS6 pathway, like COP1. It will greatly benefit the proposed model to test the spaQ/eIF2 α phospho-variant transgenes (spaQ/eIF2 α -WT, spaQ/eIF2 α -8A, and spaQ/eIF2 α -8D) to check the epistasis.

We thank the reviewer for the comment and suggestion. As mentioned above, we have failed many times generating the transgenic lines in spaQ background and we are in the middle of generating the ones driven by eIF2 α 's native promoter. For now, we can only hypothesize why this would happen. In fact, we do not think that SPA1 represses translation through eIF2 complex assembly in dark conditions. Under dark condition, SPA1 does not affect translation through eIF2 complex assembly simply because they

don't interact with each other. However, SPA1 might repress translation through TOR-RPS6 pathway under dark condition by working together with COP1. Our hypothesis is that SPAs regulate both TOR-RPS6 and eIF2 α phosphorylation pathways but independently, and SPAs play negative roles under dark but play positive roles for TOR-RPS6 regulation in the light condition. In the dark, SPAs work together with COP1 to inhibit translation through TOR-RPS6 pathway. Thus, in spaQ mutant, the translation activity is high. Upon light exposure, beside the effect of light-induced eIF2 α phosphorylation by SPAs, it is likely that SPAs are also necessary to promote TOR-RPS6 pathway through phosphorylating PIFs and other transcription factors (Paik et al., 2019), thus the translation decreased dramatically in the light. We will try to elucidate this regulatory mechanism and discuss in more detail in our future studies.

I think the authors agree with my points that SPAs do not regulate protein translation via phosphorylation of eIF2 α in darkness. The authors may need to change the model (Fig. 10) using the TOR-RPS6 module instead of the eIF2 complex in dark conditions to avoid misleading.

#2: The authors need to present more convincing data for the phosphorylation of eIF2 α dependency on SPA kinases in vivo.

In Figure 5b, the semi in vivo phosphorylation of eIF2 α should include spaQ mutant because this figure did not provide SPA1 dependency in eIF2 α phosphorylation, and the plant extracts contain massive other kinases, which may lead to false positive results.

In Figure 5c, because the native antibodies (Please provide cat no.) can detect more than one band in the hypo-phosphorylated conditions (Dark), making it not a perfect match for the assay. Moreover, the third line, which has the slightly increased p-eIF2 α , has more protein loading than the rest. I suggest generating the spaQ/eIF2 α -GFP transgenes, which were proven to show a unique band in WT, to repeat the phos-tag assay. According to the multiple phosphor-sites predicted in eIF2 α , I expect strong shifted phosphorylation bands in light and weaker ones in

spaQ mutant.

We thank the reviewer for the great comment and suggestion. We have actually tried expressing eIF2 α -GFP in Col-0 and spaQ background by AGROBEST technique and perform the phos-tag assay. However, the protein size of eIF2 α -GFP was too large to distinguish the shifted band. We have repeated this phos-tag mobility shift experiment and provided a more significant and higher-quality result as shown in Fig. 5d. The shifted band of phosphorylated eIF2 α was detected in Col-0 but not in spaQ. The loading was more equal compared to the previous result. To show that SPAs are necessary for the phosphorylation of eIF2 α , we have repeated the semi-in vivo kinase assay and included spaQ seedling sample. As shown in Fig. 5c, in spaQ mutants, eIF2 α was less phosphorylated under both dark and light conditions. Together with our new semi-in vivo kinase assay using ³²P detecting phosphorylated C-eIF2 α in both Col-0 and spaQ background and in vivo kinase assay detecting native eIF2 α in both Col-0 and spaQ backgrounds, we hope these results are more convincing.

The authors partially addressed my concerns. Because AGROBEST is a transient experiment, I would suggest the authors use HA or FLAG-tagged eIF2 α instead of eIF2 α -GFP to perform the band shift assay using phos-tag gel because the HA and FLAG are tiny tags, which will significantly address the big protein size problem of eIF2 α -GFP. If they insist on performing the phos-tag assay using non-specific (which can detect at least two bands) antibodies, I suggest adding another PPase-treated control in Col-0 samples to demonstrate that the shifted band is not a non-specific band in Col-0 background, which is somehow missing in spaQ mutant background. Anyway, this study needs PPase treatment to demonstrate that the SPAs-mediated band shift in phos-tag gel is indeed due to phosphorylation in vivo.

#3: The claim that SPA kinases regulate translation via affecting eIF2 complex formation could greatly benefit from additional experiments to check eIF2 assembly in spaQ mutant background. For example, check the eIF2 α .2 and eIF2 β interaction in spaQ/eIF2 α .2-GFP transgenes. The availability of the eIF2 β native antibody from Dr. Browning's lab, one of the authors in this work, makes the experiment easy to perform.

We thank the reviewer for the suggestion. Due to lack of eIF2 α -GFP/spaQ transgenic plants, we have performed a semi-in vivo pull-down experiment using GST-eIF2 α to pull down native eIF2 β in Col-0 and spaQ as shown in Supplementary Fig. 16. The

results showed that almost no interaction between alpha and eIF2 β can be detected in both Col-0 and spaQ, whereas eIF2 α .2 interacts with eIF2 β in Col-0 after light treatment but no interaction was detected in spaQ background.

Ecoli-purified GST-eIF2 α may lack the necessary modification, which is essential for its function. I suggest using transient expression assay through Arabidopsis protoplasts or AGROBEST instead of semi-in vitro assay to check the assembly of eIF2 complex by transiently expressing different tagged eIF2 α , β , and even γ in Col-0 and spaQ mutant and perform Co-IP experiments.

#4: Because one of the main focuses in the manuscript is to elucidate how SPA kinases regulate translation efficiency (TE), I suggest calculating TE (TE = mRNApl fold change/mRNAss fold change) instead of showing both the transcriptional and translational changes to simplify the data interpretation in Figure 2, and ranking the GO analysis base on the TE values. The current data in Figure 2 is distracting because we must focus on mRNAss and mRNApl changes.

We thank the reviewer for the suggestion. We have analyzed the TE and added this figure in Fig. 2b in the revised manuscript. As shown in Fig. 2b, most DEG showed increased TE (redder) under dark condition but decreased TE (greener) under light condition, suggesting that SPAs inhibit the translation in dark, but enhance the translation of these genes under light, which is more consistent with our polysomal profile data. We have also kept the classification analysis showing the GO terms regulated at different clusters. We think this is more informative to support our hypothesis that the translational regulation light-responsive and development-related genes results in photomorphogenic phenotypes as well as other phenotypes.

My concerns were addressed.

#5: Typically, the ratio of polysome (PL)/monosome (NP) in ribosomal profiling is dynamically changing, with more polysome companies with less monosome or the other way around, like the paper (38) cited in this manuscript. However, this dynamic

is hard to observe in the profiling figures (Figures 1 and 6), instead showing a similar trend in the manuscript. Could the authors comment on this?

We thank the reviewer for pointing this out. We have replaced the previous profiles with better profile results that were performed previously but in a different batch. We have indicated the peaks of 40S, 60S and 80S in the first profile in Fig. 1. The small peak next to 80S on the right represents disome, so our polysomal fraction contains the part with ≥ 3 ribosomes. We actually do see some dynamic changes in some profiles, especially the size changes in 80S peaks. However, the monosomal fraction sometimes looks a lot bigger because of the increase in disome after light treatment. Our profiles might be different from the paper cited because of the gradient we use (12.5%-60%) might cause different resolution observed and the sensitivity set in different labs. It also might be because that the ribosome biosynthesis is affected in spaQ and cop1-4 backgrounds.

My concerns were addressed.

#6: The AHA labeling assay in Figure S1, spaQ mutant showing more newly translated proteins than Col-0 under light conditions, indicating SPAs negatively regulate protein translation in light conditions. Does it contradict the paper's claim that SPAs positively control protein translation under light conditions?

The AHA labeling experiments were repeated more than 3 times with similar results. We have replaced the previous figures with higher-quality visuals that more accurately depict results closest to the average value as shown in Supplementary Fig. 1 in the revised version. The average value of the 3 biological repeats is provided in the table in Supplementary Fig. 1.

My concerns were addressed.

#7: Protein translation occurs in the cytoplasm, and SPA1 kinase was reported localized

in the nucleus. How does SPA1 regulate protein translation? Do SPA kinases have cytosol localization in light conditions? Please check the interaction between SPA1 and eIF2 α in vivo using BiFC assay.

We thank the reviewer for the great suggestion. To first observe the light-dependent localization of SPA1, we generated the construct of GFP fused to the C-terminus of SPA1 driven by CaMV 35S promoter and transformed it into Col-0 protoplasts. As shown in Supplementary Fig. 5, SPA1-GFP localized in the nucleus in 91.3% of the transformed protoplasts incubated under dark for 2 h before observation. However, under light condition, 37.5% the transformed protoplasts have SPA1-GFP forming speckles in the cytosol, suggesting that light does induce some translocation of SPA1 into cytoplasm. To further confirm the interaction between SPA1 and eIF2 α in the cytosol, we isolated cytosolic fraction of TAP-SPA1 from transgenic seedlings and incubated with anti-cMyc to perform co-IP (Fig. 3c). The results showed that the cytosolic TAP-SPA1 has higher interaction intensity with native eIF2 α after light treatment. We also performed BiFC to further validate the interaction between eIF2 α homologs and SPA1 in planta (Fig. 3d). The BiFC results showed that both homologs of eIF2 α can interact with SPA1 in the cytosol under light condition. Collectively, these results suggest that the interactions between SPA1 and eIF2 α homologs are light-dependent and localized in the cytosol. These results indicate that SPA1 might interact with eIF2 α under light condition, and when overexpressing SPA1 might cause them to aggregate when there are not enough eIF2 α to interact with them.

My concerns were addressed.

#8: eIF2 α -8A transgene repressed protein translation in both dark and light conditions (Figure 7); however, the transgene displayed a more extended hypocotyl elongation. Do the authors comment on this counterintuitive phenotype and how the phenotype is linked to translation?

In our hypothesis, we think that the higher translation efficiency results in

photomorphogenic phenotype, whereas lower translation efficiency results in skotomorphogenic phenotype. It could be the low translation efficiency in eIF2 α -8A transgenic plants triggers a more severe skotomorphogenesis, leading to longer, extended hypocotyl length, a more closed cotyledon and less chlorophyll synthesis. The transcriptome and translome analyses in spaQ could reflect the light-responsive genes regulated at the translational level both under dark and under light condition. We will further investigate the genes regulated at the translational level in mutative eIF2 α transgenic plants. We hope to provide more evidence in our future studies. I suggest using fresh or dry weight to evaluate the translation efficiency eIF2 α variants caused.

#9: Please indicate the SPA1 protein position in Figure 4.

We have added the labels and indicate the protein sizes in the panel in Fig. 4.

My concerns were addressed.

#10: In Figure S2, is it GST or MBP pull-down? Please confirm.

We have corrected this mistake. It is GST pull-down assay as shown in Supplementary Fig. 3 in the revised manuscript.

My concerns were addressed.

Reviewer #3 (Remarks to the Author):

The authors have addressed my concerns in the revised manuscript.

Reviewer #1 (Remarks to the Author):

Questions that would need discussion:

In the *spaQ* mutant, in the dark, translational activity seems high, and it decreases in the light. Why is that? if the proposed role of increasing translation in the light by phosphorylating eIF2 were true, why would translation decrease so much in the light compared to that in the dark? Should it not be the same as in the dark, as what happens in the *cop1* mutant?

We thank the reviewer for pointing this out. Our hypothesis is that SPAs play negative roles under dark but play positive roles in the light condition as has been shown in a recent study (Paik et al., 2019). In the dark, SPAs work together with COP1 to inhibit translation through TOR-RPS6 pathway. Thus, in *spaQ* mutant, the translation activity is high. Upon light exposure, beside the effect of light-induced eIF2 α phosphorylation by SPAs, it is possible that SPAs are also necessary to promote TOR-RPS6 pathway through phosphorylating PIFs and other transcription factors (Paik et al., 2019), thus the translation decreased dramatically in the light. We will try to elucidate this regulatory mechanism and discuss in more detail in our future studies.

We have added this postulation in the discussion part in line 580-585: "Upon exposure to light, SPAs not only induce eIF2 α phosphorylation but are also likely essential for promoting the TOR-RPS6 pathway by phosphorylating PIFs, which subsequently leads to their degradation. PIFs can act as cofactors, enhancing the E3 ligase activity of the COP1-SPA complex (Pham et al., 2018; Xu et al., 2015), thereby inhibiting the TOR-RPS6 pathway. This dual role of SPAs may explain the substantial reduction in translation observed in *spaQ* when exposed to light."

Based on the author's comments, the SPAs negatively regulate TOR-RPS6-mediated protein translation in the dark while positively regulating the process in light. In addition, SPAs promote eIF2 α phosphorylation-dependent protein translation in light. However, even if all the assumptions are correct, they still can not address reviewer #1's questions about why *spaQ* has less translation in light than in darkness. The following figure might be the expected data from reviewer #1. The authors should answer that in Col-0, all the components and signaling for translation are intact. Why is the translational induction upon light just slightly higher than that in *cop1* and *spaQ* mutants in the dark? Furthermore, if eIF2 α phosphorylation is essential for promoting translation, we should see a significant translational induction in Col-0 compared to the deficient mutants like the demonstrating figure below, not a decrease of *spaQ* translation upon light exposure (Figure 1b). The authors need to comment further to explain their unexpected discoveries.

I find that the entire hypothesis built out of the interaction between SPA1 and eIF2 alpha is a bit strange. From what is stated in the manuscript, SPA1 interacts with eIF2 through its kinase domain (when SPA1 holds domains for protein-protein interactions) and the N-ter of eIF2, but it is the C-ter of eIF2 which is phosphorylated. Interactions through the kinase domain are normally very transient... But then, in the subsequent experiments, the authors use only the C-ter to check for the SPA-mediated phosphorylation to avoid the interference of the possible phosphorylation on the N-ter of eIF2. How would SPA1 interact then with eIF2 alpha if the results of the Y2H experiments are real?

We thank the reviewer for the comment. Our Y2H analysis provided a possible interaction pattern in eukaryotic cells; however, the results might not hold true among other experimental system due to possible folding effect, transient/weak interaction, or subcellular localization. In Y2H system, the proteins need to interact in the nucleus to activate the expression of reporter gene. We have moved the Y2H data for eIF2 α fragments to Supplementary Fig. 2. To provide additional evidence that SPA1 might still interact with the C-terminus of eIF2 α , we used another system, semi-*in vivo* co-IP, using TAP-SPA1 seedlings and C-term-eIF2 α proteins purified from *E. coli*, which were the same samples used in our semi-*in vivo* kinase assay, to test their interaction. As shown in Supplementary Fig. 9 in our revised version, TAP-SPA1 could interact with the C-terminal eIF2 α when using light-treated sample, suggesting that the C-terminal eIF2 α alone can be the substrate in our kinase assay as shown in Fig. 4b and c.

I suggest removing the Supplementary Fig. 2 to avoid misleading and providing extra panels in Supplementary Fig. 9 to demonstrate that SPA1 can interact with both the C-ter and N-ter of eIF2alpha in one assay. Otherwise, use the entire length of eIF2alpha to perform the following phosphorylation assays and physiological experiments.

Another comment: why do the authors prefer a semi-*in vivo* co-IP assay instead of an *in vivo* one? Does the C-ter and/or N-ter of eIF2alpha expression have a side effect on plants?

The quality of the polysome profiles is very poor. No conclusion can or should be taken from them. In addition, can the authors explain the nature of the peak that appears between the peaks that they identify as 60S and 80S in Figure 1?

We thank the reviewer for pointing this out. We have replaced the previous profiles with better profile results that were performed previously but in a different batch. We have indicated the peaks of 40S, 60S and 80S in the first profile in Fig. 1.

Concerns were addressed.

L113 and Figure 1A. The manuscript describes the “non-polysomal fractions” as the fractions “with no greater than 2 ribosomes”. Based on this, the disome would be part of it. However, in the figure, the disomes fractions are considered polysomal.

We thank the reviewer for pointing this out. As mentioned above, we have replaced the previous profiles with better profile results that were performed previously but in a different batch. We have indicated the peaks of 40S, 60S and 80S in the first profile in

Fig. 1. The small peak next to 80S on the right represents disome, so our polysomal fraction contains the part with ≥ 3 ribosomes.

Concerns were addressed.

Figure 1B: From the legend, data is normalized to an RNA spike. How did they do this experiment? It is not described in the methods section.

We have provided detailed information regarding the spike-in normalization method used in this study in the “Methods” section in the new version of manuscript. As described in “Isolation of total RNA and polysomal RNA” in line 631: “After the centrifugation, the NP and PL fractions were separately collected and extracted with PCI method after equal amount of spike-in RNA (GeneChip™ Poly-A RNA Control Kit, Applied Biosystems, Catalog #900433) was added to each fraction.... The expression of spike-in RNA, Daptomycin (DAP) was used for normalization.”

Concerns were addressed.

About Supp Figure 1 and the metabolic labeling: It seems that only one experiment has been performed. Also, the manuscript considers that its results parallel that of the profiles. I don't think it does. In any case, it is meaningless without, at least, two additional biological replicates with their respective quantifications.

The AHA labeling experiments were repeated more than 3 times with similar results. We have replaced the previous figures with higher-quality visuals that more accurately depict results closest to the average value as shown in Supplementary Fig. 1 in the revised version.

Concerns were addressed.

Figure 2 and L143: How were the transcriptomic experiments performed?? Please, clarify, as in the text they appear as “RNA-sequencing” but from the legend, they are microarray experiments.

We thank the reviewer for pointing this out. We have corrected the mistake in the legend of Fig. 2. The transcriptomic and translational analyses were performed by RNA-sequencing using total RNA and polysomal RNA, respectively. The detailed methods for RNA isolation and RNA sequencing analysis are provided in the “Methods” section.

Concerns were addressed.

About the experiments performed to check for the interaction between eIF2 and SPA1:
a) Does the TAP-SPA1 construct also carry the Myc tag? If so, it should be stated to understand why in the co-IP experiments, anti-myc is used to detect TAP-SPA1.

Yes, it is cMyc-tagged. The full name of TAP tag is Tandem Affinity Purification tag, which has 9 c-Myc repeats as reported by Saijo et al., 2003 and Paik et al., 2019. We have stated this in the text in line 200 where TAP-SPA1 was first mentioned: “...total extract from tandem affinity purification c-Myc-SPA1 (TAP-SPA1) transgenic seedlings.”

Concerns were addressed.

b) Supp Figure 2: The figure is mislabeled. It is an anti-GST pull down, and not an MBP pull down. Also Given that eIF2-GST is interacting with all the bands of MBP-SPA1 that also appear in the input, I believe that a very important control is missing, that is to check that MBP-SPA1 is not binding the anti-GST resin/beads. MBP alone may not bind it, but SPA1 has not been tested.

We thank the reviewer for pointing this out. We have corrected the statement in the legend as in Supplementary Fig. 3 in the revised version. We have also repeated this pull-down experiment using GST only as negative control. As shown in Supplementary Fig. 3, MBP-SPA1 does not bind to GST only, suggesting that SPA1 was pulled down by eIF2 α .

The pull-down assay is not clear-cut due to the multiple bands of GST-eIF2 α and MBP-SPA1 protein purified from *E. coli*. The binding may be due to non-specific bands interacting with target proteins.

First, I suggest the authors try other purification methods to reduce the non-specific bands, such as using protein size filters to remove non-specific bands. Second, I would recommend using eIF2 α and/or SPA1 antibodies to confirm which band is correct instead of solely dependent on protein size.

c) Also, are the tagged version of SPA and eIF2 functional? Do they complement mutant phenotypes?

Yes, we believe that the tagged version of SPA1 and eIF2 α are functional. As reported by Saijo et al. in 2003, expressing TAP-SPA1 in *spa1-3* mutant can rescue the phenotype in the mutant. We are using the same line as described in Saijo et al. in 2003. Our GFP-tagged eIF2 α transgenic lines also showed significant changes in translation efficiencies and different light responses suggesting that this C-term-GFP-tagged eIF2 α still function to affect these phenotypes (Fig. 8).

Do any studies show that the eIF2 α -GFP is functional? Please list them. Otherwise, do the complementation of the eif2 α mutant. The "GFP-tagged eIF2 α transgenic lines also showed significant changes in translation efficiencies and different light responses" does not mean eIF2 α -GFP functions normally. For example, dominant-negative proteins can cause severe phenotype changes but are not functional (dead versions of WT proteins).

d) Figure 3C and Supp fig 3: Please, describe the eIF2 construct a bit. Under which promoter is it expressed? And also the position of the tag, as in the same figure (Supp 3), on one side appears to be in the N-ter and on the other side, in the C-ter.

The eIF2 α construct built in this study is a GFP-tagged protein fused to the C-terminus of eIF2 α , driven by *Cauliflower mosaic virus* 35S promoter. We have provided the detailed information for "vector construction and transgenic lines generation" in the "Method" section. We have also unified the name of eIF2 α construct as eIF2 α -GFP throughout the revised manuscript.

Concerns were addressed.

e) I have a problem with the co-IPs done using Agrobest. Maybe I am biased as I don't know anyone for whom this system works. Given that, in my opinion, the pull down experiment is lacking an important control, and that trusting the interaction between SPA1 and eIF2 is key to trust the entire manuscript, I would request other more "standard" method to prove the interaction. Ideally, co-IP using SPA1 and *eif2alpha* mutants expressing the tagged versions that complement the phenotype.

We have performed a new set of co-IP experiments using TAP-SPA1 transgenic lines and eIF2 α -GFP transgenic lines as shown in Fig. 3b. We have also performed a new set of co-IP experiments using the cytosolic fraction of TAP-SPA1 to perform IP and use anti-eIF2 α antibody to detect their interaction as shown in Fig. 3c. We have also performed BiFC to provide additional evidence that SPA1 can interact with eIF2 α in the cytosol as shown in Fig. 3d.

Concerns were addressed, but with a suggestion: load an equal amount of TAP-SPA1 protein in IP samples (130kDa), then detect the interactions to rule out the possibility that the interaction in the light is due to high SPA1 protein levels.

Figure 4: Please, indicate the protein sizes in the panels.

We have added and indicate the protein sizes in the panel in Fig. 4.

Concerns were addressed

4a: please, indicate the amounts of proteins used, both for SPA1 and also which are the increasing amounts of eIF2 α .

The amount of both SPA1 and eIF2 α substrates added in our kinase assay was 500 ng as stated in "In vitro kinase assay" method section. We have indicated the increasing amounts of eIF2 α in the figures using "+, ++, and +++" symbols in Fig. 4a and b.

Concerns were addressed

Please, describe in the method section the part corresponding to this experiment. In the main text, SPA1 is tagged with strep and eIF2 is tagged with GST. In methods, eIF2 is tagged with both strep and GST...

As stated in our "Method" section in line 726, eIF2 α was first subcloned into pASK75 for the strep tag and then cloned into pGEX-4T-1 for the GST tag. In line 755, SPA1 was fused with strep tag. We have added this information and modified the text in line 263: "...eIF2 α homologs with GST tagged to their N-terminus and strep tagged to their C-terminus (GST-eIF2 α .1/GST-eIF2 α .2) from bacteria (*E. coli*) to perform *in vitro* kinase assay."

I have concerns about tagging both ends of a protein, such as the GST-eIF2 α -Strep used here, which frequently likely affects the protein functions. Do the authors have any data and/or comments to indicate that this construction does not affect the eIF2 α function?

4b and c: please, discuss why phosphorylation on the C-ter does not explain the amount of phosphorylation of the full-length. Also, how many biological replicates have been performed of these experiments? Why use for eIF2alpha.2 experiment in which the different versions of eIF2alpha are barely expressed?

To avoid over exposure of the fragment proteins, we have lowered the amount added for the fragments. According to the trace amount of C-terminus added in the kinase assay (Fig. 4c), the phosphorylation signal is actually very strong that is comparable to the full-length if normalized. The protein amount added for eIF2 α .2 can be observed in the western blot analysis shown in Supplementary Fig. 8.

Please normalize the phosphorylation signal to the western-blot data in Supplementary Fig. 8 and show the fold change in 4b and c.

In addition, instead of the Coomassie gels, I would like to see WB to be able to quantify phosphorylation relative to the amount of protein produced.

We thank the reviewer for the suggestion. We have provided a western blot result in Supplementary Fig. 8 showing the amount for N-, M-, and C-eIF2 α fragments added in our kinase assay in Fig. 4c.

Concerns were addressed. Please comment on why the *Ecoli*-purified proteins have so many nonspecific bands in this study.

Figure 5B: how was this “semi *in vivo* kinase” experiment done? In the legend, it says that TAP-SPA1 protein was extracted. But in the methods section, L652, it says that for figure 5b, eIF2 alpha proteins were incubated with seedling extracts. If this is so, this is no proof that SPA1 is phosphorylating eIF2 alpha. This shows that the phospho-null versions of the C-ter of eIF2 do not get phosphorylated, and that the WT version gets phosphorylated in the light, but does not show a role of SPA1 on it.

We have deleted the statement in the legend. The semi-*in vivo* kinase assay was performed using seedling extracts. To show that SPAs are necessary for the phosphorylation of eIF2 α , we have repeated this experiment and included *spaQ* seedling sample. As shown in Fig. 5c, in *spaQ* mutants, eIF2 α was less phosphorylated under both dark and light conditions.

Concerns were addressed

Figure 5C: I do not understand the figure... why is the amount of eIF2alpha so low *spaQ* in L compared to the other lanes when the Rpt5 signal is stronger?

We have repeated this phos-tag mobility shift experiment and provided a more significant and higher-quality result as shown in Fig. 5d. The shifted band of phosphorylated eIF2 α was detected in Col-0 but not in *spaQ*. The loading was more equal compared to the previous result.

Due to the nonspecific recognition of eIF2alpha native antibodies and the nature of phostag assay (detecting extra shifting bands), I suggest using PPase to treat the light-treated samples before running phostag gel to confirm that the upper band is because of phosphorylation.

Is that super faint phosphorylation signal in Col-0 in L significant?

As mentioned above, we have repeated this phos-tag mobility shift experiment and provided a more significant and higher-quality result as shown in Fig. 5d. The shifted band of phosphorylated eIF2 α was detected in Col-0 but not in *spaQ*. Together with our new semi-*in vivo* kinase assay using P32 detecting phosphorylated C-eIF2 α in both Col-0 and *spaQ* background and *in vivo* kinase assay detecting native eIF2 α in both Col-0 and *spaQ* backgrounds, we hope these results are more convincing.

Please check the comments above.

L281: the “significant” reduction?

We have deleted the word “significant”. We have also repeated the phos-tag mobility shift experiment and provided a more significant result.

Please check the comments above.

L285: please, explain the mSPA1 construct that has a mutation in the kinase domain. What is this mutation? Does it lack the entire domain? Is it a nucleotide substitution? The LUC-mSPA1/*spaQ* was described previously (Paik et al, 2019; Wang et al., 2021) as stated in our “method” section in line 599. A mutant form of SPA1 (mSPA1) that has the R517E mutation in the Ser/Thr kinase domain was cloned into pCAMBIA C-Luc vector. This mutation was also described when first mentioned in line 280: “We also used a mutated form of SPA1 (mSPA1) that has the R517E mutation in Ser/Thr kinase domain in the kinase assay and no phosphorylation activity was observed (Fig. 4b; Supplementary Fig. 10).”

Concerns were addressed

And as in Figure 1, no conclusions can be made from these profiles.

We have repeated these experiments and provided better profiles with similar results.

Concerns were addressed

I don't think the data allows to think that eIF2 α 2 is “specifically” involved in embryogenesis... it can play other roles plant-wide ...

We have toned down this conclusion as stated in line 404: “Both results suggest that eIF2 α .2 might be involved in embryogenic development and seems to play a more important role in that process compared with eIF2 α .1.”

Concerns were addressed

The WT, Phospho-null, and phospho-mimetic versions of eIF2 alpha should be transformed in the eif2 alpha mutants (transforming the hets) to see if they complement and so, understand the biological significance of the phosphorylation in the C-ter.

We thank the reviewer for the suggestion. We have actually tried transforming the 3

variants into eIF2 α heterozygous mutants many times but failed. We are in the middle of crossing the 3 transgenic lines with the eIF2 α heterozygous mutants to obtain the complementation lines. We are also in the middle of generating the eIF2 α variants driven by its own promoter and transform them into eIF2 α mutants to obtain the complementation lines. However, it might take more than 6 months for us to obtain the F2 generation to be able to screen and analyze the genotypes. We hope we can report the results in our future studies.

Please use full-length versions of eIF2alpha with the native promoter to do the transgenes or do some transient experiments to demonstrate the functionality of full-length eIF2alpha phosphorylation variants.

In Figure 7A, if all transformants are in Col-0 background, why are the ones expressing the 2-8A version not responding normally to light.

We thank the reviewer for the comment. Our suspicion is that the expression of transgene has a dominant effect over the native eIF2 α . It might be because that the protein levels of transgenic eIF2 α variants are much higher than the native eIF2 α and the native eIF2 α weren't sufficient enough to be phosphorylated by SPAs upon light exposure. We have performed a western blotting showing the native eIF2 α amount in Col-0 and transgenic lines as shown in Supplementary Fig. 14. Compared to Col-0, the levels of native eIF2 α were reduced possibly due to the compensation of transgenic proteins. Once the ternary complex is formed together with 8A, it could cause dominant negative function of native eIF2 α . The smaller portion of native eIF2 α might result in the phenotypes of these transgenics.

Concerns were addressed

Figure 8A. Col-0 is not etiolated. It does not show a hook and the cotyledons are opening.

We thank the reviewer for pointing this out. We might have accidentally stretched the hook and moved the cotyledon while taking pictures. We have repeated this experiment and carefully observed the photomorphogenic phenotypes. A better-quality image is provided as shown in Fig. 8A.

Concerns were addressed

Also, please, discuss why a Col-0 background would not be responding normally to light when expressing ectopically the eIF2alpha mutant versions. It can be because the mutant versions sequester the TC, but if so, eIF2 alpha het mutants should also show a similar phenotype?

As mentioned above, we think that the expression of transgene has a dominant effect over the native eIF2 α . It might be because that the protein levels of transgenic eIF2 α variants are much higher than the native eIF2 α and might sequester and dominate the ternary complex formation. We have performed a western blotting showing the native eIF2 α amount in Col-0 and transgenic lines as shown in Supplementary Fig. 14. Compared to Col-0, the levels of native eIF2 α were reduced possibly due to the

compensation of transgenic proteins. Once the ternary complex is formed together with 8A, it could cause the dominant negative function of native eIF2 α . The smaller portion of native eIF2 α might result in the phenotypes of these transgenics. Also, the eIF2 α protein inside eIF2 α heterozygous mutant are still wild-type form of eIF2 α that can be phosphorylated and dephosphorylated by kinases and phosphatases. It might not show similar phenotype as the phosphor-null and phosphor-mimicking transgenic plants.

Concerns were addressed

Fig 9C and D: Please, draw the panels as in A and B, because it seems that GST was pulled-down and that eIF2 β interacted with GST alone with equal intensity.

We have corrected this mistake as shown in Fig. 9c and d.

Concerns were addressed

Methods:

Please, revise all methods as important information is missing from them and it would not be possible to replicate some experiments using the information provided.

We have added new sections including “Vectors construction and generation of transgenic plants”, “Fractionation of nuclear and cytosolic proteins”, “Microscopy of GFP-conjugated protein and BiFC”, “Mobility shift assay”, as well as adding more detailed information in the existing methods.

Concerns were addressed

In vitro pull-down assays: the methods refer to eIF5A and eIF4E that do not appear in the manuscript. Please, rewrite the section using the actual constructs employed in this manuscript.

We have corrected and re-written this method section which should be the assays for eIF2 β and eIF2 γ interactions.

Concerns were addressed

Please, describe the Agrobast method employed here better. It is just a copy of the original paper which also doesn't describe the method very clearly. Which is the “desire cocultivation liquid media” employed in these experiments? How do you cultivate Arabidopsis seedlings in 1 ml of Agrobacterium? Which are the conditions in which you incubate Arabidopsis “in the same growth room for 2 -3 days”?

We have deleted this method as the experiment was replaced with new sets of co-IP experiments without this technique.

Concerns were addressed

The manuscript uses “ribosome-RNA profiling” as the technique to separate polysomes along a sucrose gradient based on the number of ribosomes when its actual name is “polysome profiling”.

L121: “ribosome profiling” is a synonym of “Ribo-seq”, which is not what has been done. The manuscript should refer here to the polysome profiles.

We thank the reviewer for pointing this out. We have corrected this word to “polysome profile” throughout the manuscript.

Concerns were addressed

Figure 3B: the Y2H panel is mislabeled. SD-Leu should be SD-Leu-Trp- and the SD-Leu-Trp- should be SD-Leu-Trp-His.

We have corrected this mistake and added the missing labels in Supplementary Fig. 2 in this new version of manuscript.

Concerns were addressed

L46-47: I would not dare to say that “there is a substantial knowledge of the identities and molecular signatures associated with mRNAs regulated at the translational level”. Knowledge in this field is far from substantial. The great majority of the molecular mechanisms governing translation regulation are still unknown.

We thank the reviewer for the comment. We have modified this statement in line 46-48 in the revised manuscript: “Although there are a few studies reporting the identities and molecular signatures associated with mRNAs regulated at the translational level, little is known about the mechanisms of light-triggered regulation of cytoplasmic translation in plants.”

Concerns were addressed

Other issues:

SPA appears as SAP in many places in the manuscript.

We have corrected these mistakes in the revised manuscript.

Concerns were addressed

SUPPRESSOR OF PHYA-105 (SPA1-4) also appears as SUPPRESSOR OF PHYA-105 1 (SPA1-4).

We have corrected this mistake in the abstract.

Concerns were addressed

L63: I’d put and “However, ” before “Because”. And they don’t only have plant-specific translation control mechanisms. They also have a plant-specific translational machinery with plant-specific translation factors.

We have modified this statement as in line 63: “...however, plants have specific translational machinery with specific translation factors because of their unique biological activities, such as photosynthesis and the capacity to respond to stresses.”

Concerns were addressed

L137: what does “(DAP)” stands for? DAP stands for Daptomycin gene from *Bacillus subtilis*, a spike-in RNA added for normalization. We have added the full names for DAP in the legend of Fig. 1.

Concerns were addressed

Reviewer #1 (Remarks to the Author):

Questions that would need discussion:

In the *spaQ* mutant, in the dark, translational activity seems high, and it decreases in the light. Why is that? if the proposed role of increasing translation in the light by phosphorylating eIF2 were true, why would translation decrease so much in the light compared to that in the dark? Should it not be the same as in the dark, as what happens in the *cop1* mutant?

Response: We thank the reviewer for pointing this out. Our hypothesis is that SPAs play negative roles under dark but play positive roles in the light condition as has been shown in a recent study (Paik et al., 2019). In the dark, SPAs work together with *COPI* to inhibit translation through TOR-RPS6 pathway. Thus, in *spaQ* mutant, the translation activity is high. Upon light exposure, beside the effect of light-induced eIF2 α phosphorylation by SPAs, it is possible that SPAs are also necessary to promote TOR-RPS6 pathway through phosphorylating PIFs and other transcription factors (Paik et al., 2019), thus the translation decreased dramatically in the light. We will try to elucidate this regulatory mechanism and discuss in more detail in our future studies.

We have added this postulation in the discussion part in line 580-585: “Upon exposure to light, SPAs not only induce eIF2 α phosphorylation but are also likely essential for promoting the TOR-RPS6 pathway by phosphorylating PIFs, which subsequently leads to their degradation. PIFs can act as cofactors, enhancing the E3 ligase activity of the *COPI*-SPA complex (Pham et al., 2018; Xu et al., 2015), thereby inhibiting the TOR- RPS6 pathway. This dual role of SPAs may explain the substantial reduction in translation observed in *spaQ* when exposed to light.”

Based on the author's comments, the SPAs negatively regulate TOR-RPS6-mediated protein translation in the dark while positively regulating the process in light. In addition, SPAs promote eIF2 α phosphorylation-dependent protein translation in light. However, even if all the assumptions are correct, they still cannot address reviewer #1's questions about why *spaQ* has less translation in light than in darkness. The following figure might be the expected data from reviewer #1. The authors should answer that in Col-0, all the components and signaling for translation are intact. Why is the translational induction upon light just slightly higher than that in *cop1* and *spaQ* mutants in the dark? Furthermore, if eIF2 α phosphorylation is essential for promoting translation, we should see a significant translational induction in Col-0 compared to the deficient mutants like

the demonstrating figure below, not a decrease of *spaQ* translation upon light exposure (Figure 1b). The authors need to comment further to explain their unexpected discoveries.

Response: We thank the reviewer for the comment. We think that the translation induction in Col-0 is only slightly higher than *cop1* and *spaQ* mutants because there might be a capacity limit for translation efficiency in Col-0 due to the limited amounts of ribosomes, eIFs or translation elongation factors... etc. Thus, the intact translation induction in Col-0 through TOR-RPS6 and eIF2 pathway upon light exposure may not lead to an additive result as both might be linked to the kinase activity of SPAs under light. Having either one pathway functional might be sufficient to result in a translation efficiency that is more than 30%. However, in *spaQ* mutants, both TOR-RPS6 and eIF2 pathway might be blocked, so the translation efficiency is similar to the level of Col-0 under dark condition.

To provide additional evidence that SPAs might also regulate TOR-RPS6 through auxin signal transduction, we have shown the expression patterns of genes involved in auxin-activated signal transduction and TOR-RPS6 pathway at both transcriptional and translational level in Supplementary Fig. 2. As added in our result in line 171-183, “Interestingly, our translational analysis showed that mRNAs involved in auxin response and auxin-activated signaling pathway are overrepresented in cluster 4, suggesting that SPAs might repress the expression of these mRNAs under dark condition but promote their expression upon light exposure. To test whether SPAs might also regulate TOR-RPS6 pathway through auxin signal transduction, we selected the genes that are involved in TOR-RPS6 pathway and examined their expression patterns at both transcriptional and translational levels. As shown in Supplementary Fig. 2a, genes related to auxin-activated signal transduction show enhanced expression under dark condition but repressed expression upon light exposure. Moreover, 21 selected genes involved in TOR-RPS6 pathway, including TOR, S6K, FCS-LIKE ZINC FINGER (FLZ), LETHAL WITH SEC13 PROTEIN 8 (LST8) and RHO OF PLANTS 2 (ROP2), also show similar pattern (Supplementary Fig. 2b).

These results indicate that SPAs might play contrasting role in regulating translation through repressing auxin-TOR signal transduction under dark condition but promoting auxin-TOR signal transduction upon light exposure.” We hope that these explanations help address the reviewer’s concern.

I find that the entire hypothesis built out of the interaction between SPA1 and eIF2 alpha is a bit strange. From what is stated in the manuscript, SPA1 interacts with eIF2 through its kinase domain (when SPA1 holds domains for protein-protein interactions) and the N-ter of eIF2, but it is the C-ter of eIF2 which is phosphorylated. Interactions through the kinase domain are normally very transient... But then, in the subsequent experiments, the authors use only the C-ter to check for the SPA-mediated phosphorylation to avoid the interference of the possible phosphorylation on the N-ter of eIF2. How would SPA1 interact then with eIF2 alpha if the results of the Y2H experiments are real?

Response: We thank the reviewer for the comment. Our Y2H analysis provided a possible interaction pattern in eukaryotic cells; however, the results might not hold true among other experimental system due to possible folding effect, transient/weak interaction, or subcellular localization. In Y2H system, the proteins need to interact in the nucleus to activate the expression of a reporter gene. We have moved the Y2H data for eIF2 α fragments to Supplementary Fig. 2. To provide additional evidence that SPA1 might still interact with the C-terminus of eIF2 α , we used another system, semi-*in vivo* co-IP, using TAP-SPA1 seedlings and C-term-eIF2 α proteins purified from *E. coli*, which were the same samples used in our semi-*in vivo* kinase assay, to test their interaction. As shown in Supplementary Fig. 9 in our revised version, TAP-SPA1 could interact with the C-terminal eIF2 α when using light-treated sample, suggesting that the C-terminal eIF2 α alone can be the substrate in our kinase assay as shown in Fig. 4b and c.

I suggest removing the Supplementary Fig. 2 to avoid misleading and providing extra panels in Supplementary Fig. 9 to demonstrate that SPA1 can interact with both the C-ter and N-ter of eIF2alpha in one assay. Otherwise, use the entire length of eIF2alpha to perform the following phosphorylation assays and physiological experiments.

Another comment: why do the authors prefer a semi-*in vivo* co-IP assay instead of an *in vivo* one? Does the C-ter and/or N-ter of eIF2alpha expression have a side effect on plants?

Response: We thank the reviewer for the suggestion and comment. We have removed the original Supplementary Fig. 2 and provided extra panels in Supplementary Fig. 9 to demonstrate that SPA1 can interact with both the N-ter and C-ter of eIF2 α using *in vivo* co-IP assay performed by AGROBEST technique. New constructs carrying GFP-tagged eIF2 α -N, -M, and -C were transformed into TAP-SPA1 etiolated seedlings and harvested after exposing to light for 4 h. As added in our result in line 289-293: “To further provide evidence that SPA1 can interact with the C-terminal eIF2 α , we also performed semi-*in vivo* and *in vivo* co-IP of eIF2 α fragments. Results showed that SPA1 interacts with both N- and C-terminal fragments of eIF2 α homologs (Supplementary Fig. 9). However, only C-terminal of eIF2 α can be phosphorylated by SPA1.”

SPA1 has been shown to interact with other proteins through various domains. For example, the kinase domain of SPA1 interacts with the PHR (N-terminal) domain of CRY2 (Zou et al., 2011), while the C-terminal WD40 domain of SPA1 interacts with the CCT domain of CRY1 (Lian et al., 2011). Our study provides another example that SPA1 might interact with its partner with different domains.

References:

Zuo Z, Liu H, Liu B, Liu X, Lin C. Blue light-dependent interaction of CRY2 with SPA1 regulates COP1 activity and floral initiation in Arabidopsis. *Curr Biol*. 2011 May 24;21(10):841-7.

Lian HL, He SB, Zhang YC, Zhu DM, Zhang JY, Jia KP, Sun SX, Li L, Yang HQ. Blue-light-dependent interaction of cryptochrome 1 with SPA1 defines a dynamic signaling mechanism. *Genes Dev*. 2011 May 15;25(10):1023-8.

The quality of the polysome profiles is very poor. No conclusion can or should be taken from them. In addition, can the authors explain the nature of the peak that appears between the peaks that they identify as 60S and 80S in Figure 1?

Response: We thank the reviewer for pointing this out. We have replaced the previous profiles with better profile results that were performed previously but in a different batch. We have indicated the peaks of 40S, 60S and 80S in the first profile in Fig. 1.

Concerns were addressed.

L113 and Figure 1A. The manuscript describes the “non-polysomal fractions” as the fractions “with no greater than 2 ribosomes”. Based on this, the disome would be part of it. However, in the figure, the disomes fractions are considered polysomal.

Response: We thank the reviewer for pointing this out. As mentioned above, we have replaced the previous profiles with better profile results that were performed previously but in a different batch. We have indicated the peaks of 40S, 60S and 80S in the first profile in Fig. 1. The small peak next to 80S on the right represents disome, so our polysomal fraction contains the part with ≥ 3 ribosomes.

Concerns were addressed.

Figure 1B: From the legend, data is normalized to an RNA spike. How did they do this experiment? It is not described in the methods section.

Response: We have provided detailed information regarding the spike-in normalization method used in this study in the “Methods” section in the new version of manuscript. As described in “Isolation of total RNA and polysomal RNA” in line 631: “After the centrifugation, the NP and PL fractions were separately collected and extracted with PCI method after equal amount of spike-in RNA (GeneChip™ Poly-A RNA Control Kit, Applied Biosystems, Catalog #900433) was added to each fraction.... The expression of spike-in RNA, Daptomycin (DAP) was used for normalization.”

Concerns were addressed.

About Supp Figure 1 and the metabolic labeling: It seems that only one experiment has been performed. Also, the manuscript considers that its results parallel that of the profiles. I don't think it does. In any case, it is meaningless without, at least, two additional biological replicates with their respective quantifications.

Response: The AHA labeling experiments were repeated more than 3 times with similar results. We have replaced the previous figures with higher-quality visuals that more accurately depict results closest to the average value as shown in Supplementary Fig. 1 in the revised version.

Concerns were addressed.

Figure 2 and L143: How were the transcriptomic experiments performed?? Please, clarify, as in the text they appear as “RNA-sequencing” but from the legend, they are microarray experiments.

Response: We thank the reviewer for pointing this out. We have corrected the mistake in the legend of Fig. 2. The transcriptomic and translational analyses were performed by RNA-sequencing using total RNA and polysomal RNA, respectively. The detailed methods for RNA isolation and RNA sequencing analysis are provided in the “Methods” section.

Concerns were addressed.

About the experiments performed to check for the interaction between eIF2 and SPA1: a) Does the TAP-SPA1 construct also carry the Myc tag? If so, it should be stated to understand why in the co-IP experiments, anti-myc is used to detect TAP-SPA1. Yes, it is cMyc-tagged. The full name of Response: TAP tag is Tandem Affinity Purification tag, which has 9 c-Myc repeats as reported by Saijo et al., 2003 and Paik et al., 2019. We have stated this in the text in line 200 where TAP-SPA1 was first mentioned: “...total extract from tandem affinity purification c-Myc-SPA1 (TAP-SPA1) transgenic seedlings.”

Concerns were addressed.

b) Supp Figure 2: The figure is mislabeled. It is an anti-GST pull down, and not an MBP pull down. Also Given that eIF2-GST is interacting with all the bands of MBP-SPA1 that also appear in the input, I believe that a very important control is missing, that is to check that MBP-SPA1 is not binding the anti-GST resin/beads. MBP alone may not bind it, but SPA1 has not been tested.

Response: We thank the reviewer for pointing this out. We have corrected the statement in the legend as in Supplementary Fig. 3 in the revised version. We have also repeated this pull-down experiment using GST only as negative control. As shown in Supplementary Fig. 3, MBP-SPA1 does not bind to GST only, suggesting that SPA1 was pulled down by eIF2 α .

The pull-down assay is not clear-cut due to the multiple bands of GST-eIF2 α and MBP-SPA1 protein purified from *E. coli*. The binding may be due to non-specific bands interacting with target proteins.

First, I suggest the authors try other purification methods to reduce the non-specific bands, such as using protein size filters to remove non-specific bands. Second, I would recommend using eIF2 α and/or SPA1 antibodies to confirm which band is correct instead of solely dependent on protein size.

Response: We thank the reviewer for the suggestions. We have modified our purification methods using higher concentration of the detergent during purification and washing. We also added protease inhibitor and DTT to minimize the protein degradation during the purification and incubation time. As shown in Supplementary Fig. 3, we have provided a much clearer result with less non-specific bands for both GST-eIF2 α and MBP-SPA1. In fact, many lower bands might be truncated recombinant proteins that resulted from protein degradation. Although we haven't used SPA1 antibody to confirm the identity of MBP-SPA1 bands due to lack of native SPA1 antibody, we have used eIF2 α antibody to confirm the identity of GST-eIF2 α bands as shown below. We can see that the band near 63 kDa is indeed eIF2 α . Pull-down assays using the same MBP-SPA1 construct were also conducted in Paik et al. (2019) and Wang et al. (2021). Similar pattern for MBP-SPA1 was also observed in these studies.

c) Also, are the tagged version of SPA and eIF2 functional? Do they complement mutant phenotypes?

Response: Yes, we believe that the tagged version of SPA1 and eIF2 α are functional. As reported by Saijo et al. in 2003, expressing TAP-SPA1 in *spa1-3* mutant can rescue the phenotype in the mutant. We are using the same line as described in Saijo et al. in 2003. Our GFP-tagged eIF2 α transgenic lines also showed significant changes in translation efficiencies and different light responses suggesting that this C-term-GFP-tagged eIF2 α still function to affect these phenotypes (Fig. 8).

Do any studies show that the eIF2 α -GFP is functional? Please list them. Otherwise, do the complementation of the *eif2 α* mutant. The "GFP-tagged eIF2 α transgenic lines also showed significant changes in translation efficiencies and different light responses" does not mean eIF2 α -GFP functions normally. For example, dominant-negative proteins can cause severe phenotype changes but are not functional (dead versions of WT proteins).

Response: Previous study reported by Hodgson et al. (2019, as shown below) has shown that the eIF2 α -GFP is functional *in vivo*. They have shown that this eIF2 α -GFP can associate with eIF2B and shuttle into eIF2B bodies in response to stress. Another study by Kondratyev et al. (2007) also tested the functionality of eIF2 α -GFP variants. They have shown that the phosphorylated eIF2 α -GFP is necessary and sufficient for the pericentriolar localization at the ER-derived quality control compartment. Despite previous reports showing that eIF2 α -GFP is functional, we have crossed our eIF2 α -GFP overexpression lines with two *eif2 α* heterozygous knockout mutants (SAIL_864 and SAIL_1156) as their homozygous lines were embryogenic lethal. We have screened for the T2 generation and obtained overexpressors in both homozygous knockout mutant lines which should be lethal without overexpressing eIF2 α -GFP. The silique images were shown in Supplementary Fig. 14b. Also added in our result in line 433-437: "To examine the functionality of GFP conjugated eIF2 α .2, we generated complementary lines of eIF2 α .2-WT, 8A, 8D in *eif2 α .2* mutants. These complementary lines produced viable seeds (Supplementary Fig 14b), in contrast to the embryogenic lethal seeds observed in *eif2 α .2* heterozygous knockout mutants (Supplementary Fig. 13b), suggesting that the GFP fusion eIF2 α .2 proteins are functional."

References:

Hodgson, R. E. et al. Cellular eIF2B subunit localization: implications for the integrated stress response and its control by small molecule drugs. *Mol. Biol. Cell* **30**, 942-958. (2019).

Kondratyev, M. et al. PERK-dependent compartmentalization of ERAD and unfolded protein response machineries during ER stress. *Exp. Cell Res.* **313**, 3395-3407. (2007).

d) Figure 3C and Supp fig 3: Please, describe the eIF2 construct a bit. Under which promoter is it expressed? And also the position of the tag, as in the same figure (Supp 3), on one side appears to be in the N-ter and on the other side, in the C-ter.

The eIF2 α construct built in this study is a GFP-tagged protein fused to the C-terminus of eIF2 α , driven by *Cauliflower mosaic virus 35S* promoter. We have provided the detailed information for “vector construction and transgenic lines generation” in the “Method” section. We have also unified the name of eIF2 α construct as eIF2 α -GFP throughout the revised manuscript.

Concerns were addressed.

e) I have a problem with the co-IPs done using Agrobest. Maybe I am biased as I don't know anyone for whom this system works. Given that, in my opinion, the pull down experiment is lacking an important control, and that trusting the interaction between SPA1 and eIF2 is key to trust the entire manuscript, I would request other more “standard” method to prove the interaction. Ideally, co-IP using SPA1 and *eif2alpha* mutants expressing the tagged versions that complement the phenotype.

Response: We have performed a new set of co-IP experiments using TAP-SPA1 transgenic lines and eIF2 α -GFP transgenic lines as shown in Fig. 3b. We have also performed a new set of co-IP experiments using the cytosolic fraction of TAP-SPA1 to perform IP and use anti-eIF2 α antibody to detect their interaction as shown in Fig. 3c. We have also performed BiFC to provide additional evidence that SPA1 can interact with eIF2 α in the cytosol as shown in Fig. 3d.

Concerns were addressed, but with a suggestion: load an equal amount of TAP-SPA1 protein in IP samples (130kDa), then detect the interactions to rule out the possibility that the interaction in the light is due to high SPA1 protein levels.

Response: We have reperformed the co-IP experiment using the cytosolic fraction of TAP-SPA1 and loaded equal amount of TAP-SPA1 protein in our IP samples. As shown in Fig. 3c, equal amount of TAP-SPA1 were added in our IP samples and eIF2 α only interacts with TAP-SPA1 under light condition.

Figure 4: Please, indicate the protein sizes in the panels.

Response: We have added and indicate the protein sizes in the panel in Fig. 4.

Concerns were addressed

4a: please, indicate the amounts of proteins used, both for SPA1 and also which are the increasing amounts of eIF2 α .

Response: The amount of both SPA1 and eIF2 α substrates added in our kinase assay was 500 ng as stated in “*In vitro* kinase assay” method section. We have indicated the increasing amounts of eIF2 α in the figures using “+, ++, and +++” symbols in Fig. 4a and b.

Concerns were addressed

Please, describe in the method section the part corresponding to this experiment. In the main text, SPA1 is tagged with strep and eIF2 is tagged with GST. In methods, eIF2 is tagged with both strep and GST...

Response: As stated in our “Method” section in line 726, eIF2 α was first subcloned into pASK75 for the strep tag and then cloned into pGEX-4T-1 for the GST tag. In line 755, SPA1 was fused with strep tag. We have added this information and modified the text in line 263: “...eIF2 α homologs with GST tagged to their N-terminus and strep tagged to their C-terminus (GST-eIF2 α .1/GST-eIF2 α .2) from bacteria (*E. coli*) to perform *in vitro* kinase assay.”

I have concerns about tagging both ends of a protein, such as the GST-eIF2 α -Strep used here, which frequently likely affects the protein functions. Do the authors have any data and/or comments to indicate that this construction does not affect the eIF2 α function?

Response: To make eIF2 α more soluble and easier to purify, we used GST tag to generate recombinant protein. GST-eIF2 α recombinant proteins were reported to be functional in many previous studies as shown below. However, they are prone to generate truncated and non-specific bands. Thus, we added an eight-amino acid-strep tag (Trp-Ser-His-Pro-Gln-Phe-Glu-Lys) to make it purer and cleaner. The strep tag is also very small that has a milder effect on protein structure.

References:

Dey, M. et al. (2011). Requirement for kinase-induced conformational change in eukaryotic initiation factor 2 α (eIF2 α) restricts phosphorylation of Ser51. *Proc. Natl. Acad. Sci. U.S.A.* **108**, 4316-4321.

Dey, M. et al. (2005). PKR and GCN2 kinases and guanine nucleotide exchange factor eukaryotic translation initiation factor 2B (eIF2B) recognize overlapping surfaces on eIF2 α . *Mol. Cell. Biol.* **25**, 3063-3075.

4b and c: please, discuss why phosphorylation on the C-ter does not explain the amount of phosphorylation of the full-length. Also, how many biological replicates have been performed of these experiments? Why use for eIF2 α .2 experiment in which the different versions of eIF2 α are barely expressed?

Response: To avoid over exposure of the fragment proteins, we have lowered the amount added for the fragments. According to the trace amount of C-terminus added in the kinase assay (Fig. 4c), the phosphorylation signal is actually very strong that is comparable to the full-length if normalized. The protein amount added for eIF2 α .2 can be observed in the western blot analysis shown in Supplementary Fig. 8.

Please normalize the phosphorylation signal to the western-blot data in Supplementary Fig. 8 and show the fold change in 4b and c.

Response: We thank the reviewer for the suggestion. We have repeated the western blot analyses and obtained a clearer result as shown in Supplementary Fig. 8. We have also normalized the phosphorylation signal to the western blot data as shown in Fig. 4d. As added in our figure legend for Fig. 4d: “A table showing the relative phosphorylation signal intensities of autoradiogram normalized to protein contents detected by immunoblotting as shown in Supplementary Fig. 8. The ratio of the phosphorylation of full length was set to 1 for analysis.” However, the phosphorylation

signal shown in autoradiogram might have additive effect due to longer exposure time, so the normalized signal intensities for C-ter and full length might not be equal.

In addition, instead of the Coomassie gels, I would like to see WB to be able to quantify phosphorylation relative to the amount of protein produced.

Response: We thank the reviewer for the suggestion. We have provided a western blot result in Supplementary Fig. 8 showing the amount for N-, M-, and C-eIF2 α fragments added in our kinase assay in Fig. 4c.

Concerns were addressed. Please comment on why the *Ecoli*-purified proteins have so many nonspecific bands in this study.

Response: We thank the reviewer for the comment. We have provided a new western blot result with better quality in Supplementary Fig. 8. The protein purified in our previous batch for the Western blot analysis has many non-specific bands because the column for purification was reused for more than 10 times. We have purchased new columns for purification and performed new Western blot analysis.

Figure 5B: how was this “semi *in vivo* kinase” experiment done? In the legend, it says that TAP-SPA1 protein was extracted. But in the methods section, L652, it says that for figure 5b, eIF2 alpha proteins were incubated with seedling extracts. If this is so, this is no proof that SPA1 is phosphorylating eIF2 alpha. This shows that the phospho-null versions of the C-ter of eIF2 do not get phosphorylated, and that the WT version gets phosphorylated in the light, but does not show a role of SPA1 on it.

Response: We have deleted the statement in the legend. The semi-*in vivo* kinase assay was performed using seedling extracts. To show that SPAs are necessary for the phosphorylation of eIF2 α , we have repeated this experiment and included *spaQ* seedling sample. As shown in Fig. 5c, in *spaQ* mutants, eIF2 α was less phosphorylated under both dark and light conditions.

Concerns were addressed

Figure 5C: I do not understand the figure... why is the amount of eIF2alpha so low *spaQ* in L compared to the other lanes when the Rpt5 signal is stronger?

Response: We have repeated this phos-tag mobility shift experiment and provided a more significant and higher-quality result as shown in Fig. 5d. The shifted band of phosphorylated eIF2 α was detected in Col-0 but not in *spaQ*. The loading was more equal compared to the previous result.

Due to the nonspecific recognition of eIF2alpha native antibodies and the nature of phostag assay (detecting extra shifting bands), I suggest using PPase to treat the light-treated samples before running phostag gel to confirm that the upper band is because of phosphorylation.

Response: We thank the reviewer for the comment and suggestion. We have reperfomed the phostag assay using CIP treatment to confirm that the upper band is the phosphorylated eIF2 α as shown in Fig. 5d. As added in our result in line 354-356, “The *in vivo* phosphorylation status of eIF2 α was further examined by utilizing 15 μ M Phos-tag containing SDS-PAGE gels. In wild type plants treated with white light, only the inactivated boiled calf intestinal alkaline phosphatase (CIP) treated eIF2 α showed mobility shift.”

Is that super faint phosphorylation signal in Col-0 in L significant?

Response: As mentioned above, we have repeated this phos-tag mobility shift experiment and provided a more significant and higher-quality result as shown in Fig. 5d. The shifted band of phosphorylated eIF2 α was detected in Col-0 but not in *spaQ*. Together with our new semi-*in vivo* kinase assay using P32 detecting phosphorylated C-eIF2 α in both Col-0 and *spaQ* background and *in vivo* kinase assay detecting native eIF2 α in both Col-0 and *spaQ* backgrounds, we hope these results are more convincing.

Please check the comments above.

Response: We thank the reviewer for the comment. As mentioned above, we have reperfomed the phostag assay using CIP treatment to confirm that the upper band is the phosphorylated eIF2 α as shown in Fig. 5d. As added in our result in line 354-356, “The *in vivo* phosphorylation status of eIF2 α was further examined by utilizing 15 μ M Phos-tag containing SDS-PAGE gels. In wild type plants treated with white light, only the inactivated boiled calf intestinal alkaline phosphatase (CIP) treated eIF2 α showed mobility shift.”

L281: the “significant” reduction?

Response: We have deleted the word “significant”. We have also repeated the phos-tag mobility shift experiment and provided a more significant result.

Please check the comments above.

Response: We thank the reviewer for the comment. As mentioned above, we have reperfomed the phostag assay using CIP treatment to confirm that the upper band is the phosphorylated eIF2 α as shown in Fig. 5d. As described in our result in line 354-356, “The *in vivo* phosphorylation status of eIF2 α was further examined by utilizing 15 μ M Phos-tag containing SDS-PAGE gels. In wild type plants treated with white light, only the inactivated boiled calf intestinal alkaline phosphatase (CIP) treated eIF2 α showed mobility shift.”

L285: please, explain the mSPA1 construct that has a mutation in the kinase domain.

What is this mutation? Does it lack the entire domain? Is it a nucleotide substitution?

Response: The LUC-mSPA1/*spaQ* was described previously (Paik et al, 2019; Wang et al., 2021) as stated in our “method” section in line 599. A mutant form of SPA1 (mSPA1) that has the R517E mutation in the Ser/Thr kinase domain was cloned into pCAMBIA C-Luc vector. This mutation was also described when first mentioned in line 280: “We also used a mutated form of SPA1 (mSPA1) that has the R517E mutation in Ser/Thr kinase domain in the kinase assay and no phosphorylation activity was observed (Fig. 4b; Supplementary Fig. 10).”

Concerns were addressed

And as in Figure 1, no conclusions can be made from these profiles.

Response: We have repeated these experiments and provided better profiles with similar results.

Concerns were addressed

I don't think the data allows to think that eIF2alpha 2 is "specifically" involved in embryogenesis... it can play other roles plant-wide ...

Response: We have toned down this conclusion as stated in line 404: "Both results suggest that eIF2 α .2 might be involved in embryogenic development and seems to play a more important role in that process compared with eIF2 α .1."

Concerns were addressed

The WT, Phospho-null, and phospho-mimetic versions of eIF2 alpha should be transformed in the *eif2* alpha mutants (transforming the hets) to see if they complement and so, understand the biological significance of the phosphorylation in the C-ter.

Response: We thank the reviewer for the suggestion. We have actually tried transforming the 3 variants into eIF2 α heterozygous mutants many times but failed. We are in the middle of crossing the 3 transgenic lines with the eIF2 α heterozygous mutants to obtain the complementation lines. We are also in the middle of generating the eIF2 α variants driven by its own promoter and transform them into eIF2 α mutants to obtain the complementation lines. However, it might take more than 6 months for us to obtain the F2 generation to be able to screen and analyze the genotypes. We hope we can report the results in our future studies.

Please use full-length versions of eIF2alpha with the native promoter to do the transgenes or do some transient experiments to demonstrate the functionality of full-length eIF2alpha phosphorylation variants.

Response: We thank the reviewer for the comment and suggestion. According to the reviewer's suggestion (also mentioned above), despite previous reports showing that eIF2 α -GFP is functional, we have crossed our eIF2 α -GFP overexpression lines with two *eif2a* heterozygous knockout mutants (SAIL_864 and SAIL_1156) as their homozygous lines were embryogenic lethal. We have screened for the T2 generation and obtained overexpressors in both homozygous knockout mutant lines which should be lethal without overexpressing eIF2 α -GFP. The silique images were shown in Supplementary Fig. 14b. Also added in our result in line 433-437: "To examine the functionality of GFP conjugated eIF2 α .2, we generated complementary lines of eIF2 α .2-WT, 8A, 8D in *eif2a*.2 mutants. These complementary lines produced viable seeds (Supplementary Fig 14b), in contrast

to the embryogenic lethal seeds observed in *eif2a.2* heterozygous knockout mutants (Supplementary Fig. 13b), suggesting that the GFP fusion eIF2 $\alpha.2$ proteins are functional.”

Moreover, we have also transformed full-length eIF2 α variants to perform transient assays using AGROBEST technique in *spaQ* etiolated seedlings to support that SPAs promote protein translation through phosphorylating eIF2 α C-terminus *in vivo* upon light exposure. As shown in Fig. 7c in this revised manuscript, also described in our result in line 453-462: “To further verify that the translation changes in *spaQ* mutant are due to defective phosphorylation of C-terminal eIF2 α *in vivo*, we have transiently expressed eIF2 $\alpha.2$ -WT, 8A, and 8D using AGROBEST transfection technique and performed polysome profile analyses to see whether ectopic expression of phospho-mimicking eIF2 $\alpha.2$ could rescue the repressed translation in *spaQ* mutants under light condition. As shown in Fig. 7c, both eIF2 $\alpha.2$ -WT and 8A overexpressing lines display similar pattern with *spaQ* (Fig. 1a), where the translation efficiency cannot be highly induced upon light exposure. However, seedlings overexpressing eIF2 $\alpha.2$ -8D showed high translation efficiencies in both dark and light treatments. Consistent with our ribosome profiling data, the quantification results by isolating NP and PL RNA also show the same patterns with polysome profiles (Fig. 7d).” These results also show that the full-length eIF2 α phosphorylation variants might be functional.

In Figure 7A, if all transformants are in Col-0 background, why are the ones expressing the 2-8A version not responding normally to light.

Response: We thank the reviewer for the comment. Our suspicion is that the expression of transgene has a dominant effect over the native eIF2 α . It might be because that the protein levels of transgenic eIF2 α variants are much higher than the native eIF2 α and the native eIF2 α weren't sufficient enough to be phosphorylated by SPAs upon light exposure. We have performed a western blotting showing the native eIF2 α amount in Col-0 and transgenic lines as shown in Supplementary Fig. 14. Compared to Col-0, the levels of native eIF2 α were reduced possibly due to the compensation of transgenic proteins. Once the ternary complex is formed together with 8A, it could cause dominant negative function of native eIF2 α . The smaller portion of native eIF2 α might result in the phenotypes of these transgenics.

Concerns were addressed

Figure 8A. Col-0 is not etiolated. It does not show a hook and the cotyledons are opening.

Response: We thank the reviewer for pointing this out. We might have accidentally stretched the hook and moved the cotyledon while taking pictures. We have repeated this experiment and carefully observed the photomorphogenic phenotypes. A better-quality image is provided as shown in Fig. 8A.

Concerns were addressed

Also, please, discuss why a Col-0 background would not be responding normally to light when expressing ectopically the eIF2 α mutant versions. It can be because the mutant versions sequester the TC, but if so, eIF2 α het mutants should also show a similar phenotype?

Response: As mentioned above, we think that the expression of transgene has a dominant effect over the native eIF2 α . It might be because that the protein levels of transgenic eIF2 α variants are much higher than the native eIF2 α and might sequester and dominate the ternary complex formation. We have performed a western blotting showing the native eIF2 α amount in Col-0 and transgenic lines as shown in Supplementary Fig. 14. Compared to Col-0, the levels of native eIF2 α were reduced possibly due to the compensation of transgenic proteins. Once the ternary complex is formed together with 8A, it could cause the dominant negative function of native eIF2 α . The smaller portion of native eIF2 α might result in the phenotypes of these transgenics. Also, the eIF2 α protein inside eIF2 α heterozygous mutant are still wild-type form of eIF2 α that can be phosphorylated and dephosphorylated by kinases and phosphatases. It might not show similar phenotype as the phosphor-null and phosphor-mimicking transgenic plants.

Concerns were addressed

Fig 9C and D: Please, draw the panels as in A and B, because it seems that GST was pulled-down and that eIF2 β interacted with GST alone with equal intensity.

Response: We have corrected this mistake as shown in Fig. 9c and d.

Concerns were addressed

Methods:

Please, revise all methods as important information is missing from them and it would not be possible to replicate some experiments using the information provided.

Response: We have added new sections including “Vectors construction and generation of transgenic plants”, “Fractionation of nuclear and cytosolic proteins”, “Microscopy of GFP-conjugated protein and BiFC”, “Mobility shift assay”, as well as adding more detailed information in the existing methods.

Concerns were addressed

In vitro pull-down assays: the methods refer to eIF5A and eIF4E that do not appear in the manuscript. Please, rewrite the section using the actual constructs employed in this manuscript.

Response: We have corrected and re-written this method section which should be the assays for eIF2 β and eIF2 γ interactions.

Concerns were addressed

Please, describe the Agrobacterium method employed here better. It is just a copy of the original paper which also doesn't describe the method very clearly. Which is the “desire cocultivation liquid media” employed in these experiments? How do you cultivate Arabidopsis seedlings in 1 ml of Agrobacterium? Which are the conditions in which you incubate Arabidopsis “in the same growth room for 2 -3 days”?

Response: We have deleted this method as the experiment was replaced with new sets of co-IP experiments without this technique.

Concerns were addressed

The manuscript uses “ribosome-RNA profiling” as the technique to separate polysomes along a sucrose gradient based on the number of ribosomes when its actual name is “polysome profiling”.

L121: “ribosome profiling” is a synonym of “Ribo-seq”, which is not what has been done. The manuscript should refer here to the polysome profiles.

Response: We thank the reviewer for pointing this out. We have corrected this word to “polysome profile” throughout the manuscript.

Concerns were addressed

Figure 3B: the Y2H panel is mislabeled. SD-Leu should be SD-Leu-Trp- and the SD- Leu-Trp- should be SD-Leu-Trp-His.

Response: We have corrected this mistake and added the missing labels in Supplementary Fig. 2 in this new version of manuscript.

Concerns were addressed

L46-47: I would not dare to say that “there is a substantial knowledge of the identities and molecular signatures associated with mRNAs regulated at the translational level”. Knowledge in this field is far from substantial. The great majority of the molecular mechanisms governing translation regulation are still unknown.

Response: We thank the reviewer for the comment. We have modified this statement in line 46-48 in the revised manuscript: “Although there are a few studies reporting the identities and molecular signatures associated with mRNAs regulated at the translational level, little is known about the mechanisms of light-triggered regulation of cytoplasmic translation in plants.”

Concerns were addressed

Other issues:

SPA appears as SAP in many places in the manuscript.

Response: We have corrected these mistakes in the revised manuscript.

Concerns were addressed

SUPPRESSOR OF PHYA-105 (SPA1-4) also appears as SUPPRESSOR OF PHYA- 105 1 (SPA1-4).

Response: We have corrected this mistake in the abstract.

Concerns were addressed

L63: I'd put and "However, " before "Because". And they don't only have plant-specific translation control mechanisms. They also have a plant-specific translational machinery with plant-specific translation factors.

Response: We have modified this statement as in line 63: "...however, plants have specific translational machinery with specific translation factors because of their unique biological activities, such as photosynthesis and the capacity to respond to stresses."

Concerns were addressed

L137: what does "(DAP)" stands for?

Response: DAP stands for Daptomycin gene from *Bacillus subtilis*, a spike-in RNA added for normalization. We have added the full names for DAP in the legend of Fig. 1.

Concerns were addressed

Reviewer #2 (Remarks to the Author):

#1: The author claimed that the SPA regulated protein translation via phosphorylating eIF2alpha C-terminus. However, no genetic evidence (epistasis analysis) supports this claim. Instead, they provided separate data by showing that *spaQ* mutant regulates protein translation (Figure 1 and 2), SPA1 could phosphorylate eIF2alpha (Figure 4 and 5), and eIF2alpha phosphorylation regulate protein translation (Figure 7) without closing the regulatory loop.

Ectopic expression of eIF2alpha phospho-variants in *spaQ* mutant should be tested to close the loop by checking if eIF2alpha-8D or 8A could restore *spaQ*'s translation, enhancing translation in darkness while inhibiting translation in light. Importantly, the transgene *spaQ*/eIF2alpha will also

benefit the eIF2 complex assembly assay in Figure 9 and the phos-tag gel assay in Figure 5, which will be stated below.

Response: We thank the reviewer for the comment and suggestion. Actually, we have tried to ectopically express the phosphor-variant of eIF2 α , eIF2 α -8A and -8D, in *spaQ* background by transforming the variants into *spaQ* by floral dipping and crossing. However, we have failed many times to generate the eIF2 α variants in *spaQ* mutants. It might be because that over-expression of the variant in *spaQ* is somehow toxic. We are currently trying to generate the transgenics driven by eIF2 α 's native promoter. Nevertheless, to show that SPAs are necessary for the phosphorylation of eIF2 α , we have repeated this phos-tag mobility shift experiment and provided a more significant and higher-quality result as shown in Fig. 5d. The shifted band of phosphorylated eIF2 α was detected in Col-0 but not in *spaQ*. The loading was more equal compared to the previous result. We have also repeated the semi-in vivo kinase assay and included *spaQ* seedling sample. As shown in Fig. 5c, in *spaQ* mutants, eIF2 α was less phosphorylated under both dark and light conditions. To manifest that SPAs are necessary for the ternary complex formation, we have also performed semi-in vivo pull-down assay and showed that eIF2 α could not interact with eIF2 β in *spaQ* mutants (Supplementary Fig. 16).

The transgenes expressing eIF2 α variants in *spaQ* are critical for the study. Without checking the translational output using them, I am not fully convinced that the translation changes in *spaQ* mutant are due to directly phosphorylate eIF2 α in vivo. Meanwhile, it is difficult for me to imagine the toxic effect of eIF2 α -8D in *spaQ* mutant because, theoretically, it will recover the *spaQ* translation to the WT level in light conditions, leading to better plant growth (Fig. 1). Since the toxic effect is likely due to overexpression of the truncated version of eIF2 α (C-terminus), I look forward to the native promoter-driven eIF2 α C-terminus transgenes in the *spaQ* mutant or would suggest using full-length eIF2 α variants to do the transgenes or at least do some transient assays using *spaQ* protoplasts or AGROBEST established in the study to support further their discoveries that SPAs regulate protein translation in light is through phosphorylating eIF2 α C-terminus in vivo.

Response: We thank the reviewer for the comment and suggestion. Following the reviewer's suggestion, we have transformed full-length eIF2 α variants to perform transient assays using AGROBEST technique in *spaQ* etiolated seedlings to support that SPAs promote protein

translation through phosphorylating eIF2 α C-terminus *in vivo* upon light exposure. As shown in Fig. 7c in this revised manuscript, also described in our result in line 453-462: “To further verify that the translation changes in *spaQ* mutant are due to defective phosphorylation of C-terminal eIF2 α *in vivo*, we have transiently expressed eIF2 α .2-WT, 8A, and 8D using AGROBEST transfection technique and performed polysome profile analyses to see whether ectopic expression of phospho-mimicking eIF2 α .2 could rescue the repressed translation in *spaQ* mutants under light condition. As shown in Fig. 7c, both eIF2 α .2-WT and 8A overexpressing lines display similar pattern with *spaQ* (Fig. 1a), where the translation efficiency cannot be highly induced upon light exposure. However, seedlings overexpressing eIF2 α .2-8D showed high translation efficiencies in both dark and light treatments. Consistent with our ribosome profiling data, the quantification results by isolating NP and PL RNA also show the same patterns with polysome profiles (Fig. 7d).” The transient protein expression levels of eIF2 α .2 variants in *spaQ* mutants were also verified by immunoblotting as shown in Supplementary Fig. S16.

I realized that the authors may need to use p56-eIF2 α phospho-antibodies to check if SPAs could phosphorylate the site S56 in eIF2 α . The assays performed in Figure 4 could not rule out the possibility that SPAs were able to phosphorylate eIF2 α N-terminus (S56) and C-terminus simultaneously because the S56A protein used in Figure 4 ruled out the possibility, which will affect the rationality of the following experiments only using C-terminal eIF2 α to check the photomorphogenesis in Figure 8.

Response: We thank the reviewer for the comment and suggestion. We have used p56-eIF2 α phospho-antibodies to check if SPAs could phosphorylate the site S56 in eIF2 α as shown in the middle panel in Fig. 5d. In *spaQ* mutants, we can still detect similar p56-eIF2 α band as shown in Col-0, suggesting that SPAs are not responsible for the S56 phosphorylation of eIF2 α . Our S56A protein used in Fig. 4 also ruled out the possibility that SPA1 phosphorylate S56A of eIF2 α . Moreover, except for the phosphorylation assay conducted in Fig. 5b and 5c using radioactive ³²P, all the transgenic lines used in the following experiments, including polysome profiles and photomorphogenesis, are the overexpression lines of “full-length” eIF2 α variants, not the C-term.

In the proposed model (Figure 10), the SPA1 represses protein translation through eIF2 complex assembly in dark conditions. However, the translational outputs from *spaQ* mutant (Figure 1) and

eIF2 α .2-8A transgene (Figure 7) under dark conditions are opposite. Furthermore, no protein interaction was detected between SPA1 and eIF2 α in the dark, as shown in Figure 3c. These data raised concerns that SPA-regulated translation is not through eIF2 α phosphorylation, at least in darkness. As the author wrote in the introduction, SPA kinases repress translation in the dark might through TOR/RPS6 pathway, like *COPI*. It will greatly benefit the proposed model to test the *spaQ*/eIF2 α phospho-variant transgenes (*spaQ*/eIF2 α -WT, *spaQ*/eIF2 α -8A, and *spaQ*/eIF2 α -8D) to check the epistasis.

Response: We thank the reviewer for the comment and suggestion. As mentioned above, we have failed many times generating the transgenic lines in *spaQ* background and we are in the middle of generating the ones driven by eIF2 α 's native promoter. For now, we can only hypothesize why this would happen. In fact, we do not think that SPA1 represses translation through eIF2 complex assembly in dark conditions. Under dark condition, SPA1 does not affect translation through eIF2 complex assembly simply because they don't interact with each other. However, SPA1 might repress translation through TOR-RPS6 pathway under dark condition by working together with *COPI*. Our hypothesis is that SPAs regulate both TOR-RPS6 and eIF2 α phosphorylation pathways but independently, and SPAs play negative roles under dark but play positive roles for TOR-RPS6 regulation in the light condition. In the dark, SPAs work together with *COPI* to inhibit translation through TOR-RPS6 pathway. Thus, in *spaQ* mutant, the translation activity is high. Upon light exposure, beside the effect of light-induced eIF2 α phosphorylation by SPAs, it is likely that SPAs are also necessary to promote TOR-RPS6 pathway through phosphorylating PIFs and other transcription factors (Paik et al., 2019), thus the translation decreased dramatically in the light. We will try to elucidate this regulatory mechanism and discuss in more detail in our future studies.

I think the authors agree with my points that SPAs do not regulate protein translation via phosphorylation of eIF2 α in darkness. The authors may need to change the model (Fig. 10) using the TOR-RPS6 module instead of the eIF2 complex in dark conditions to avoid misleading.

Response: We thank the reviewer for the suggestion. We have modified our model as shown in Fig. 10. In the dark condition, the eIF2 complex was substituted with TOR-RPS6 module to avoid misleading. In the dark, SPAs might regulate translation through repressing auxin-activated TOR-RPS6 pathway. As also mentioned above, to provide additional evidence that SPAs might also regulate TOR-RPS6 through auxin signal transduction, we have shown the expression patterns of

genes involved in auxin-activated signal transduction and TOR-RPS6 pathway at both transcriptional and translational level in Supplementary Fig. 2. As added in our result in line 171-183, “Interestingly, our translomic analysis showed that mRNAs involved in auxin response and auxin-activated signaling pathway are overrepresented in cluster 4, suggesting that SPAs might repress the expression of these mRNAs under dark condition but promote their expression upon light exposure. To test whether SPAs might also regulate TOR-RPS6 pathway through auxin signal transduction, we selected the genes that are involved in TOR-RPS6 pathway and examined their expression patterns at both transcriptional and translational levels. As shown in Supplementary Fig. 2a, genes related to auxin-activated signal transduction show enhanced expression under dark condition but repressed expression upon light exposure. Moreover, 21 selected genes involved in TOR-RPS6 pathway, including TOR, S6K, FCS-LIKE ZINC FINGER (FLZ), LETHAL WITH SEC13 PROTEIN 8 (LST8) and RHO OF PLANTS 2 (ROP2), also show similar pattern (Supplementary Fig. 2b). These results indicate that SPAs might play contrasting role in regulating translation through repressing auxin-TOR signal transduction under dark condition but promoting auxin-TOR signal transduction upon light exposure.”

#2: The authors need to present more convincing data for the phosphorylation of eIF2alpha dependency on SPA kinases in vivo.

In Figure 5b, the semi in vivo phosphorylation of eIF2alpha should include *spaQ* mutant because this figure did not provide SPA1 dependency in eIF2alpha phosphorylation, and the plant extracts contain massive other kinases, which may lead to false positive results.

In Figure 5c, because the native antibodies (Please provide cat no.) can detect more than one band in the hypo-phosphorylated conditions (Dark), making it not a perfect match for the assay. Moreover, the third line, which has the slightly increased p-eIF2alpha, has more protein loading than the rest. I suggest generating the *spaQ*/eIF2alpha-GFP transgenes, which were proven to show a unique band in WT, to repeat the phos-tag assay. According to the multiple phosphor-sites predicted in eIF2alpha, I expect strong shifted phosphorylation bands in light and weaker ones in *spaQ* mutant.

Response: We thank the reviewer for the great comment and suggestion. We have actually tried expressing eIF2 α -GFP in Col-0 and *spaQ* background by AGROBEST technique and perform the phos-tag assay. However, the protein size of eIF2 α -GFP was too large to distinguish the shifted

band. We have repeated this phos-tag mobility shift experiment and provided a more significant and higher-quality result as shown in Fig. 5d. The shifted band of phosphorylated eIF2 α was detected in Col-0 but not in *spaQ*. The loading was more equal compared to the previous result. To show that SPAs are necessary for the phosphorylation of eIF2 α , we have repeated the semi-in vivo kinase assay and included *spaQ* seedling sample. As shown in Fig. 5c, in *spaQ* mutants, eIF2 α was less phosphorylated under both dark and light conditions. Together with our new semi-in vivo kinase assay using ³²P detecting phosphorylated C-eIF2 α in both Col-0 and *spaQ* background and in vivo kinase assay detecting native eIF2 α in both Col-0 and *spaQ* backgrounds, we hope these results are more convincing.

The authors partially addressed my concerns. Because AGROBEST is a transient experiment, I would suggest the authors use HA or FLAG-tagged eIF2 α instead of eIF2 α -GFP to perform the band shift assay using phos-tag gel because the HA and FLAG are tiny tags, which will significantly address the big protein size problem of eIF2 α -GFP. If they insist on performing the phos-tag assay using non-specific (which can detect at least two bands) antibodies, I suggest adding another PPase-treated control in Col-0 samples to demonstrate that the shifted band is not a non-specific band in Col-0 background, which is somehow missing in *spaQ* mutant background. Anyway, this study needs PPase treatment to demonstrate that the SPAs-mediated band shift in phos-tag gel is indeed due to phosphorylation in vivo.

Response: We thank the reviewer for the comment and suggestion. We have reformed the phostag assay using CIP treatment to confirm that the upper band is the phosphorylated eIF2 α as shown in Fig. 5d. As added in our result in line 354-356, “The *in vivo* phosphorylation status of eIF2 α was further examined by utilizing 15 μ M Phos-tag containing SDS-PAGE gels. In wild type plants treated with white light, only the inactivated boiled calf intestinal alkaline phosphatase (CIP) treated eIF2 α showed mobility shift.”

#3: The claim that SPA kinases regulate translation via affecting eIF2 complex formation could greatly benefit from additional experiments to check eIF2 assembly in *spaQ* mutant background. For example, check the eIF2 α .2 and eIF2 β interaction in *spaQ*/eIF2 α .2-GFP transgenes. The availability of the eIF2 β native antibody from Dr. Browning's lab, one of the authors in this work, makes the experiment easy to perform.

We thank the reviewer for the suggestion. Due to lack of eIF2 α -GFP/*spaQ* transgenic plants, we have performed a semi-in vivo pull-down experiment using GST-eIF2 α to pull down native eIF2 β in Col-0 and *spaQ* as shown in Supplementary Fig. 16. The results showed that almost no interaction between alpha and eIF2 β can be detected in both Col-0 and *spaQ*, whereas eIF2 $\alpha.2$ interacts with eIF2 β in Col-0 after light treatment but no interaction was detected in *spaQ* background.

Ecoli-purified GST-eIF2 α may lack the necessary modification, which is essential for its function. I suggest using transient expression assay through Arabidopsis protoplasts or AGROBEST instead of semi-in vitro assay to check the assembly of eIF2 complex by transiently expressing different tagged eIF2 α , β , and even γ in Col-0 and *spaQ* mutant and perform Co-IP experiments.

Response: We thank the reviewer for the comment and suggestion. We have performed co-IP experiments using transiently expressed eIF2 β -GFP and eIF2 γ -GFP in both Col-0 and *spaQ* seedlings using AGROBEST transformation method (Fig. 9c, d, f). As described in our result in line 541-547: “To further verify the complex formation in planta, we also check the eIF2 α -eIF2 β and eIF2 α -eIF2 γ interaction in Col-0 and *spaQ* background by performing *in vivo* co-IP assay using AGROBEST technique. Five-day-old etiolated seedlings of Col-0 and *spaQ* were transfected with GFP conjugated eIF2 β and eIF2 γ for transient expression. The results showed that in the light-treated Col-0 sample, eIF2 β and eIF2 γ exhibited a stronger interaction with native eIF2 α . In contrast, the interaction intensity between native eIF2 α and eIF2 β or eIF2 γ in light-treated *spaQ* significantly decreased. (Fig. 9c, d, f).”

#4: Because one of the main focuses in the manuscript is to elucidate how SPA kinases regulate translation efficiency (TE), I suggest calculating TE (TE = mRNApl fold change/mRNAss fold change) instead of showing both the transcriptional and translational changes to simplify the data interpretation in Figure 2, and ranking the GO analysis base on the TE values. The current data in Figure 2 is distracting because we must focus on mRNAss and mRNApl changes.

We thank the reviewer for the suggestion. We have analyzed the TE and added this figure in Fig. 2b in the revised manuscript. As shown in Fig. 2b, most DEG showed increased TE (redder) under dark condition but decreased TE (greener) under light condition, suggesting that SPAs inhibit the translation in dark, but enhance the translation of these genes under light, which is more consistent

with our polysomal profile data. We have also kept the classification analysis showing the GO terms regulated at different clusters. We think this is more informative to support our hypothesis that the translational regulation light-responsive and development-related genes results in photomorphogenic phenotypes as well as other phenotypes.

My concerns were addressed.

#5: Typically, the ratio of polysome (PL)/monosome (NP) in ribosomal profiling is dynamically changing, with more polysome companies with less monosome or the other way around, like the paper (38) cited in this manuscript. However, this dynamic is hard to observe in the profiling figures (Figures 1 and 6), instead showing a similar trend in the manuscript. Could the authors comment on this?

We thank the reviewer for pointing this out. We have replaced the previous profiles with better profile results that were performed previously but in a different batch. We have indicated the peaks of 40S, 60S and 80S in the first profile in Fig. 1. The small peak next to 80S on the right represents disome, so our polysomal fraction contains the part with ≥ 3 ribosomes. We actually do see some dynamic changes in some profiles, especially the size changes in 80S peaks. However, the monosomal fraction sometimes looks a lot bigger because of the increase in disome after light treatment. Our profiles might be different from the paper cited because of the gradient we use (12.5%-60%) might cause different resolution observed and the sensitivity set in different labs. It also might be because that the ribosome biosynthesis is affected in *spaQ* and *cop1-4* backgrounds.

My concerns were addressed.

#6: The AHA labeling assay in Figure S1, *spaQ* mutant showing more newly translated proteins than Col-0 under light conditions, indicating SPAs negatively regulate protein translation in light conditions. Does it contradict the paper's claim that SPAs positively control protein translation under light conditions?

The AHA labeling experiments were repeated more than 3 times with similar results. We have replaced the previous figures with higher-quality visuals that more accurately depict results closest

to the average value as shown in Supplementary Fig. 1 in the revised version. The average value of the 3 biological repeats is provided in the table in Supplementary Fig. 1.

My concerns were addressed.

#7: Protein translation occurs in the cytoplasm, and SPA1 kinase was reported localized in the nucleus. How does SPA1 regulate protein translation? Do SPA kinases have cytosol localization in light conditions? Please check the interaction between SPA1 and eIF2 α in vivo using BiFC assay.

We thank the reviewer for the great suggestion. To first observe the light-dependent localization of SPA1, we generated the construct of GFP fused to the C-terminus of SPA1 driven by CaMV 35S promoter and transformed it into Col-0 protoplasts. As shown in Supplementary Fig. 5, SPA1-GFP localized in the nucleus in 91.3% of the transformed protoplasts incubated under dark for 2 h before observation. However, under light condition, 37.5% the transformed protoplasts have SPA1-GFP forming speckles in the cytosol, suggesting that light does induce some translocation of SPA1 into cytoplasm. To further confirm the interaction between SPA1 and eIF2 α in the cytosol, we isolated cytosolic fraction of TAP-SPA1 from transgenic seedlings and incubated with anti-cMyc to perform co-IP (Fig. 3c). The results showed that the cytosolic TAP-SPA1 has higher interaction intensity with native eIF2 α after light treatment. We also performed BiFC to further validate the interaction between eIF2 α homologs and SPA1 in planta (Fig. 3d). The BiFC results showed that both homologs of eIF2 α can interact with SPA1 in the cytosol under light condition. Collectively, these results suggest that the interactions between SPA1 and eIF2 α homologs are light-dependent and localized in the cytosol. These results indicate that SPA1 might interact with eIF2 α under light condition, and when overexpressing SPA1 might cause them to aggregate when there are not enough eIF2 α to interact with them.

My concerns were addressed.

#8: eIF2 α -8A transgene repressed protein translation in both dark and light conditions (Figure 7); however, the transgene displayed a more extended hypocotyl elongation. Do the authors comment on this counterintuitive phenotype and how the phenotype is linked to translation?

In our hypothesis, we think that the higher translation efficiency results in photomorphogenic phenotype, whereas lower translation efficiency results in skotomorphogenic phenotype. It could be the low translation efficiency in eIF2 α -8A transgenic plants triggers a more severe skotomorphogenesis, leading to longer, extended hypocotyl length, a more closed cotyledon and less chlorophyll synthesis. The transcriptome and translome analyses in *spaQ* could reflect the light-responsive genes regulated at the translational level both under dark and under light condition. We will further investigate the genes regulated at the translational level in mutative eIF2 α transgenic plants. We hope to provide more evidence in our future studies.

I suggest using fresh or dry weight to evaluate the translation efficiency eIF2 α variants caused.

Response: We thank the reviewer for the suggestion. We have measured the fresh and dry weight in these transgenic lines following the reviewer's suggestion (Supplementary Fig. 15a, b). To further verify their translation efficiencies and protein synthesis, we have also performed *de novo* protein synthesis analysis using AHA labeling method (Supplementary Fig. 15d). As described in our result in line 448-453: "To further verify the protein production in these transgenic lines, we measured their fresh and dry weight, as well as relative water content. The results showed that both eIF2 α .2-WT and 8D had higher fresh and dry weight than eIF2 α .2-8A (Supplementary Fig. 15a, b), whereas the relative water content in these lines did not exhibit significant difference (Supplementary Fig. 15c). We also performed *de novo* protein synthesis analysis and obtained similar pattern compared to the result of polysome profiles (Supplementary Fig. 15d)."

#9: Please indicate the SPA1 protein position in Figure 4.

We have added the labels and indicate the protein sizes in the panel in Fig. 4.

My concerns were addressed.

#10: In Figure S2, is it GST or MBP pull-down? Please confirm.

We have corrected this mistake. It is GST pull-down assay as shown in Supplementary Fig. 3 in the revised manuscript.

My concerns were addressed.

Reviewer #3 (Remarks to the Author):

The authors have addressed my concerns in the revised manuscript.

REVIEWERS' COMMENTS

Reviewer #2 (Remarks to the Author):

Thanks for the author's hard work. They addressed all my concerns.

Only one question remains: I do not think it is reasonable to normalize the previous batch of data (fig. 4) to the new set of data (Supplementary Fig. 8), even though it does not affect the author's conclusions.

Reviewer #1 (Remarks to the Author):

Questions that would need discussion:

In the *spaQ* mutant, in the dark, translational activity seems high, and it decreases in the light. Why is that? if the proposed role of increasing translation in the light by phosphorylating eIF2 were true, why would translation decrease so much in the light compared to that in the dark? Should it not be the same as in the dark, as what happens in the *cop1* mutant?

Response: We thank the reviewer for pointing this out. Our hypothesis is that SPAs play negative roles under dark but play positive roles in the light condition as has been shown in a recent study (Paik et al., 2019). In the dark, SPAs work together with COP1 to inhibit translation through TOR-RPS6 pathway. Thus, in *spaQ* mutant, the translation activity is high. Upon light exposure, beside the effect of light-induced eIF2 α phosphorylation by SPAs, it is possible that SPAs are also necessary to promote TOR-RPS6 pathway through phosphorylating PIFs and other transcription factors (Paik et al., 2019), thus the translation decreased dramatically in the light. We will try to elucidate this regulatory mechanism and discuss in more detail in our future studies. We have added this postulation in the discussion part in line 580-585: "Upon exposure to light, SPAs not only induce eIF2 α phosphorylation but are also likely essential for promoting the TOR-RPS6 pathway by phosphorylating PIFs, which subsequently leads to their degradation. PIFs can act as cofactors, enhancing the E3 ligase activity of the COP1-SPA complex (Pham et al., 2018; Xu et al., 2015), thereby inhibiting the TOR- RPS6 pathway. This dual role of SPAs may explain the substantial reduction in translation observed in *spaQ* when exposed to light."

Based on the author's comments, the SPAs negatively regulate TOR-RPS6-mediated protein translation in the dark while positively regulating the process in light. In addition, SPAs promote eIF2 α phosphorylation-dependent protein translation in light. However, even if all the assumptions are correct, they still cannot address reviewer #1's questions about why *spaQ* has less translation in light

than in darkness. The following figure might be the expected data from reviewer #1. The authors should answer that in Col-0, all the components and signaling for translation are intact. Why is the translational induction upon light just slightly higher than that in cop1 and spaQ mutants in the dark? Furthermore, if eIF2 α phosphorylation is essential for promoting translation, we should see a significant translational induction in Col-0 compared to the deficient mutants like the demonstrating figure below, not a decrease of spaQ translation upon light exposure (Figure 1b). The authors need to comment further to explain their unexpected discoveries.

Response: We thank the reviewer for the comment. We think that the translation induction in Col-0 is only slightly higher than cop1 and spaQ mutants because there might be a capacity limit for translation efficiency in Col-0 due to the limited amounts of ribosomes, eIFs or translation

elongation factors... etc. Thus, the intact translation induction in Col-0 through TOR-RPS6 and eIF2 pathway upon light exposure may not lead to an additive result as both might be linked to the kinase activity of SPAs under light. Having either one pathway functional might be sufficient to result in a translation efficiency that is more than 30%. However, in spaQ mutants, both TOR-RPS6 and eIF2 pathway might be blocked, so the translation efficiency is similar to the level of Col-0 under dark condition.

To provide additional evidence that SPAs might also regulate TOR-RPS6 through auxin signal transduction, we have shown the expression patterns of genes involved in auxin-activated signal transduction and TOR-RPS6 pathway at both transcriptional and translational level in Supplementary Fig. 2. As added in our result in line 171-183, "Interestingly, our translomic analysis showed that mRNAs involved in auxin response and auxin-activated signaling pathway are overrepresented in cluster 4, suggesting that SPAs might repress the expression of these mRNAs under dark condition but promote their expression upon light exposure. To test whether SPAs might also regulate TOR-RPS6 pathway through auxin signal transduction, we selected the genes that are involved in TOR-RPS6 pathway and examined their expression patterns at both transcriptional and translational levels. As shown in Supplementary Fig. 2a, genes related to auxin-activated signal transduction show enhanced expression under dark condition but repressed expression upon light exposure. Moreover, 21 selected genes involved in TOR-RPS6 pathway, including TOR, S6K, FCS-LIKE ZINC FINGER (FLZ), LETHAL WITH SEC13 PROTEIN 8 (LST8) and RHO OF PLANTS 2 (ROP2), also show similar pattern (Supplementary Fig. 2b). These results indicate that SPAs might play contrasting role in regulating translation through repressing auxin-TOR signal transduction under dark condition but promoting auxin-TOR signal transduction upon light exposure." We hope that these explanations help address the reviewer's concern.

Concerns were addressed.

I find that the entire hypothesis built out of the interaction between SPA1 and eIF2 alpha is a bit strange. From what is stated in the manuscript, SPA1 interacts with eIF2 through its kinase domain (when SPA1 holds domains for protein-protein interactions) and the N-ter of eIF2, but it is the C-ter of eIF2 which is phosphorylated. Interactions through the kinase domain are normally very transient... But then, in the subsequent experiments, the authors use only the C-ter to check for the SPA-mediated phosphorylation to avoid the interference of the possible phosphorylation on the N-ter of eIF2. How would SPA1 interact then with eIF2 alpha if the results of the Y2H experiments are real?

Response: We thank the reviewer for the comment. Our Y2H analysis provided a possible interaction pattern in eukaryotic cells; however, the results might not hold true among other experimental system due to possible folding effect, transient/weak interaction, or subcellular localization. In Y2H system, the proteins need to interact in the nucleus to activate the expression of a reporter gene. We have moved the Y2H data for eIF2 α fragments to Supplementary Fig. 2. To provide additional evidence that SPA1 might still interact with the C-terminus of eIF2 α , we used another system, semi-in vivo co-IP, using TAP-SPA1 seedlings and C-term-eIF2 α proteins purified from *E. coli*, which were the same samples used in our semi-in vivo kinase assay, to test their interaction. As shown in Supplementary Fig. 9 in our revised version, TAP-SPA1 could interact with the C-terminal eIF2 α when using light-treated sample, suggesting that the C-terminal eIF2 α alone can be the substrate in our kinase assay as shown in Fig. 4b and c.

I suggest removing the Supplementary Fig. 2 to avoid misleading and providing extra panels in Supplementary Fig. 9 to demonstrate that SPA1 can interact with both the C-ter and N-ter of eIF2alpha in one assay. Otherwise, use the entire length of eIF2alpha to perform the following phosphorylation assays and physiological experiments.

Another comment: why do the authors prefer a semi-in vivo co-IP assay instead of an in vivo one? Does the C-ter and/or N-ter of eIF2alpha expression have a side effect on plants?

Response: We thank the reviewer for the suggestion and comment. We have removed the original Supplementary Fig. 2 and provided extra panels in Supplementary Fig. 9 to demonstrate that SPA1 can interact with both the N-ter and C-ter of eIF2 α using in vivo co-IP assay performed by AGROBEST technique. New constructs carrying GFP-tagged eIF2 α -N, -M, and -C were transformed into TAP-SPA1 etiolated seedlings and harvested after exposing to light for 4 h. As added in our result in line 289-293:

“To further provide evidence that SPA1 can interact with the C-terminal eIF2 α , we also performed semi-*in vivo* and *in vivo* co-IP of eIF2 α fragments. Results showed that SPA1 interacts with both N- and C-terminal fragments of eIF2 α homologs

(Supplementary Fig. 9). However, only C-terminal of eIF2 α can be phosphorylated by SPA1.”

SPA1 has been shown to interact with other proteins through various domains. For example, the

kinase domain of SPA1 interacts with the PHR (N-terminal) domain of CRY2 (Zou et al., 2011),

while the C-terminal WD40 domain of SPA1 interacts with the CCT domain of CRY1 (Lian et al., 2011).

Our study provides another example that SPA1 might interact with its partner with different domains.

References:

Zuo Z, Liu H, Liu B, Liu X, Lin C. Blue light-dependent interaction of CRY2 with SPA1 regulates

COP1 activity and floral initiation in Arabidopsis. *Curr Biol.* 2011 May 24;21(10):841-7.

Lian HL, He SB, Zhang YC, Zhu DM, Zhang JY, Jia KP, Sun SX, Li L, Yang HQ. Blue-light-

dependent interaction of cryptochrome 1 with SPA1 defines a dynamic signaling mechanism.

Genes Dev. 2011 May 15;25(10):1023-8.

Concerns were addressed.

The quality of the polysome profiles is very poor. No conclusion can or should be taken from them. In addition, can the authors explain the nature of the peak that appears between the peaks that they identify as 60S and 80S in Figure 1?

Response: We thank the reviewer for pointing this out. We have replaced the previous profiles with better profile results that were performed previously but in a different batch. We have indicated the peaks of 40S, 60S and 80S in the first profile in Fig. 1.

Concerns were addressed.

L113 and Figure 1A. The manuscript describes the “non-polysomal fractions” as the fractions

“with no greater than 2 ribosomes”. Based on this, the disome would be part of it. However, in the figure, the disomes fractions are considered polysomal.

Response: We thank the reviewer for pointing this out. As mentioned above, we have replaced the previous profiles with better profile results that were performed previously but in a different batch. We

have indicated the peaks of 40S, 60S and 80S in the first profile in Fig. 1. The small peak next to 80S on the right represents disome, so our polysomal fraction contains the part with 3 ribosomes.

Concerns were addressed.

Figure 1B: From the legend, data is normalized to an RNA spike. How did they do this experiment? It is not described in the methods section.

Response: We have provided detailed information regarding the spike-in normalization method used in this study in the “Methods” section in the new version of manuscript. As described in “Isolation of total RNA and polysomal RNA” in line 631: “After the centrifugation, the NP and PL fractions were separately collected and extracted with PCI method after equal amount of spike- in RNA (GeneChip™ Poly-A RNA Control Kit, Applied Biosystems, Catalog #900433) was added to each fraction.... The expression of spike-in RNA, Daptomycin (DAP) was used for normalization.”

Concerns were addressed.

About Supp Figure 1 and the metabolic labeling: It seems that only one experiment has been performed. Also, the manuscript considers that its results parallel that of the profiles. I don't think it does. In any case, it is meaningless without, at least, two additional biological replicates with their respective quantifications.

Response: The AHA labeling experiments were repeated more than 3 times with similar results. We have replaced the previous figures with higher-quality visuals that more accurately depict results closest to the average value as shown in Supplementary Fig. 1 in the revised version.

Concerns were addressed.

Figure 2 and L143: How were the transcriptomic experiments performed?? Please, clarify, as in the text they appear as “RNA-sequencing” but from the legend, they are microarray experiments.

Response: We thank the reviewer for pointing this out. We have corrected the mistake in the legend of Fig. 2. The transcriptomic and translomic analyses were performed by RNA-sequencing using total RNA and polysomal RNA, respectively. The detailed methods for RNA isolation and RNA sequencing analysis are provided in the “Methods” section.

Concerns were addressed.

About the experiments performed to check for the interaction between eIF2 and SPA1: a) Does the TAP-SPA1 construct also carry the Myc tag? If so, it should be stated to understand why in the co-IP experiments, anti-myc is used to detect TAP-SPA1.

Response: Yes, it is cMyc-tagged. The full name of TAP tag is Tandem Affinity Purification tag, which has 9 c-Myc repeats as reported by Saijo et al., 2003 and Paik et al., 2019. We have stated this in the text in line 200 where TAP-SPA1 was first mentioned: “...total extract from tandem affinity purification c-Myc-SPA1 (TAP-SPA1) transgenic seedlings.”

Concerns were addressed.

b) Supp Figure 2: The figure is mislabeled. It is an anti-GST pull down, and not an MBP pull down. Also Given that eIF2-GST is interacting with all the bands of MBP-SPA1 that also appear in the input, I believe that a very important control is missing, that is to check that MBP-SPA1 is not binding the anti-GST resin/beads. MBP alone may not bind it, but SPA1 has not been tested.

Response: We thank the reviewer for pointing this out. We have corrected the statement in the legend as in Supplementary Fig. 3 in the revised version. We have also repeated this pull-down experiment using GST only as negative control. As shown in Supplementary Fig. 3, MBP-SPA1 does not bind to GST only, suggesting that SPA1 was pulled down by eIF2 α .

The pull-down assay is not clear-cut due to the multiple bands of GST-eIF2 α and MBP-SPA1 protein purified from Ecoli. The binding may be due to non-specific bands interacting with target proteins. First, I suggest the authors try other purification methods to reduce the non-specific bands, such as using protein size filters to remove non-specific bands. Second, I would recommend using eIF2 α and/or SPA1 antibodies to confirm which band is correct instead of solely dependent on protein size.

Response: We thank the reviewer for the suggestions. We have modified our purification methods using higher concentration of the detergent during purification and washing. We also added protease inhibitor and DTT to minimize the protein degradation during the purification and incubation time. As shown in Supplementary Fig. 3, we have provided a much clearer result with less non-specific bands for both GST-eIF2 α and MBP-SPA1. In fact, many lower bands might be truncated recombinant proteins that resulted from protein degradation. Although we haven't used SPA1 antibody to confirm the identity of MBP-SPA1 bands due to lack of native SPA1 antibody, we have used eIF2 α antibody to confirm the identity of GST-eIF2 α bands as shown below. We can see that the band near 63 kDa is indeed eIF2 α . Pull-down assays using the same MBP-SPA1 construct were also conducted in Paik et al. (2019) and Wang et al. (2021). Similar pattern for MBP-SPA1 was also observed in these studies.

Concerns were addressed.

c) Also, are the tagged version of SPA and eIF2 functional? Do they complement mutant phenotypes?

Response: Yes, we believe that the tagged version of SPA1 and eIF2 α are functional. As reported by Saijo et al. in 2003, expressing TAP-SPA1 in *spa1-3* mutant can rescue the phenotype in the mutant. We are using the same line as described in Saijo et al. in 2003. Our GFP-tagged eIF2 α transgenic lines also showed significant changes in translation efficiencies and different light responses suggesting that this C-term-GFP-tagged eIF2 α still function to affect these phenotypes (Fig. 8).

Do any studies show that the eIF2 α -GFP is functional? Please list them. Otherwise, do the complementation of the *eif2 α* mutant. The "GFP-tagged eIF2 α transgenic lines also showed significant changes in translation efficiencies and different light responses" does not mean eIF2 α -GFP functions normally. For example, dominant-negative proteins can cause severe phenotype changes but are not functional (dead versions of WT proteins).

Response: Previous study reported by Hodgson et al. (2019, as shown below) has shown that the eIF2 α -GFP is functional *in vivo*. They have shown that this eIF2 α -GFP can associate with eIF2B and shuttle into eIF2B bodies in response to stress. Another study by Kondratyev et al. (2007) also tested the functionality of eIF2 α -GFP variants. They have shown that the phosphorylated eIF2 α -GFP is necessary

and sufficient for the pericentriolar localization at the ER-derived quality control compartment. Despite previous reports showing that eIF2 α -GFP is functional, we have crossed our eIF2 α -GFP overexpression lines with two eif2 α heterozygous knockout mutants (SAIL_864 and SAIL_1156) as their homozygous lines were embryogenic lethal. We have screened for the T2 generation and obtained overexpressors in both homozygous knockout mutant lines which should be lethal without overexpressing eIF2 α -GFP. The silique images were shown in Supplementary Fig. 14b. Also added in our result in line 433-437: “To examine the functionality of GFP conjugated eIF2 α .2, we generated complementary lines of eIF2 α .2-WT, 8A, 8D in eif2 α .2 mutants. These complementary lines produced viable seeds (Supplementary Fig 14b), in contrast to the embryogenic lethal seeds observed in eif2 α .2 heterozygous knockout mutants (Supplementary Fig. 13b), suggesting that the GFP fusion eIF2 α .2 proteins are functional.”

References:Hodgson, R. E. et al. Cellular eIF2B subunit localization: implications for the integrated stress response and its control by small molecule drugs. *Mol. Biol. Cell* 30, 942-958. (2019).

Kondratyev, M. et al. PERK-dependent compartmentalization of ERAD and unfolded protein response machineries during ER stress. *Exp. Cell Res.* 313, 3395-3407. (2007).

Concerns were addressed.

d) Figure 3C and Supp fig 3: Please, describe the eIF2 construct a bit. Under which promoter is it expressed? And also the position of the tag, as in the same figure (Supp 3), on one side appears to be in the N-ter and on the other side, in the C-ter.

The eIF2 α construct built in this study is a GFP-tagged protein fused to the C-terminus of eIF2 α , driven by Cauliflower mosaic virus 35S promoter. We have provided the detailed information for “vector construction and transgenic lines generation” in the “Method” section. We have also unified the name of eIF2 α construct as eIF2 α -GFP throughout the revised manuscript.

Concerns were addressed.

e) I have a problem with the co-IPs done using Agrobast. Maybe I am biased as I don't know anyone for whom this system works. Given that, in my opinion, the pull-down experiment is lacking an important control, and that trusting the interaction between SPA1 and eIF2 is key to

trust the entire manuscript, I would request other more “standard” method to prove the interaction. Ideally, co-IP using SPA1 and eif2alpha mutants expressing the tagged versions that complement the phenotype.

Response: We have performed a new set of co-IP experiments using TAP-SPA1 transgenic lines and eIF2 α -GFP transgenic lines as shown in Fig. 3b. We have also performed a new set of co-IP experiments using the cytosolic fraction of TAP-SPA1 to perform IP and use anti-eIF2 α antibody to detect their interaction as shown in Fig. 3c. We have also performed BiFC to provide additional evidence that SPA1 can interact with eIF2 α in the cytosol as shown in Fig. 3d.

Concerns were addressed, but with a suggestion: load an equal amount of TAP-SPA1 protein in IP samples (130kDa), then detect the interactions to rule out the possibility that the interaction in the light is due to high SPA1 protein levels.

Response: We have reperformed the co-IP experiment using the cytosolic fraction of TAP-SPA1 and loaded equal amount of TAP-SPA1 protein in our IP samples. As shown in Fig. 3c, equal amount of TAP-SPA1 were added in our IP samples and eIF2 α only interacts with TAP-SPA1 under light condition.

Figure 4: Please, indicate the protein sizes in the panels.

Response: We have added and indicate the protein sizes in the panel in Fig. 4.

Concerns were addressed.

4a: please, indicate the amounts of proteins used, both for SPA1 and also which are the increasing amounts of eIF2alpha.

Response: The amount of both SPA1 and eIF2 α substrates added in our kinase assay was 500 ng as stated in “In vitro kinase assay” method section. We have indicated the increasing amounts of eIF2 α in the figures using “+, ++, and +++” symbols in Fig. 4a and b.

Concerns were addressed.

Please, describe in the method section the part corresponding to this experiment. In the main text, SPA1 is tagged with strep and eIF2 is tagged with GST. In methods, eIF2 is tagged with both strep and GST...

Response: As stated in our "Method" section in line 726, eIF2 α was first subcloned into pASK75 for the strep tag and then cloned into pGEX-4T-1 for the GST tag. In line 755, SPA1 was fused with strep tag. We have added this information and modified the text in line 263: "...eIF2 α homologs with GST tagged to their N-terminus and strep tagged to their C-terminus (GST-eIF2 α .1/GST-eIF2 α .2) from bacteria (*E. coli*) to perform in vitro kinase assay."

I have concerns about tagging both ends of a protein, such as the GST-eIF2 α -Strep used here, which frequently likely affects the protein functions. Do the authors have any data and/or comments to indicate that this construction does not affect the eIF2 α function?

Response: To make eIF2 α more soluble and easier to purify, we used GST tag to generate recombinant protein. GST-eIF2 α recombinant proteins were reported to be functional in many previous studies as shown below. However, they are prone to generate truncated and non-specific bands. Thus, we added an eight-amino acid-strep tag (Trp-Ser-His-Pro-Gln-Phe-Glu-Lys) to make it purer and cleaner. The strep tag is also very small that has a milder effect on protein structure.

References:

Dey, M. et al. (2011). Requirement for kinase-induced conformational change in eukaryotic initiation factor 2 α (eIF2 α) restricts phosphorylation of Ser51. *Proc. Natl. Acad. Sci. U.S.A.* 108, 4316-4321.

Dey, M. et al. (2005). PKR and GCN2 kinases and guanine nucleotide exchange factor eukaryotic translation initiation factor 2B (eIF2B) recognize overlapping surfaces on eIF2 α . *Mol. Cell. Biol.* 25, 3063-3075.

Concerns were addressed.

4b and c: please, discuss why phosphorylation on the C-ter does not explain the amount of

phosphorylation of the full-length. Also, how many biological replicates have been performed of these experiments? Why use for eIF2alpha.2 experiment in which the different versions of eIF2alpha are barely expressed?

Response: To avoid over exposure of the fragment proteins, we have lowered the amount added for the fragments. According to the trace amount of C-terminus added in the kinase assay (Fig. 4c), the phosphorylation signal is actually very strong that is comparable to the full-length if normalized. The protein amount added for eIF2 α .2 can be observed in the western blot analysis shown in Supplementary Fig. 8.

Please normalize the phosphorylation signal to the western-blot data in Supplementary Fig. 8 and show the fold change in 4b and c.

Response: We thank the reviewer for the suggestion. We have repeated the western blot analyses and obtained a clearer result as shown in Supplementary Fig. 8. We have also normalized the phosphorylation signal to the western blot data as shown in Fig. 4d. As added in our figure legend for Fig. 4d: "A table showing the relative phosphorylation signal intensities of autoradiogram normalized to protein contents detected by immunoblotting as shown in Supplementary Fig. 8. The ratio of the phosphorylation of full length was set to 1 for analysis." However, the phosphorylation signal shown in autoradiogram might have additive effect due to longer exposure time, so the normalized signal intensities for C-ter and full length might not be equal.

Concerns were addressed.

In addition, instead of the Coomassie gels, I would like to see WB to be able to quantify phosphorylation relative to the amount of protein produced.

Response: We thank the reviewer for the suggestion. We have provided a western blot result in Supplementary Fig. 8 showing the amount for N-, M-, and C-eIF2 α fragments added in our kinase assay in Fig. 4c.

Concerns were addressed. Please comment on why the Ecoli-purified proteins have so many nonspecific bands in this study.

Response: We thank the reviewer for the comment. We have provided a new western blot result with better quality in Supplementary Fig. 8. The protein purified in our previous batch for the

Western blot analysis has many non-specific bands because the column for purification was reused for more than 10 times. We have purchased new columns for purification and performed new Western blot analysis.

Figure 5B: how was this “semi in vivo kinase” experiment done? In the legend, it says that TAP-

SPA1 protein was extracted. But in the methods section, L652, it says that for figure 5b, eIF2 alpha proteins were incubated with seedling extracts. If this is so, this is no proof that SPA1 is phosphorylating eIF2 alpha. This shows that the phospho-null versions of the C-ter of eIF2 do not get phosphorylated, and that the WT version gets phosphorylated in the light, but does not show a role of SPA1 on it.

Response: We have deleted the statement in the legend. The semi-in vivo kinase assay was performed using seedling extracts. To show that SPAs are necessary for the phosphorylation of eIF2 α , we have repeated this experiment and included spaQ seedling sample. As shown in Fig. 5c, in spaQ mutants, eIF2 α was less phosphorylated under both dark and light conditions.

Concerns were addressed.

Figure 5C: I do not understand the figure... why is the amount of eIF2alpha so low spaQ in L compared to the other lanes when the Rpt5 signal is stronger?

Response: We have repeated this phos-tag mobility shift experiment and provided a more significant and higher-quality result as shown in Fig. 5d. The shifted band of phosphorylated eIF2 α was detected in Col-0 but not in spaQ. The loading was more equal compared to the previous result.

Due to the nonspecific recognition of eIF2alpha native antibodies and the nature of phostag assay (detecting extra shifting bands), I suggest using PPase to treat the light-treated samples before running phostag gel to confirm that the upper band is because of phosphorylation.

Response: We thank the reviewer for the comment and suggestion. We have reperformed the phostag assay using CIP treatment to confirm that the upper band is the phosphorylated eIF2 α as

shown in Fig. 5d. As added in our result in line 354-356, “The in vivo phosphorylation status of eIF2 α was further examined by utilizing 15 μ M Phos-tag containing SDS-PAGE gels. In wild type plants treated with white light, only the inactivated boiled calf intestinal alkaline phosphatase (CIP) treated eIF2 α showed mobility shift.”

Concerns were addressed.

Is that super faint phosphorylation signal in Col-0 in L significant?

Response: As mentioned above, we have repeated this phos-tag mobility shift experiment and provided a more significant and higher-quality result as shown in Fig. 5d. The shifted band of phosphorylated eIF2 α was detected in Col-0 but not in spaQ. Together with our new semi-in vivo kinase assay using P32 detecting phosphorylated C-eIF2 α in both Col-0 and spaQ background and in vivo kinase assay detecting native eIF2 α in both Col-0 and spaQ backgrounds, we hope these results are more convincing.

Please check the comments above.

Response: We thank the reviewer for the comment. As mentioned above, we have re-performed the phostag assay using CIP treatment to confirm that the upper band is the phosphorylated eIF2 α as shown in Fig. 5d. As added in our result in line 354-356, “The in vivo phosphorylation status of eIF2 α was further examined by utilizing 15 μ M Phos-tag containing SDS-PAGE gels. In wild type plants treated with white light, only the inactivated boiled calf intestinal alkaline phosphatase (CIP) treated eIF2 α showed mobility shift.

Concerns were addressed.

L281: the “significant” reduction?

Response: We have deleted the word “significant”. We have also repeated the phos-tag mobility shift experiment and provided a more significant result.

Please check the comments above.

Response: We thank the reviewer for the comment. As mentioned above, we have reperformed the phostag assay using CIP treatment to confirm that the upper band is the phosphorylated eIF2 α as shown in Fig. 5d. As described in our result in line 354-356, "The in vivo phosphorylation status of eIF2 α was further examined by utilizing 15 μ M Phos-tag containing SDS-PAGE gels. In wild type plants treated with white light, only the inactivated boiled calf intestinal alkaline phosphatase (CIP) treated eIF2 α showed mobility shift."

Concerns were addressed.

L285: please, explain the mSPA1 construct that has a mutation in the kinase domain.

What is this mutation? Does it lack the entire domain? Is it a nucleotide substitution?

Response: The LUC-mSPA1/spaQ was described previously (Paik et al, 2019; Wang et al., 2021) as stated in our "method" section in line 599. A mutant form of SPA1 (mSPA1) that has the R517E mutation in the Ser/Thr kinase domain was cloned into pCAMBIA C-Luc vector. This mutation was also described when first mentioned in line 280: "We also used a mutated form of SPA1 (mSPA1) that has the R517E mutation in Ser/Thr kinase domain in the kinase assay and no phosphorylation activity was observed (Fig. 4b; Supplementary Fig. 10)."

Concerns were addressed.

And as in Figure 1, no conclusions can be made from these profiles.

Response: We have repeated these experiments and provided better profiles with similar results.

Concerns were addressed.

I don't think the data allows to think that eIF2 α 2 is "specifically" involved in embryogenesis... it can play other roles plant-wide ...

Response: We have toned down this conclusion as stated in line 404: "Both results suggest that

eIF2 α .2 might be involved in embryogenic development and seems to play a more important role in that process compared with eIF2 α .1.”

Concerns were addressed.

The WT, Phospho-null, and phospho-mimetic versions of eIF2 alpha should be transformed in the eif2 alpha mutants (transforming the hets) to see if they complement and so, understand the biological significance of the phosphorylation in the C-ter.

Response: We thank the reviewer for the suggestion. We have actually tried transforming the 3 variants into eIF2 α heterozygous mutants many times but failed. We are in the middle of crossing the 3 transgenic lines with the eIF2 α heterozygous mutants to obtain the complementation lines. We are also in the middle of generating the eIF2 α variants driven by its own promoter and transform them into eIF2 α mutants to obtain the complementation lines. However, it might take more than 6 months for us to obtain the F2 generation to be able to screen and analyze the genotypes. We hope we can report the results in our future studies.

Please use full-length versions of eIF2alpha with the native promoter to do the transgenes or do some transient experiments to demonstrate the functionality of full-length eIF2alpha phosphorylation variants.

Response: We thank the reviewer for the comment and suggestion. According to the reviewer's suggestion (also mentioned above), despite previous reports showing that eIF2 α -GFP is functional, we have crossed our eIF2 α -GFP overexpression lines with two eif2 α heterozygous knockout mutants (SAIL_864 and SAIL_1156) as their homozygous lines were embryogenic lethal. We have screened for the T2 generation and obtained overexpressors in both homozygous knockout mutant lines which should be lethal without overexpressing eIF2 α -GFP. The siliques images were shown in Supplementary Fig. 14b. Also added in our result in line 433-437: “To examine the functionality of GFP conjugated eIF2 α .2, we generated complementary lines of eIF2 α .2-WT, 8A, 8D in eif2 α .2 mutants. These complementary lines produced viable seeds (Supplementary Fig 14b), in contrast to the embryogenic lethal seeds observed in eif2 α .2 heterozygous knockout mutants (Supplementary Fig. 13b), suggesting that the GFP fusion eIF2 α .2 proteins are functional.” Moreover, we have also transformed full-length eIF2 α variants to perform transient assays using AGROBEST technique in spaQ etiolated seedlings to support that SPAs promote protein translation through phosphorylating eIF2 α C-terminus in vivo upon light exposure. As shown in Fig. 7c in this revised manuscript, also described in our result in line 453-462: “To further verify that the translation changes in spaQ mutant are due to defective phosphorylation of C-terminal eIF2 α in vivo, we have transiently expressed eIF2 α .2-WT, 8A, and 8D using AGROBEST transfection technique and performed polysome profile analyses to see whether ectopic expression of phospho-mimicking eIF2 α .2 could rescue the repressed translation in spaQ mutants under light condition. As shown in Fig. 7c, both

eIF2 α .2-WT and 8A overexpressing lines display similar pattern with spaQ (Fig. 1a), where the translation efficiency cannot be highly induced upon light exposure. However, seedlings overexpressing eIF2 α .2-8D showed high translation efficiencies in both dark and light treatments. Consistent with our ribosome profiling data, the quantification results by isolating NP and PL RNA also show the same patterns with polysome profiles (Fig. 7d).” These results also show that the full-length eIF2 α phosphorylation variants might be functional.

Concerns were addressed.

In Figure 7A, if all transformants are in Col-0 background, why are the ones expressing the 2-8A version not responding normally to light.

Response: We thank the reviewer for the comment. Our suspicion is that the expression of transgene has a dominant effect over the native eIF2 α . It might be because that the protein levels of transgenic eIF2 α variants are much higher than the native eIF2 α and the native eIF2 α weren't sufficient enough to be phosphorylated by SPAs upon light exposure. We have performed a western blotting showing the native eIF2 α amount in Col-0 and transgenic lines as shown in Supplementary Fig. 14. Compared to Col-0, the levels of native eIF2 α were reduced possibly due to the compensation of transgenic proteins. Once the ternary complex is formed together with 8A, it could cause dominant negative function of native eIF2 α . The smaller portion of native eIF2 α might result in the phenotypes of these transgenics.

Concerns were addressed.

Figure 8A. Col-0 is not etiolated. It does not show a hook and the cotyledons are opening.

Response: We thank the reviewer for pointing this out. We might have accidentally stretched the hook and moved the cotyledon while taking pictures. We have repeated this experiment and carefully observed the photomorphogenic phenotypes. A better-quality image is provided as shown in Fig. 8A.

Concerns were addressed.

Also, please, discuss why a Col-0 background would not be responding normally to light when expressing ectopically the eIF2 α mutant versions. It can be because the mutant versions sequester the TC, but if so, eIF2 α het mutants should also show a similar phenotype?

Response: As mentioned above, we think that the expression of transgene has a dominant effect

over the native eIF2 α . It might be because that the protein levels of transgenic eIF2 α variants are much higher than the native eIF2 α and might sequester and dominate the ternary complex formation. We have performed a western blotting showing the native eIF2 α amount in Col-0 and

transgenic lines as shown in Supplementary Fig. 14. Compared to Col-0, the levels of native eIF2 α were reduced possibly due to the compensation of transgenic proteins. Once the ternary complex is formed together with 8A, it could cause the dominant negative function of native eIF2 α . The smaller portion of native eIF2 α might result in the phenotypes of these transgenics. Also, the eIF2 α protein inside eIF2 α heterozygous mutant are still wild-type form of eIF2 α that can be phosphorylated and dephosphorylated by kinases and phosphatases. It might not show similar phenotype as the phosphor-null and phosphor-mimicking transgenic plants.

Concerns were addressed.

Fig 9C and D: Please, draw the panels as in A and B, because it seems that GST was pulled-down and that eIF2 β interacted with GST alone with equal intensity.

Response: We have corrected this mistake as shown in Fig. 9c and d.

Concerns were addressed.

Methods:

Please, revise all methods as important information is missing from them and it would not be possible to replicate some experiments using the information provided.

Response: We have added new sections including “Vectors construction and generation of

transgenic plants”, “Fractionation of nuclear and cytosolic proteins”, “Microscopy of GFP-

conjugated protein and BiFC”, “Mobility shift assay”, as well as adding more detailed information in the existing methods.

Concerns were addressed.

In vitro pull-down assays: the methods refer to eIF5A and eIF4E that do not appear in the manuscript. Please, rewrite the section using the actual constructs employed in this manuscript.

We have corrected and re-written this method section which should be the assays for eIF2 β and eIF2 γ interactions.

Concerns were addressed.

Please, describe the Agrobacterium method employed here better. It is just a copy of the original paper which also doesn't describe the method very clearly. Which is the "desire cocultivation liquid media" employed in these experiments? How do you cultivate Arabidopsis seedlings in 1 ml of Agrobacterium? Which are the conditions in which you incubate Arabidopsis "in the same growth room for 2 -3 days"?

We have deleted this method as the experiment was replaced with new sets of co-IP experiments without this technique.

Concerns were addressed.

The manuscript uses "ribosome-RNA profiling" as the technique to separate polysomes along a sucrose gradient based on the number of ribosomes when its actual name is "polysome profiling".

L121: "ribosome profiling" is a synonym of "Ribo-seq", which is not what has been done. The manuscript should refer here to the polysome profiles.

We thank the reviewer for pointing this out. We have corrected this word to "polysome profile" throughout the manuscript.

Concerns were addressed.

Figure 3B: the Y2H panel is mislabeled. SD-Leu should be SD-Leu-Trp- and the SD-Leu-Trp- should be SD-Leu-Trp-His.

We have corrected this mistake and added the missing labels in Supplementary Fig. 2 in this new version of manuscript.

Concerns were addressed.

L46-47: I would not dare to say that “there is a substantial knowledge of the identities and molecular signatures associated with mRNAs regulated at the translational level”. Knowledge in this field is far from substantial. The great majority of the molecular mechanisms governing translation regulation are still unknown.

We thank the reviewer for the comment. We have modified this statement in line 46-48 in the revised manuscript: “Although there are a few studies reporting the identities and molecular signatures associated with mRNAs regulated at the translational level, little is known about the mechanisms of light-triggered regulation of cytoplasmic translation in plants.”

Concerns were addressed.

Other issues:

SPA appears as SAP in many places in the manuscript.

We have corrected these mistakes in the revised manuscript.

Concerns were addressed.

SUPPRESSOR OF PHYA-105 (SPA1-4) also appears as SUPPRESSOR OF PHYA-105 1 (SPA1-4).

We have corrected this mistake in the abstract.

Concerns were addressed.

L63: I'd put and "However," before "Because". And they don't only have plant-specific translation control mechanisms. They also have a plant-specific translational machinery with plant-specific translation factors.

We have modified this statement as in line 63: "...however, plants have specific translational machinery with specific translation factors because of their unique biological activities, such as photosynthesis and the capacity to respond to stresses."

Concerns were addressed.

L137: what does "(DAP)" stands for?

DAP stands for Daptomycin gene from *Bacillus subtilis*, a spike-in RNA added for normalization. We have added the full names for DAP in the legend of Fig. 1.

Concerns were addressed.

Reviewer #2 (Remarks to the Author):

#1: The author claimed that the SPA regulated protein translation via phosphorylating eIF2alpha C-terminus. However, no genetic evidence (epistasis analysis) supports this claim. Instead, they provided separate data by showing that spaQ mutant regulates protein translation (Figure 1 and 2), SPA1 could phosphorylate eIF2alpha (Figure 4 and 5), and eIF2alpha phosphorylation regulate protein translation (Figure 7) without closing the regulatory loop. Ectopic expression of eIF2alpha phospho-variants in spaQ mutant should be tested to close the loop by checking if eIF2alpha-8D or 8A could restore spaQ's translation, enhancing translation in darkness while inhibiting translation in light. Importantly, the transgene spaQ/eIF2alpha will also benefit the eIF2 complex assembly assay in Figure 9 and the phos-tag gel assay in Figure 5, which will be stated below.

Response: We thank the reviewer for the comment and suggestion. Actually, we have tried to ectopically express the phosphor-variant of eIF2 α , eIF2 α -8A and -8D, in spaQ background by

transforming the variants into spaQ by floral dipping and crossing. However, we have failed many times to generate the eIF2 α variants in spaQ mutants. It might be because that over-expression of the variant in spaQ is somehow toxic. We are currently trying to generate the transgenics driven by eIF2 α 's native

promoter. Nevertheless, to show that SPAs are necessary for the phosphorylation of eIF2 α , we have repeated this phos-tag mobility shift experiment and provided a more significant and higher-quality result as shown in Fig. 5d. The shifted band of phosphorylated eIF2 α was detected in Col-0 but not in spaQ. The loading was more equal compared to the previous result. We have also repeated the semi-in vivo kinase assay and included spaQ seedling sample. As shown in Fig. 5c, in spaQ mutants, eIF2 α was less phosphorylated under both dark and light conditions.

To manifest that SPAs are necessary for the ternary complex formation, we have also performed semi-in vivo pull-down assay and showed that eIF2 α could not interact with eIF2 β in spaQ mutants (Supplementary Fig. 16).

The transgenes expressing eIF2 α variants in spaQ are critical for the study. Without checking the translational output using them, I am not fully convinced that the translation changes in spaQ mutant are due to directly phosphorylate eIF2 α in vivo. Meanwhile, it is difficult for me to imagine the toxic effect of eIF2 α -8D in spaQ mutant because, theoretically, it will recover the spaQ translation to the WT level in light conditions, leading to better plant growth (Fig. 1). Since the toxic effect is likely due to overexpression of the truncated version of eIF2 α (C-terminus), I look forward to the native promoter-driven eIF2 α C-terminus transgenes in the spaQ mutant or would suggest using full-length eIF2 α variants to do the transgenes or at least do some transient assays using spaQ protoplasts or AGROBEST established in the study to support further their discoveries that SPAs regulate protein translation in light is through phosphorylating eIF2 α C-terminus in vivo.

Response: We thank the reviewer for the comment and suggestion. Following the reviewer's suggestion, we have transformed full-length eIF2 α variants to perform transient assays using AGROBEST technique in spaQ etiolated seedlings to support that SPAs promote protein translation through phosphorylating eIF2 α C-terminus in vivo upon light exposure. As shown in Fig. 7c in this revised manuscript, also described in our result in line 453-462: "To further verify that the translation changes in spaQ mutant are due to defective phosphorylation of C-terminal eIF2 α in vivo, we have transiently expressed eIF2 α .2-WT, 8A, and 8D using AGROBEST transfection technique and performed polysome profile analyses to see whether ectopic expression of phospho-mimicking eIF2 α .2 could rescue the repressed translation in spaQ mutants under light condition. As shown in Fig. 7c, both eIF2 α .2-WT and 8A overexpressing lines display similar pattern with spaQ (Fig. 1a), where the translation efficiency cannot be highly induced upon light exposure. However, seedlings overexpressing eIF2 α .2-8D showed high translation efficiencies in both dark and light treatments. Consistent with our ribosome profiling data, the quantification results by isolating NP and PL RNA also show the same patterns with polysome profiles (Fig. 7d)." The transient protein expression levels of eIF2 α .2 variants in spaQ mutants were also verified by immunoblotting as shown in Supplementary Fig. S16.

Concerns were addressed.

I realized that the authors may need to use p56-eIF2 α phospho-antibodies to check if SPAs could phosphorylate the site S56 in eIF2 α . The assays performed in Figure 4 could not rule out the

possibility that SPAs were able to phosphorylate eIF2 α N-terminus (S56) and C-terminus simultaneously because the S56A protein used in Figure 4 ruled out the possibility, which will affect the rationality of the following experiments only using C-terminal eIF2 α to check the

photomorphogenesis in Figure 8.

Response: We thank the reviewer for the comment and suggestion. We have used p56-eIF2 α phospho-antibodies to check if SPAs could phosphorylate the site S56 in eIF2 α as shown in the

middle panel in Fig. 5d. In spaQ mutants, we can still detect similar p56-eIF2 α band as shown in

Col-0, suggesting that SPAs are not responsible for the S56 phosphorylation of eIF2 α . Our S56A

protein used in Fig. 4 also ruled out the possibility that SPA1 phosphorylate S56A of eIF2 α .

Moreover, except for the phosphorylation assay conducted in Fig. 5b and 5c using radioactive ^{32}P , all the transgenic lines used in the following experiments, including polysome profiles and

photomorphogenesis, are the overexpression lines of “full-length” eIF2 α variants, not the C-term.

Concerns were addressed.

In the proposed model (Figure 10), the SPA1 represses protein translation through eIF2 complex assembly in dark conditions. However, the translational outputs from spaQ mutant (Figure 1) and eIF2 α .2-8A transgene (Figure 7) under dark conditions are opposite. Furthermore, no protein

interaction was detected between SPA1 and eIF2 α in the dark, as shown in Figure 3c. These

data raised concerns that SPA-regulated translation is not through eIF2 α phosphorylation, at

least in darkness. As the author wrote in the introduction, SPA kinases repress translation in the

dark might through TOR/RPS6 pathway, like COP1. It will greatly benefit the proposed model to test the spaQ/eIF2 α phospho-variant transgenes (spaQ/eIF2 α -WT, spaQ/eIF2 α -8A, and spaQ/eIF2 α -8D) to check the epistasis.

Response: We thank the reviewer for the comment and suggestion. As mentioned above, we have failed many times generating the transgenic lines in spaQ background and we are in the middle of generating the ones driven by eIF2 α 's native promoter. For now, we can only hypothesize why this would happen. In

fact, we do not think that SPA1 represses translation through eIF2 complex assembly in dark conditions.

Under dark condition, SPA1 does not affect translation through eIF2 complex assembly simply because

they don't interact with each other. However, SPA1 might repress translation through TOR-RPS6 pathway under dark condition by working together with COP1. Our hypothesis is that SPAs regulate both TOR-

RPS6 and eIF2 α phosphorylation pathways but independently, and SPAs play negative roles under dark

but play positive roles for TOR-RPS6 regulation in the light condition. In the dark, SPAs work together

with COP1 to inhibit translation through TOR-RPS6 pathway. Thus, in spaQ mutant, the translation

activity is high. Upon light exposure, beside the effect of light-induced eIF2 α phosphorylation by SPAs, it

is likely that SPAs are also necessary to promote TOR-RPS6 pathway through phosphorylating PIFs and other transcription factors (Paik et al., 2019), thus the translation decreased dramatically in the light. We will try to elucidate this regulatory mechanism and discuss in more detail in our future studies.

I think the authors agree with my points that SPAs do not regulate protein translation via

phosphorylation of eIF2 α in darkness. The authors may need to change the model (Fig. 10) using the TOR-RPS6 module instead of the eIF2 complex in dark conditions to avoid misleading.

Response: We thank the reviewer for the suggestion. We have modified our model as shown in Fig.10. In the dark condition, the eIF2 complex was substituted with TOR-RPS6 module to avoid

misleading. In the dark, SPAs might regulate translation through repressing auxin-activated TOR-RPS6 pathway. As also mentioned above, to provide additional evidence that SPAs might also regulate TOR-RPS6 through auxin signal transduction, we have shown the expression patterns of genes involved in auxin-activated signal transduction and TOR-RPS6 pathway at both

transcriptional and translational level in Supplementary Fig. 2. As added in our result in line 171-

183, "Interestingly, our translomic analysis showed that mRNAs involved in auxin response and auxin-activated signaling pathway are overrepresented in cluster 4, suggesting that SPAs might repress the expression of these mRNAs under dark condition but promote their expression upon light exposure. To test whether SPAs might also regulate TOR-RPS6 pathway through auxin signal transduction, we selected the genes that are involved in TOR-RPS6 pathway and examined their expression patterns at both transcriptional and translational levels. As shown in Supplementary Fig. 2a, genes related to auxin-activated signal transduction show enhanced expression under dark condition but repressed expression upon light exposure. Moreover, 21 selected genes involved in TOR-RPS6 pathway, including TOR, S6K, FCS-LIKE ZINC FINGER (FLZ), LETHAL WITH SEC13 PROTEIN 8 (LST8) and RHO OF PLANTS 2 (ROP2), also show similar pattern (Supplementary Fig. 2b). These results indicate that SPAs might play contrasting role in regulating translation through repressing auxin-TOR signal transduction under dark condition but promoting auxin-TOR signal transduction upon light exposure."

#2: The authors need to present more convincing data for the phosphorylation of eIF2 α dependency on SPA kinases in vivo. In Figure 5b, the semi in vivo phosphorylation of eIF2 α should include spaQ mutant because this figure did not provide SPA1 dependency in eIF2 α phosphorylation, and the plant extracts contain massive other kinases, which may lead to false positive results. In Figure 5c, because the native antibodies (Please provide cat no.) can detect more than one band in the hypo-phosphorylated conditions (Dark), making it not a perfect match for the assay. Moreover, the third line, which has the slightly increased p-eIF2 α , has more protein loading than the rest. I suggest generating the spaQ/eIF2 α -GFP transgenes, which were proven to show a unique band in WT, to repeat the phos-tag assay. According to the multiple phosphor-sites predicted in eIF2 α , I expect strong shifted phosphorylation bands in light and weaker ones in spaQ mutant.

Response: We thank the reviewer for the great comment and suggestion. We have actually tried expressing eIF2 α -GFP in Col-0 and spaQ background by AGROBEST technique and perform the phos-tag assay. However, the protein size of eIF2 α -GFP was too large to distinguish the shifted band. We have

repeated this phos-tag mobility shift experiment and provided a more significant and higher-quality result as shown in Fig. 5d. The shifted band of phosphorylated eIF2 α was detected in Col-0 but not in spaQ. The loading was more equal compared to the previous result. To show that SPAs are necessary for the phosphorylation of eIF2 α , we have repeated the semi-in vivo kinase assay and included spaQ seedling sample. As shown in Fig. 5c, in spaQ mutants, eIF2 α was less phosphorylated under both dark and light conditions. Together with our new semi-in vivo kinase assay using 32P detecting phosphorylated C-eIF2 α in both Col-0 and spaQ background and in vivo kinase assay detecting native eIF2 α in both Col-0 and spaQ backgrounds, we hope these results are more convincing.

The authors partially addressed my concerns. Because AGROBEST is a transient experiment, I would suggest the authors use HA or FLAG-tagged eIF2 α instead of eIF2 α -GFP to perform the band shift assay using phos-tag gel because the HA and FLAG are tiny tags, which will significantly address the big protein size problem of eIF2 α -GFP. If they insist on performing the phos-tag assay using non-specific (which can detect at least two bands) antibodies, I suggest adding another PPase-treated control in Col-0 samples to demonstrate that the shifted band is not a non-specific band in Col-0 background, which is somehow missing in spaQ mutant background. Anyway, this study needs PPase treatment to demonstrate that the SPAs-mediated band shift in phos-tag gel is indeed due to phosphorylation in vivo.

Response: We thank the reviewer for the comment and suggestion. We have reperformed the phostag assay using CIP treatment to confirm that the upper band is the phosphorylated eIF2 α as shown in Fig. 5d. As added in our result in line 354-356, "The in vivo phosphorylation status of eIF2 α was further examined by utilizing 15 μ M Phos-tag containing SDS-PAGE gels. In wild type plants treated with white light, only the inactivated boiled calf intestinal alkaline phosphatase (CIP) treated eIF2 α showed mobility shift."

Concerns were addressed.

#3: The claim that SPA kinases regulate translation via affecting eIF2 complex formation could greatly benefit from additional experiments to check eIF2 assembly in spaQ mutant background.

For example, check the eIF2 α .2 and eIF2 β interaction in spaQ/eIF2 α .2-GFP transgenes. The availability of the eIF2 β native antibody from Dr. Browning's lab, one of the authors in this work, makes the experiment easy to perform.

We thank the reviewer for the suggestion. Due to lack of eIF2 α -GFP/spaQ transgenic plants, we have performed a semi-in vivo pull-down experiment using GST-eIF2 α to pull down native eIF2 β in Col-0 and spaQ as shown in Supplementary Fig. 16. The results showed that almost no

interaction between alpha and eIF2 β can be detected in both Col-0 and spaQ, whereas eIF2 α .2

interacts with eIF2 β in Col-0 after light treatment but no interaction was detected in spaQ background.

Ecoli-purified GST-eIF2 α may lack the necessary modification, which is essential for its function. I suggest using transient expression assay through Arabidopsis protoplasts or AGROBEST instead of semi-in vitro assay to check the assembly of eIF2 complex by transiently expressing different tagged eIF2 α , β , and even γ in Col-0 and spaQ mutant and perform Co-IP experiments.

Response: We thank the reviewer for the comment and suggestion. We have performed co-IP experiments using transiently expressed eIF2 β -GFP and eIF2 γ -GFP in both Col-0 and spaQ seedlings using AGROBEST transformation method (Fig. 9c, d, f). As described in our result in line 541-547: "To further verify the complex formation in planta, we also check the eIF2 α -eIF2 β and eIF2 α -eIF2 γ interaction in Col-0 and spaQ background by performing in vivo co-IP assay using AGROBEST technique. Five-day-old etiolated seedlings of Col-0 and spaQ were transfected with GFP conjugated eIF2 β and eIF2 γ for transient expression. The results showed that in the light-treated Col-0 sample, eIF2 β and eIF2 γ exhibited a stronger interaction with native eIF2 α . In contrast, the interaction intensity between native eIF2 α and eIF2 β or eIF2 γ in light-treated spaQ significantly decreased. (Fig. 9c, d, f)."

Concerns were addressed.

#4: Because one of the main focuses in the manuscript is to elucidate how SPA kinases regulate translation efficiency (TE), I suggest calculating TE (TE = mRNApl fold change/mRNAss fold change) instead of showing both the transcriptional and translational changes to simplify the data interpretation in Figure 2, and ranking the GO analysis base on the TE values. The current data in Figure 2 is distracting because we must focus on mRNAss and mRNApl changes.

We thank the reviewer for the suggestion. We have analyzed the TE and added this figure in Fig. 2b in the revised manuscript. As shown in Fig. 2b, most DEG showed increased TE (redder) under dark condition but decreased TE (greener) under light condition, suggesting that SPAs inhibit the translation in dark, but enhance the translation of these genes under light, which is more consistent with our polysomal profile data. We have also kept the classification analysis showing the GO terms regulated at different clusters. We think this is more informative to support our hypothesis that the translational regulation light-responsive and development-related genes results in photomorphogenic phenotypes as well as other phenotypes.

My concerns were addressed.

#5: Typically, the ratio of polysome (PL)/monosome (NP) in ribosomal profiling is dynamically changing, with more polysome companies with less monosome or the other way around, like the paper (38) cited in this manuscript. However, this dynamic is hard to observe in the profiling figures (Figures 1 and 6), instead showing a similar trend in the manuscript. Could the authors comment on this?

We thank the reviewer for pointing this out. We have replaced the previous profiles with better profile results that were performed previously but in a different batch. We have indicated the peaks of 40S, 60S and 80S in the first profile in Fig. 1. The small peak next to 80S on the right represents disome, so our polysomal fraction contains the part with ≥ 3 ribosomes. We actually do see some dynamic changes in some profiles, especially the size changes in 80S peaks. However, the monosomal fraction sometimes looks a lot bigger because of the increase in disome after light treatment. Our profiles might be different from the paper cited because of the gradient we use (12.5%-60%) might cause different resolution observed and the sensitivity set in different labs. It also might be because that the ribosome biosynthesis is affected in spaQ and cop1-4 backgrounds.

My concerns were addressed.

#6: The AHA labeling assay in Figure S1, spaQ mutant showing more newly translated proteins than Col-0 under light conditions, indicating SPAs negatively regulate protein translation in light conditions. Does it contradict the paper's claim that SPAs positively control protein translation under light conditions?

The AHA labeling experiments were repeated more than 3 times with similar results. We have replaced the previous figures with higher-quality visuals that more accurately depict results closest to the average value as shown in Supplementary Fig. 1 in the revised version. The average value of the 3 biological repeats is provided in the table in Supplementary Fig. 1.

My concerns were addressed.

#7: Protein translation occurs in the cytoplasm, and SPA1 kinase was reported localized in the nucleus. How does SPA1 regulate protein translation? Do SPA kinases have cytosol localization in light conditions? Please check the interaction between SPA1 and eIF2alpha in vivo using BiFC assay.

We thank the reviewer for the great suggestion. To first observe the light-dependent localization of SPA1, we generated the construct of GFP fused to the C-terminus of SPA1 driven by CaMV 35S promoter and transformed it into Col-0 protoplasts. As shown in Supplementary Fig. 5, SPA1-GFP localized in the nucleus in 91.3% of the transformed protoplasts incubated under dark for 2 h before observation. However, under light condition, 37.5% the transformed protoplasts have SPA1-GFP forming

speckles in the cytosol, suggesting that light does induce some translocation of SPA1 into cytoplasm. To further confirm the interaction between SPA1 and eIF2 α in the cytosol, we isolated cytosolic fraction of TAP-SPA1 from transgenic seedlings and incubated with anti-cMyc to perform co-IP (Fig. 3c). The results showed that the cytosolic TAP-SPA1 has higher interaction intensity with native eIF2 α after light treatment. We also performed BiFC to further validate the interaction between eIF2 α homologs and SPA1 in planta (Fig. 3d). The BiFC results showed that both homologs of eIF2 α can interact with SPA1 in the cytosol under light condition. Collectively, these results suggest that the interactions between SPA1 and eIF2 α homologs are light-dependent and localized in the cytosol. These results indicate that SPA1 might interact with eIF2 α under light condition, and when overexpressing SPA1 might cause them to aggregate when there are not enough eIF2 α to interact with them.

My concerns were addressed.

#8: eIF2 α -8A transgene repressed protein translation in both dark and light conditions (Figure 7); however, the transgene displayed a more extended hypocotyl elongation. Do the authors comment on this counterintuitive phenotype and how the phenotype is linked to translation?

In our hypothesis, we think that the higher translation efficiency results in photomorphogenic phenotype, whereas lower translation efficiency results in skotomorphogenic phenotype. It could be the low translation efficiency in eIF2 α -8A transgenic plants triggers a more severe skotomorphogenesis, leading to longer, extended hypocotyl length, a more closed cotyledon and less chlorophyll synthesis. The transcriptome and translome analyses in spaQ could reflect the light-responsive genes regulated at the translational level both under dark and under light condition. We will further investigate the genes regulated at the translational level in mutative eIF2 α transgenic plants. We hope to provide more evidence in our future studies.

I suggest using fresh or dry weight to evaluate the translation efficiency eIF2 α variants caused.

Response: We thank the reviewer for the suggestion. We have measured the fresh and dry weight in these transgenic lines following the reviewer's suggestion (Supplementary Fig. 15a, b). To

further verify their translation efficiencies and protein synthesis, we have also performed de novo protein synthesis analysis using AHA labeling method (Supplementary Fig. 15d). As described in our result in line 448-453: "To further verify the protein production in these transgenic lines, we measured their fresh and dry weight, as well as relative water content. The results showed that both eIF2 α .2-WT and 8D had higher fresh and dry weight than eIF2 α .2-8A (Supplementary Fig. 15a, b), whereas the relative water content in these lines did not exhibit significant difference (Supplementary Fig. 15c). We also performed de novo protein synthesis analysis and obtained similar pattern compared to the result of polysome profiles (Supplementary Fig. 15d)."

Concerns were addressed.

#9: Please indicate the SPA1 protein position in Figure 4.

We have added the labels and indicate the protein sizes in the panel in Fig. 4.

My concerns were addressed.

#10: In Figure S2, is it GST or MBP pull-down? Please confirm.

We have corrected this mistake. It is GST pull-down assay as shown in Supplementary Fig. 3 in the revised manuscript.

My concerns were addressed.

Reviewer #1 (Remarks to the Author):

Questions that would need discussion:

In the *spaQ* mutant, in the dark, translational activity seems high, and it decreases in the light. Why is that? if the proposed role of increasing translation in the light by phosphorylating eIF2 were true, why would translation decrease so much in the light compared to that in the dark? Should it not be the same as in the dark, as what happens in the *cop1* mutant?

Response: We thank the reviewer for pointing this out. Our hypothesis is that SPAs play negative roles under dark but play positive roles in the light condition as has been shown in a recent study (Paik et al., 2019). In the dark, SPAs work together with *COPI* to inhibit translation through TOR-RPS6 pathway. Thus, in *spaQ* mutant, the translation activity is high. Upon light exposure, beside the effect of light-induced eIF2 α phosphorylation by SPAs, it is possible that SPAs are also necessary to promote TOR-RPS6 pathway through phosphorylating PIFs and other transcription factors (Paik et al., 2019), thus the translation decreased dramatically in the light. We will try to elucidate this regulatory mechanism and discuss in more detail in our future studies.

We have added this postulation in the discussion part in line 580-585: “Upon exposure to light, SPAs not only induce eIF2 α phosphorylation but are also likely essential for promoting the TOR-RPS6 pathway by phosphorylating PIFs, which subsequently leads to their degradation. PIFs can act as cofactors, enhancing the E3 ligase activity of the *COPI*-SPA complex (Pham et al., 2018; Xu et al., 2015), thereby inhibiting the TOR- RPS6 pathway. This dual role of SPAs may explain the substantial reduction in translation observed in *spaQ* when exposed to light.”

Based on the author's comments, the SPAs negatively regulate TOR-RPS6-mediated protein translation in the dark while positively regulating the process in light. In addition, SPAs promote eIF2 α phosphorylation-dependent protein translation in light. However, even if all the assumptions are correct, they still cannot address reviewer #1's questions about why *spaQ* has less translation in light than in darkness. The following figure might be the expected data from reviewer #1. The authors should answer that in Col-0, all the components and signaling for translation are intact. Why is the translational induction upon light just slightly higher than that in *cop1* and *spaQ* mutants in the dark? Furthermore, if eIF2 α phosphorylation is essential for promoting translation, we should see a significant translational induction in Col-0 compared to the deficient mutants like

the demonstrating figure below, not a decrease of *spaQ* translation upon light exposure (Figure 1b). The authors need to comment further to explain their unexpected discoveries.

Response: We thank the reviewer for the comment. We think that the translation induction in Col-0 is only slightly higher than *cop1* and *spaQ* mutants because there might be a capacity limit for translation efficiency in Col-0 due to the limited amounts of ribosomes, eIFs or translation elongation factors... etc. Thus, the intact translation induction in Col-0 through TOR-RPS6 and eIF2 pathway upon light exposure may not lead to an additive result as both might be linked to the kinase activity of SPAs under light. Having either one pathway functional might be sufficient to result in a translation efficiency that is more than 30%. However, in *spaQ* mutants, both TOR-RPS6 and eIF2 pathway might be blocked, so the translation efficiency is similar to the level of Col-0 under dark condition.

To provide additional evidence that SPAs might also regulate TOR-RPS6 through auxin signal transduction, we have shown the expression patterns of genes involved in auxin-activated signal transduction and TOR-RPS6 pathway at both transcriptional and translational level in Supplementary Fig. 2. As added in our result in line 171-183, “Interestingly, our translational analysis showed that mRNAs involved in auxin response and auxin-activated signaling pathway are overrepresented in cluster 4, suggesting that SPAs might repress the expression of these mRNAs under dark condition but promote their expression upon light exposure. To test whether SPAs might also regulate TOR-RPS6 pathway through auxin signal transduction, we selected the genes that are involved in TOR-RPS6 pathway and examined their expression patterns at both transcriptional and translational levels. As shown in Supplementary Fig. 2a, genes related to auxin-activated signal transduction show enhanced expression under dark condition but repressed expression upon light exposure. Moreover, 21 selected genes involved in TOR-RPS6 pathway, including TOR, S6K, FCS-LIKE ZINC FINGER (FLZ), LETHAL WITH SEC13 PROTEIN 8 (LST8) and RHO OF PLANTS 2 (ROP2), also show similar pattern (Supplementary Fig. 2b).

These results indicate that SPAs might play contrasting role in regulating translation through repressing auxin-TOR signal transduction under dark condition but promoting auxin-TOR signal transduction upon light exposure.” We hope that these explanations help address the reviewer’s concern.

Concerns were addressed.

I find that the entire hypothesis built out of the interaction between SPA1 and eIF2 alpha is a bit strange. From what is stated in the manuscript, SPA1 interacts with eIF2 through its kinase domain (when SPA1 holds domains for protein-protein interactions) and the N-ter of eIF2, but it is the C-ter of eIF2 which is phosphorylated. Interactions through the kinase domain are normally very transient... But then, in the subsequent experiments, the authors use only the C-ter to check for the SPA-mediated phosphorylation to avoid the interference of the possible phosphorylation on the N-ter of eIF2. How would SPA1 interact then with eIF2 alpha if the results of the Y2H experiments are real?

Response: We thank the reviewer for the comment. Our Y2H analysis provided a possible interaction pattern in eukaryotic cells; however, the results might not hold true among other experimental system due to possible folding effect, transient/weak interaction, or subcellular localization. In Y2H system, the proteins need to interact in the nucleus to activate the expression of a reporter gene. We have moved the Y2H data for eIF2 α fragments to Supplementary Fig. 2. To provide additional evidence that SPA1 might still interact with the C-terminus of eIF2 α , we used another system, semi-*in vivo* co-IP, using TAP-SPA1 seedlings and C-term-eIF2 α proteins purified from *E. coli*, which were the same samples used in our semi-*in vivo* kinase assay, to test their interaction. As shown in Supplementary Fig. 9 in our revised version, TAP-SPA1 could interact with the C-terminal eIF2 α when using light-treated sample, suggesting that the C-terminal eIF2 α alone can be the substrate in our kinase assay as shown in Fig. 4b and c.

I suggest removing the Supplementary Fig. 2 to avoid misleading and providing extra panels in Supplementary Fig. 9 to demonstrate that SPA1 can interact with both the C-ter and N-ter of eIF2alpha in one assay. Otherwise, use the entire length of eIF2alpha to perform the following phosphorylation assays and physiological experiments.

Another comment: why do the authors prefer a semi-*in vivo* co-IP assay instead of an *in vivo* one?
Does the C-ter and/or N-ter of eIF2 α expression have a side effect on plants?

Response: We thank the reviewer for the suggestion and comment. We have removed the original Supplementary Fig. 2 and provided extra panels in Supplementary Fig. 9 to demonstrate that SPA1 can interact with both the N-ter and C-ter of eIF2 α using *in vivo* co-IP assay performed by AGROBEST technique. New constructs carrying GFP-tagged eIF2 α -N, -M, and -C were transformed into TAP-SPA1 etiolated seedlings and harvested after exposing to light for 4 h. As added in our result in line 289-293: “To further provide evidence that SPA1 can interact with the C-terminal eIF2 α , we also performed semi-*in vivo* and *in vivo* co-IP of eIF2 α fragments. Results showed that SPA1 interacts with both N- and C-terminal fragments of eIF2 α homologs (Supplementary Fig. 9). However, only C-terminal of eIF2 α can be phosphorylated by SPA1.”

SPA1 has been shown to interact with other proteins through various domains. For example, the kinase domain of SPA1 interacts with the PHR (N-terminal) domain of CRY2 (Zou et al., 2011), while the C-terminal WD40 domain of SPA1 interacts with the CCT domain of CRY1 (Lian et al., 2011). Our study provides another example that SPA1 might interact with its partner with different domains.

References:

Zuo Z, Liu H, Liu B, Liu X, Lin C. Blue light-dependent interaction of CRY2 with SPA1 regulates COP1 activity and floral initiation in Arabidopsis. *Curr Biol*. 2011 May 24;21(10):841-7.

Lian HL, He SB, Zhang YC, Zhu DM, Zhang JY, Jia KP, Sun SX, Li L, Yang HQ. Blue-light-dependent interaction of cryptochrome 1 with SPA1 defines a dynamic signaling mechanism. *Genes Dev*. 2011 May 15;25(10):1023-8.

Concerns were addressed.

The quality of the polysome profiles is very poor. No conclusion can or should be taken from them. In addition, can the authors explain the nature of the peak that appears between the peaks that they identify as 60S and 80S in Figure 1?

Response: We thank the reviewer for pointing this out. We have replaced the previous profiles with better profile results that were performed previously but in a different batch. We have indicated the peaks of 40S, 60S and 80S in the first profile in Fig. 1.

Concerns were addressed.

L113 and Figure 1A. The manuscript describes the “non-polysomal fractions” as the fractions “with no greater than 2 ribosomes”. Based on this, the disome would be part of it. However, in the figure, the disomes fractions are considered polysomal.

Response: We thank the reviewer for pointing this out. As mentioned above, we have replaced the previous profiles with better profile results that were performed previously but in a different batch. We have indicated the peaks of 40S, 60S and 80S in the first profile in Fig. 1. The small peak next to 80S on the right represents disome, so our polysomal fraction contains the part with ≥ 3 ribosomes.

Concerns were addressed.

Figure 1B: From the legend, data is normalized to an RNA spike. How did they do this experiment? It is not described in the methods section.

Response: We have provided detailed information regarding the spike-in normalization method used in this study in the “Methods” section in the new version of manuscript. As described in “Isolation of total RNA and polysomal RNA” in line 631: “After the centrifugation, the NP and PL fractions were separately collected and extracted with PCI method after equal amount of spike-in RNA (GeneChip™ Poly-A RNA Control Kit, Applied Biosystems, Catalog #900433) was added to each fraction.... The expression of spike-in RNA, Daptomycin (DAP) was used for normalization.”

Concerns were addressed.

About Supp Figure 1 and the metabolic labeling: It seems that only one experiment has been performed. Also, the manuscript considers that its results parallel that of the profiles. I don't think it does. In any case, it is meaningless without, at least, two additional biological replicates with their respective quantifications.

Response: The AHA labeling experiments were repeated more than 3 times with similar results. We have replaced the previous figures with higher-quality visuals that more accurately depict results closest to the average value as shown in Supplementary Fig. 1 in the revised version.

Concerns were addressed.

Figure 2 and L143: How were the transcriptomic experiments performed?? Please, clarify, as in the text they appear as “RNA-sequencing” but from the legend, they are microarray experiments.

Response: We thank the reviewer for pointing this out. We have corrected the mistake in the legend of Fig. 2. The transcriptomic and translomic analyses were performed by RNA-sequencing using total RNA and polysomal RNA, respectively. The detailed methods for RNA isolation and RNA sequencing analysis are provided in the “Methods” section.

Concerns were addressed.

About the experiments performed to check for the interaction between eIF2 and SPA1: a) Does the TAP-SPA1 construct also carry the Myc tag? If so, it should be stated to understand why in the co-

IP experiments, anti-myc is used to detect TAP-SPA1. Yes, it is cMyc-tagged. The full name of

Response: TAP tag is Tandem Affinity Purification tag, which has 9 c-Myc repeats as reported by Saijo et al., 2003 and Paik et al., 2019. We have stated this in the text in line 200 where TAP-SPA1 was first mentioned: “...total extract from tandem affinity purification c-Myc-SPA1 (TAP-SPA1) transgenic seedlings.”

Concerns were addressed.

b) Supp Figure 2: The figure is mislabeled. It is an anti-GST pull down, and not an MBP pull down. Also Given that eIF2-GST is interacting with all the bands of MBP-SPA1 that also appear in the input, I believe that a very important control is missing, that is to check that MBP-SPA1 is not binding the anti-GST resin/beads. MBP alone may not bind it, but SPA1 has not been tested.

Response: We thank the reviewer for pointing this out. We have corrected the statement in the legend as in Supplementary Fig. 3 in the revised version. We have also repeated this pull-down experiment using GST only as negative control. As shown in Supplementary Fig. 3, MBP-SPA1 does not bind to GST only, suggesting that SPA1 was pulled down by eIF2 α .

The pull-down assay is not clear-cut due to the multiple bands of GST-eIF2 α and MBP-SPA1 protein purified from *E. coli*. The binding may be due to non-specific bands interacting with target proteins.

First, I suggest the authors try other purification methods to reduce the non-specific bands, such as using protein size filters to remove non-specific bands. Second, I would recommend using eIF2 α and/or SPA1 antibodies to confirm which band is correct instead of solely dependent on protein size.

Response: We thank the reviewer for the suggestions. We have modified our purification methods using higher concentration of the detergent during purification and washing. We also added protease inhibitor and DTT to minimize the protein degradation during the purification and incubation time. As shown in Supplementary Fig. 3, we have provided a much clearer result with less non-specific bands for both GST-eIF2 α and MBP-SPA1. In fact, many lower bands might be truncated recombinant proteins that resulted from protein degradation. Although we haven't used SPA1 antibody to confirm the identity of MBP-SPA1 bands due to lack of native SPA1 antibody, we have used eIF2 α antibody to confirm the identity of GST-eIF2 α bands as shown below. We can see that the band near 63 kDa is indeed eIF2 α . Pull-down assays using the same MBP-SPA1 construct were also conducted in Paik et al. (2019) and Wang et al. (2021). Similar pattern for MBP-SPA1 was also observed in these studies.

Concerns were addressed.

c) Also, are the tagged version of SPA and eIF2 functional? Do they complement mutant phenotypes?

Response: Yes, we believe that the tagged version of SPA1 and eIF2 α are functional. As reported by Saijo et al. in 2003, expressing TAP-SPA1 in *spa1-3* mutant can rescue the phenotype in the mutant. We are using the same line as described in Saijo et al. in 2003. Our GFP-tagged eIF2 α transgenic lines also showed significant changes in translation efficiencies and different light responses suggesting that this C-term-GFP-tagged eIF2 α still function to affect these phenotypes (Fig. 8).

Do any studies show that the eIF2 α -GFP is functional? Please list them. Otherwise, do the complementation of the *eif2 α* mutant. The "GFP-tagged eIF2 α transgenic lines also showed significant changes in translation efficiencies and different light responses" does not mean eIF2 α -GFP functions normally. For example, dominant-negative proteins can cause severe phenotype changes but are not functional (dead versions of WT proteins).

Response: Previous study reported by Hodgson et al. (2019, as shown below) has shown that the eIF2 α -GFP is functional *in vivo*. They have shown that this eIF2 α -GFP can associate with eIF2B and shuttle into eIF2B bodies in response to stress. Another study by Kondratyev et al. (2007) also tested the functionality of eIF2 α -GFP variants. They have shown that the phosphorylated eIF2 α -GFP is necessary and sufficient for the pericentriolar localization at the ER-derived quality control compartment. Despite previous reports showing that eIF2 α -GFP is functional, we have crossed our eIF2 α -GFP overexpression lines with two *eif2 α* heterozygous knockout mutants (SAIL_864 and

SAIL_1156) as their homozygous lines were embryogenic lethal. We have screened for the T2 generation and obtained overexpressors in both homozygous knockout mutant lines which should be lethal without overexpressing eIF2 α -GFP. The silique images were shown in Supplementary Fig. 14b. Also added in our result in line 433-437: “To examine the functionality of GFP conjugated eIF2 α .2, we generated complementary lines of eIF2 α .2-WT, 8A, 8D in *eif2 α .2* mutants. These complementary lines produced viable seeds (Supplementary Fig 14b), in contrast to the embryogenic lethal seeds observed in *eif2 α .2* heterozygous knockout mutants (Supplementary Fig. 13b), suggesting that the GFP fusion eIF2 α .2 proteins are functional.”

References:

Hodgson, R. E. et al. Cellular eIF2B subunit localization: implications for the integrated stress response and its control by small molecule drugs. *Mol. Biol. Cell* **30**, 942-958. (2019).

Kondratyev, M. et al. PERK-dependent compartmentalization of ERAD and unfolded protein response machineries during ER stress. *Exp. Cell Res.* **313**, 3395-3407. (2007).

Concerns were addressed.

d) Figure 3C and Supp fig 3: Please, describe the eIF2 construct a bit. Under which promoter is it expressed? And also the position of the tag, as in the same figure (Supp 3), on one side appears to be in the N-ter and on the other side, in the C-ter.

The eIF2 α construct built in this study is a GFP-tagged protein fused to the C-terminus of eIF2 α , driven by *Cauliflower mosaic virus* 35S promoter. We have provided the detailed information for “vector construction and transgenic lines generation” in the “Method” section. We have also unified the name of eIF2 α construct as eIF2 α -GFP throughout the revised manuscript.

Concerns were addressed.

e) I have a problem with the co-IPs done using Agrobest. Maybe I am biased as I don't know anyone for whom this system works. Given that, in my opinion, the pull down experiment is lacking an important control, and that trusting the interaction between SPA1 and eIF2 is key to trust the entire manuscript, I would request other more “standard” method to prove the interaction.

Ideally, co-IP using SPA1 and *eif2alpha* mutants expressing the tagged versions that complement the phenotype.

Response: We have performed a new set of co-IP experiments using TAP-SPA1 transgenic lines and eIF2 α -GFP transgenic lines as shown in Fig. 3b. We have also performed a new set of co-IP experiments using the cytosolic fraction of TAP-SPA1 to perform IP and use anti-eIF2 α antibody to detect their interaction as shown in Fig. 3c. We have also performed BiFC to provide additional evidence that SPA1 can interact with eIF2 α in the cytosol as shown in Fig. 3d.

Concerns were addressed, but with a suggestion: load an equal amount of TAP-SPA1 protein in IP samples (130kDa), then detect the interactions to rule out the possibility that the interaction in the light is due to high SPA1 protein levels.

Response: We have reperformed the co-IP experiment using the cytosolic fraction of TAP-SPA1 and loaded equal amount of TAP-SPA1 protein in our IP samples. As shown in Fig. 3c, equal amount of TAP-SPA1 were added in our IP samples and eIF2 α only interacts with TAP-SPA1 under light condition.

Concerns were addressed.

Figure 4: Please, indicate the protein sizes in the panels.

Response: We have added and indicate the protein sizes in the panel in Fig. 4.

Concerns were addressed

4a: please, indicate the amounts of proteins used, both for SPA1 and also which are the increasing amounts of eIF2 α .

Response: The amount of both SPA1 and eIF2 α substrates added in our kinase assay was 500 ng as stated in “*In vitro* kinase assay” method section. We have indicated the increasing amounts of eIF2 α in the figures using “+, ++, and +++” symbols in Fig. 4a and b.

Concerns were addressed

Please, describe in the method section the part corresponding to this experiment. In the main text, SPA1 is tagged with strep and eIF2 is tagged with GST. In methods, eIF2 is tagged with both strep and GST...

Response: As stated in our “Method” section in line 726, eIF2 α was first subcloned into pASK75 for the strep tag and then cloned into pGEX-4T-1 for the GST tag. In line 755, SPA1 was fused with strep tag. We have added this information and modified the text in line 263: “...eIF2 α homologs with GST tagged to their N-terminus and strep tagged to their C-terminus (GST-eIF2 α .1/GST-eIF2 α .2) from bacteria (*E. coli*) to perform *in vitro* kinase assay.”

I have concerns about tagging both ends of a protein, such as the GST-eIF2 α -Strep used here, which frequently likely affects the protein functions. Do the authors have any data and/or comments to indicate that this construction does not affect the eIF2 α function?

Response: To make eIF2 α more soluble and easier to purify, we used GST tag to generate recombinant protein. GST-eIF2 α recombinant proteins were reported to be functional in many previous studies as shown below. However, they are prone to generate truncated and non-specific bands. Thus, we added an eight-amino acid-strep tag (Trp-Ser-His-Pro-Gln-Phe-Glu-Lys) to make it purer and cleaner. The strep tag is also very small that has a milder effect on protein structure.

References:

Dey, M. et al. (2011). Requirement for kinase-induced conformational change in eukaryotic initiation factor 2 α (eIF2 α) restricts phosphorylation of Ser51. *Proc. Natl. Acad. Sci. U.S.A.* **108**, 4316-4321.

Dey, M. et al. (2005). PKR and GCN2 kinases and guanine nucleotide exchange factor eukaryotic translation initiation factor 2B (eIF2B) recognize overlapping surfaces on eIF2 α . *Mol. Cell. Biol.* **25**, 3063-3075.

Concerns were addressed.

4b and c: please, discuss why phosphorylation on the C-ter does not explain the amount of phosphorylation of the full-length. Also, how many biological replicates have been performed of

these experiments? Why use for eIF2alpha.2 experiment in which the different versions of eIF2alpha are barely expressed?

Response: To avoid over exposure of the fragment proteins, we have lowered the amount added for the fragments. According to the trace amount of C-terminus added in the kinase assay (Fig. 4c), the phosphorylation signal is actually very strong that is comparable to the full-length if normalized. The protein amount added for eIF2 α .2 can be observed in the western blot analysis shown in Supplementary Fig. 8.

Please normalize the phosphorylation signal to the western-blot data in Supplementary Fig. 8 and show the fold change in 4b and c.

Response: We thank the reviewer for the suggestion. We have repeated the western blot analyses and obtained a clearer result as shown in Supplementary Fig. 8. We have also normalized the phosphorylation signal to the western blot data as shown in Fig. 4d. As added in our figure legend for Fig. 4d: “A table showing the relative phosphorylation signal intensities of autoradiogram normalized to protein contents detected by immunoblotting as shown in Supplementary Fig. 8. The ratio of the phosphorylation of full length was set to 1 for analysis.” However, the phosphorylation signal shown in autoradiogram might have additive effect due to longer exposure time, so the normalized signal intensities for C-ter and full length might not be equal.

Only one question remains: I do not think it is reasonable to normalize the previous batch of data (fig. 4) to the new set of data (Supplementary Fig. 8), even though it does not affect the author’s conclusions.

Response: We thank the reviewer for the comment. Because we cannot perform Coomassie blue staining, which is the exact same gel that we use for our autoradiography phosphorylation signal detection, and western blot using the exact same SDS-PAGE gel, so we tried to use the western blot signal with equal loading as the protein loaded in our kinase assay to normalize our previous data (Fig. 4). We think this is the best way we can do to perform normalization.

In addition, instead of the Coomassie gels, I would like to see WB to be able to quantify phosphorylation relative to the amount of protein produced.

Response: We thank the reviewer for the suggestion. We have provided a western blot result in Supplementary Fig. 8 showing the amount for N-, M-, and C-eIF2 α fragments added in our kinase assay in Fig. 4c.

Concerns were addressed. Please comment on why the *Ecoli*-purified proteins have so many nonspecific bands in this study.

Response: We thank the reviewer for the comment. We have provided a new western blot result with better quality in Supplementary Fig. 8. The protein purified in our previous batch for the Western blot analysis has many non-specific bands because the column for purification was reused for more than 10 times. We have purchased new columns for purification and performed new Western blot analysis.

Concerns were addressed.

Figure 5B: how was this “semi *in vivo* kinase” experiment done? In the legend, it says that TAP-SPA1 protein was extracted. But in the methods section, L652, it says that for figure 5b, eIF2 alpha proteins were incubated with seedling extracts. If this is so, this is no proof that SPA1 is phosphorylating eIF2 alpha. This shows that the phospho-null versions of the C-ter of eIF2 do not get phosphorylated, and that the WT version gets phosphorylated in the light, but does not show a role of SPA1 on it.

Response: We have deleted the statement in the legend. The semi-*in vivo* kinase assay was performed using seedling extracts. To show that SPAs are necessary for the phosphorylation of eIF2 α , we have repeated this experiment and included *spaQ* seedling sample. As shown in Fig. 5c, in *spaQ* mutants, eIF2 α was less phosphorylated under both dark and light conditions.

Concerns were addressed

Figure 5C: I do not understand the figure... why is the amount of eIF2alpha so low *spaQ* in L compared to the other lanes when the Rpt5 signal is stronger?

Response: We have repeated this phos-tag mobility shift experiment and provided a more significant and higher-quality result as shown in Fig. 5d. The shifted band of phosphorylated eIF2 α was detected in Col-0 but not in *spaQ*. The loading was more equal compared to the previous result.

Due to the nonspecific recognition of eIF2alpha native antibodies and the nature of phostag assay (detecting extra shifting bands), I suggest using PPase to treat the light-treated samples before running phostag gel to confirm that the upper band is because of phosphorylation.

Response: We thank the reviewer for the comment and suggestion. We have reperformed the phostag assay using CIP treatment to confirm that the upper band is the phosphorylated eIF2 α as shown in Fig. 5d. As added in our result in line 354-356, “The *in vivo* phosphorylation status of eIF2 α was further examined by utilizing 15 μ M Phos-tag containing SDS-PAGE gels. In wild type plants treated with white light, only the inactivated boiled calf intestinal alkaline phosphatase (CIP) treated eIF2 α showed mobility shift.”

Concerns were addressed.

Is that super faint phosphorylation signal in Col-0 in L significant?

Response: As mentioned above, we have repeated this phos-tag mobility shift experiment and provided a more significant and higher-quality result as shown in Fig. 5d. The shifted band of phosphorylated eIF2 α was detected in Col-0 but not in *spaQ*. Together with our new semi-*in vivo* kinase assay using P32 detecting phosphorylated C-eIF2 α in both Col-0 and *spaQ* background and *in vivo* kinase assay detecting native eIF2 α in both Col-0 and *spaQ* backgrounds, we hope these results are more convincing.

Please check the comments above.

Response: We thank the reviewer for the comment. As mentioned above, we have reperformed the phostag assay using CIP treatment to confirm that the upper band is the phosphorylated eIF2 α as shown in Fig. 5d. As added in our result in line 354-356, “The *in vivo* phosphorylation status of eIF2 α was further examined by utilizing 15 μ M Phos-tag containing SDS-PAGE gels. In wild type

plants treated with white light, only the inactivated boiled calf intestinal alkaline phosphatase (CIP) treated eIF2 α showed mobility shift.”

Concerns were addressed.

L281: the “significant” reduction?

Response: We have deleted the word “significant”. We have also repeated the phos-tag mobility shift experiment and provided a more significant result.

Please check the comments above.

Response: We thank the reviewer for the comment. As mentioned above, we have reperformed the phostag assay using CIP treatment to confirm that the upper band is the phosphorylated eIF2 α as shown in Fig. 5d. As described in our result in line 354-356, “The *in vivo* phosphorylation status of eIF2 α was further examined by utilizing 15 μ M Phos-tag containing SDS-PAGE gels. In wild type plants treated with white light, only the inactivated boiled calf intestinal alkaline phosphatase (CIP) treated eIF2 α showed mobility shift.”

Concerns were addressed.

L285: please, explain the mSPA1 construct that has a mutation in the kinase domain.

What is this mutation? Does it lack the entire domain? Is it a nucleotide substitution?

Response: The LUC-mSPA1/*spaQ* was described previously (Paik et al, 2019; Wang et al., 2021) as stated in our “method” section in line 599. A mutant form of SPA1 (mSPA1) that has the R517E mutation in the Ser/Thr kinase domain was cloned into pCAMBIA C-Luc vector. This mutation was also described when first mentioned in line 280: “We also used a mutated form of SPA1 (mSPA1) that has the R517E mutation in Ser/Thr kinase domain in the kinase assay and no phosphorylation activity was observed (Fig. 4b; Supplementary Fig. 10).”

Concerns were addressed

And as in Figure 1, no conclusions can be made from these profiles.

Response: We have repeated these experiments and provided better profiles with similar results.

Concerns were addressed

I don't think the data allows to think that eIF2alpha 2 is "specifically" involved in embryogenesis... it can play other roles plant-wide ...

Response: We have toned down this conclusion as stated in line 404: "Both results suggest that eIF2 α .2 might be involved in embryogenic development and seems to play a more important role in that process compared with eIF2 α .1."

Concerns were addressed

The WT, Phospho-null, and phospho-mimetic versions of eIF2 alpha should be transformed in the eif2 alpha mutants (transforming the hets) to see if they complement and so, understand the biological significance of the phosphorylation in the C-ter.

Response: We thank the reviewer for the suggestion. We have actually tried transforming the 3 variants into eIF2 α heterozygous mutants many times but failed. We are in the middle of crossing the 3 transgenic lines with the eIF2 α heterozygous mutants to obtain the complementation lines. We are also in the middle of generating the eIF2 α variants driven by its own promoter and transform them into eIF2 α mutants to obtain the complementation lines. However, it might take more than 6 months for us to obtain the F2 generation to be able to screen and analyze the genotypes. We hope we can report the results in our future studies.

Please use full-length versions of eIF2alpha with the native promoter to do the transgenes or do some transient experiments to demonstrate the functionality of full-length eIF2alpha phosphorylation variants.

Response: We thank the reviewer for the comment and suggestion. According to the reviewer's suggestion (also mentioned above), despite previous reports showing that eIF2 α -GFP is functional, we have crossed our eIF2 α -GFP overexpression lines with two *eif2 α* heterozygous knockout

mutants (SAIL_864 and SAIL_1156) as their homozygous lines were embryogenic lethal. We have screened for the T2 generation and obtained overexpressors in both homozygous knockout mutant lines which should be lethal without overexpressing eIF2 α -GFP. The silique images were shown in Supplementary Fig. 14b. Also added in our result in line 433-437: “To examine the functionality of GFP conjugated eIF2 α .2, we generated complementary lines of eIF2 α .2-WT, 8A, 8D in *eif2 α .2* mutants. These complementary lines produced viable seeds (Supplementary Fig 14b), in contrast to the embryogenic lethal seeds observed in *eif2 α .2* heterozygous knockout mutants (Supplementary Fig. 13b), suggesting that the GFP fusion eIF2 α .2 proteins are functional.”

Moreover, we have also transformed full-length eIF2 α variants to perform transient assays using AGROBEST technique in *spaQ* etiolated seedlings to support that SPAs promote protein translation through phosphorylating eIF2 α C-terminus *in vivo* upon light exposure. As shown in Fig. 7c in this revised manuscript, also described in our result in line 453-462: “To further verify that the translation changes in *spaQ* mutant are due to defective phosphorylation of C-terminal eIF2 α *in vivo*, we have transiently expressed eIF2 α .2-WT, 8A, and 8D using AGROBEST transfection technique and performed polysome profile analyses to see whether ectopic expression of phospho-mimicking eIF2 α .2 could rescue the repressed translation in *spaQ* mutants under light condition. As shown in Fig. 7c, both eIF2 α .2-WT and 8A overexpressing lines display similar pattern with *spaQ* (Fig. 1a), where the translation efficiency cannot be highly induced upon light exposure. However, seedlings overexpressing eIF2 α .2-8D showed high translation efficiencies in both dark and light treatments. Consistent with our ribosome profiling data, the quantification results by isolating NP and PL RNA also show the same patterns with polysome profiles (Fig. 7d).” These results also show that the full-length eIF2 α phosphorylation variants might be functional.

Concerns were addressed.

In Figure 7A, if all transformants are in Col-0 background, why are the ones expressing the 2-8A version not responding normally to light.

Response: We thank the reviewer for the comment. Our suspicion is that the expression of transgene has a dominant effect over the native eIF2 α . It might be because that the protein levels of transgenic eIF2 α variants are much higher than the native eIF2 α and the native eIF2 α weren't sufficient enough to be phosphorylated by SPAs upon light exposure. We have performed a western

blotting showing the native eIF2 α amount in Col-0 and transgenic lines as shown in Supplementary Fig. 14. Compared to Col-0, the levels of native eIF2 α were reduced possibly due to the compensation of transgenic proteins. Once the ternary complex is formed together with 8A, it could cause dominant negative function of native eIF2 α . The smaller portion of native eIF2 α might result in the phenotypes of these transgenics.

Concerns were addressed

Figure 8A. Col-0 is not etiolated. It does not show a hook and the cotyledons are opening.

Response: We thank the reviewer for pointing this out. We might have accidentally stretched the hook and moved the cotyledon while taking pictures. We have repeated this experiment and carefully observed the photomorphogenic phenotypes. A better-quality image is provided as shown in Fig. 8A.

Concerns were addressed

Also, please, discuss why a Col-0 background would not be responding normally to light when expressing ectopically the eIF2 α mutant versions. It can be because the mutant versions sequester the TC, but if so, eIF2 α het mutants should also show a similar phenotype?

Response: As mentioned above, we think that the expression of transgene has a dominant effect over the native eIF2 α . It might be because that the protein levels of transgenic eIF2 α variants are much higher than the native eIF2 α and might sequester and dominate the ternary complex formation. We have performed a western blotting showing the native eIF2 α amount in Col-0 and transgenic lines as shown in Supplementary Fig. 14. Compared to Col-0, the levels of native eIF2 α were reduced possibly due to the compensation of transgenic proteins. Once the ternary complex is formed together with 8A, it could cause the dominant negative function of native eIF2 α . The smaller portion of native eIF2 α might result in the phenotypes of these transgenics. Also, the eIF2 α protein inside eIF2 α heterozygous mutant are still wild-type form of eIF2 α that can be phosphorylated and dephosphorylated by kinases and phosphatases. It might not show similar phenotype as the phosphor-null and phosphor-mimicking transgenic plants.

Concerns were addressed

Fig 9C and D: Please, draw the panels as in A and B, because it seems that GST was pulled-down and that eIF2 β interacted with GST alone with equal intensity.

Response: We have corrected this mistake as shown in Fig. 9c and d.

Concerns were addressed

Methods:

Please, revise all methods as important information is missing from them and it would not be possible to replicate some experiments using the information provided.

Response: We have added new sections including “Vectors construction and generation of transgenic plants”, “Fractionation of nuclear and cytosolic proteins”, “Microscopy of GFP-conjugated protein and BiFC”, “Mobility shift assay”, as well as adding more detailed information in the existing methods.

Concerns were addressed

In vitro pull-down assays: the methods refer to eIF5A and eIF4E that do not appear in the manuscript. Please, rewrite the section using the actual constructs employed in this manuscript.

Response: We have corrected and re-written this method section which should be the assays for eIF2 β and eIF2 γ interactions.

Concerns were addressed

Please, describe the Agrobast method employed here better. It is just a copy of the original paper which also doesn't describe the method very clearly. Which is the “desire cocultivation liquid media” employed in these experiments? How do you cultivate Arabidopsis seedlings in 1 ml of Agrobacterium? Which are the conditions in which you incubate Arabidopsis “in the same growth room for 2 -3 days”?

Response: We have deleted this method as the experiment was replaced with new sets of co-IP experiments without this technique.

Concerns were addressed

The manuscript uses “ribosome-RNA profiling” as the technique to separate polysomes along a sucrose gradient based on the number of ribosomes when its actual name is “polysome profiling”. L121: “ribosome profiling” is a synonym of “Ribo-seq”, which is not what has been done. The manuscript should refer here to the polysome profiles.

Response: We thank the reviewer for pointing this out. We have corrected this word to “polysome profile” throughout the manuscript.

Concerns were addressed

Figure 3B: the Y2H panel is mislabeled. SD-Leu should be SD-Leu-Trp- and the SD- Leu-Trp- should be SD-Leu-Trp-His.

Response: We have corrected this mistake and added the missing labels in Supplementary Fig. 2 in this new version of manuscript.

Concerns were addressed

L46-47: I would not dare to say that “there is a substantial knowledge of the identities and molecular signatures associated with mRNAs regulated at the translational level”. Knowledge in this field is far from substantial. The great majority of the molecular mechanisms governing translation regulation are still unknown.

Response: We thank the reviewer for the comment. We have modified this statement in line 46-48 in the revised manuscript: “Although there are a few studies reporting the identities and molecular signatures associated with mRNAs regulated at the translational level, little is known about the mechanisms of light-triggered regulation of cytoplasmic translation in plants.”

Concerns were addressed

Other issues:

SPA appears as SAP in many places in the manuscript.

Response: We have corrected these mistakes in the revised manuscript.

Concerns were addressed

SUPPRESSOR OF PHYA-105 (SPA1-4) also appears as SUPPRESSOR OF PHYA- 105 1 (SPA1-4).

Response: We have corrected this mistake in the abstract.

Concerns were addressed

L63: I'd put and "However, " before "Because". And they don't only have plant-specific translation control mechanisms. They also have a plant-specific translational machinery with plant-specific translation factors.

Response: We have modified this statement as in line 63: "...however, plants have specific translational machinery with specific translation factors because of their unique biological activities, such as photosynthesis and the capacity to respond to stresses."

Concerns were addressed

L137: what does "(DAP)" stands for?

Response: DAP stands for Daptomycin gene from *Bacillus subtilis*, a spike-in RNA added for normalization. We have added the full names for DAP in the legend of Fig. 1.

Concerns were addressed

Reviewer #2 (Remarks to the Author):

#1: The author claimed that the SPA regulated protein translation via phosphorylating eIF2alpha C-terminus. However, no genetic evidence (epistasis analysis) supports this claim. Instead, they provided separate data by showing that *spaQ* mutant regulates protein translation (Figure 1 and 2), SPA1 could phosphorylate eIF2alpha (Figure 4 and 5), and eIF2alpha phosphorylation regulate protein translation (Figure 7) without closing the regulatory loop.

Ectopic expression of eIF2alpha phospho-variants in *spaQ* mutant should be tested to close the loop by checking if eIF2alpha-8D or 8A could restore *spaQ*'s translation, enhancing translation in darkness while inhibiting translation in light. Importantly, the transgene *spaQ*/eIF2alpha will also benefit the eIF2 complex assembly assay in Figure 9 and the phos-tag gel assay in Figure 5, which will be stated below.

Response: We thank the reviewer for the comment and suggestion. Actually, we have tried to ectopically express the phosphor-variant of eIF2 α , eIF2 α -8A and -8D, in *spaQ* background by transforming the variants into *spaQ* by floral dipping and crossing. However, we have failed many times to generate the eIF2 α variants in *spaQ* mutants. It might be because that over-expression of the variant in *spaQ* is somehow toxic. We are currently trying to generate the transgenics driven by eIF2 α 's native promoter. Nevertheless, to show that SPAs are necessary for the phosphorylation of eIF2 α , we have repeated this phos-tag mobility shift experiment and provided a more significant and higher-quality result as shown in Fig. 5d. The shifted band of phosphorylated eIF2 α was detected in Col-0 but not in *spaQ*. The loading was more equal compared to the previous result. We have also repeated the semi-in vivo kinase assay and included *spaQ* seedling sample. As shown in Fig. 5c, in *spaQ* mutants, eIF2 α was less phosphorylated under both dark and light conditions. To manifest that SPAs are necessary for the ternary complex formation, we have also performed semi-in vivo pull-down assay and showed that eIF2 α could not interact with eIF2 β in *spaQ* mutants (Supplementary Fig. 16).

The transgenes expressing eIF2 α variants in *spaQ* are critical for the study. Without checking the translational output using them, I am not fully convinced that the translation changes in *spaQ* mutant are due to directly phosphorylate eIF2 α in vivo. Meanwhile, it is difficult for me to imagine the toxic effect of eIF2 α -8D in *spaQ* mutant because, theoretically, it will recover the *spaQ* translation to the WT level in light conditions, leading to better plant growth (Fig. 1). Since the

toxic effect is likely due to overexpression of the truncated version of eIF2 α (C-terminus), I look forward to the native promoter-driven eIF2 α C-terminus transgenes in the *spaQ* mutant or would suggest using full-length eIF2 α variants to do the transgenes or at least do some transient assays using *spaQ* protoplasts or AGROBEST established in the study to support further their discoveries that SPAs regulate protein translation in light is through phosphorylating eIF2 α C-terminus *in vivo*.

Response: We thank the reviewer for the comment and suggestion. Following the reviewer's suggestion, we have transformed full-length eIF2 α variants to perform transient assays using AGROBEST technique in *spaQ* etiolated seedlings to support that SPAs promote protein translation through phosphorylating eIF2 α C-terminus *in vivo* upon light exposure. As shown in Fig. 7c in this revised manuscript, also described in our result in line 453-462: "To further verify that the translation changes in *spaQ* mutant are due to defective phosphorylation of C-terminal eIF2 α *in vivo*, we have transiently expressed eIF2 α .2-WT, 8A, and 8D using AGROBEST transfection technique and performed polysome profile analyses to see whether ectopic expression of phospho-mimicking eIF2 α .2 could rescue the repressed translation in *spaQ* mutants under light condition. As shown in Fig. 7c, both eIF2 α .2-WT and 8A overexpressing lines display similar pattern with *spaQ* (Fig. 1a), where the translation efficiency cannot be highly induced upon light exposure. However, seedlings overexpressing eIF2 α .2-8D showed high translation efficiencies in both dark and light treatments. Consistent with our ribosome profiling data, the quantification results by isolating NP and PL RNA also show the same patterns with polysome profiles (Fig. 7d)." The transient protein expression levels of eIF2 α .2 variants in *spaQ* mutants were also verified by immunoblotting as shown in Supplementary Fig. S16.

I realized that the authors may need to use p56-eIF2 α phospho-antibodies to check if SPAs could phosphorylate the site S56 in eIF2 α . The assays performed in Figure 4 could not rule out the possibility that SPAs were able to phosphorylate eIF2 α N-terminus (S56) and C-terminus simultaneously because the S56A protein used in Figure 4 ruled out the possibility, which will affect the rationality of the following experiments only using C-terminal eIF2 α to check the photomorphogenesis in Figure 8.

Response: We thank the reviewer for the comment and suggestion. We have used p56-eIF2 α phospho-antibodies to check if SPAs could phosphorylate the site S56 in eIF2 α as shown in the middle panel in Fig. 5d. In *spaQ* mutants, we can still detect similar p56-eIF2 α band as shown in

Col-0, suggesting that SPAs are not responsible for the S56 phosphorylation of eIF2 α . Our S56A protein used in Fig. 4 also ruled out the possibility that SPA1 phosphorylate S56A of eIF2 α . Moreover, except for the phosphorylation assay conducted in Fig. 5b and 5c using radioactive ^{32}P , all the transgenic lines used in the following experiments, including polysome profiles and photomorphogenesis, are the overexpression lines of “full-length” eIF2 α variants, not the C-term.

My concerns were addressed.

In the proposed model (Figure 10), the SPA1 represses protein translation through eIF2 complex assembly in dark conditions. However, the translational outputs from *spaQ* mutant (Figure 1) and eIF2 α .2-8A transgene (Figure 7) under dark conditions are opposite. Furthermore, no protein interaction was detected between SPA1 and eIF2 α in the dark, as shown in Figure 3c. These data raised concerns that SPA-regulated translation is not through eIF2 α phosphorylation, at least in darkness. As the author wrote in the introduction, SPA kinases repress translation in the dark might through TOR/RPS6 pathway, like *COPI*. It will greatly benefit the proposed model to test the *spaQ*/eIF2 α phospho-variant transgenes (*spaQ*/eIF2 α -WT, *spaQ*/eIF2 α -8A, and *spaQ*/eIF2 α -8D) to check the epistasis.

Response: We thank the reviewer for the comment and suggestion. As mentioned above, we have failed many times generating the transgenic lines in *spaQ* background and we are in the middle of generating the ones driven by eIF2 α 's native promoter. For now, we can only hypothesize why this would happen. In fact, we do not think that SPA1 represses translation through eIF2 complex assembly in dark conditions. Under dark condition, SPA1 does not affect translation through eIF2 complex assembly simply because they don't interact with each other. However, SPA1 might repress translation through TOR-RPS6 pathway under dark condition by working together with *COPI*. Our hypothesis is that SPAs regulate both TOR-RPS6 and eIF2 α phosphorylation pathways but independently, and SPAs play negative roles under dark but play positive roles for TOR-RPS6 regulation in the light condition. In the dark, SPAs work together with *COPI* to inhibit translation through TOR-RPS6 pathway. Thus, in *spaQ* mutant, the translation activity is high. Upon light exposure, beside the effect of light-induced eIF2 α phosphorylation by SPAs, it is likely that SPAs are also necessary to promote TOR-RPS6 pathway through phosphorylating PIFs and other

transcription factors (Paik et al., 2019), thus the translation decreased dramatically in the light. We will try to elucidate this regulatory mechanism and discuss in more detail in our future studies.

I think the authors agree with my points that SPAs do not regulate protein translation via phosphorylation of eIF2 α in darkness. The authors may need to change the model (Fig. 10) using the TOR-RPS6 module instead of the eIF2 complex in dark conditions to avoid misleading.

Response: We thank the reviewer for the suggestion. We have modified our model as shown in Fig. 10. In the dark condition, the eIF2 complex was substituted with TOR-RPS6 module to avoid misleading. In the dark, SPAs might regulate translation through repressing auxin-activated TOR-RPS6 pathway. As also mentioned above, to provide additional evidence that SPAs might also regulate TOR-RPS6 through auxin signal transduction, we have shown the expression patterns of genes involved in auxin-activated signal transduction and TOR-RPS6 pathway at both transcriptional and translational level in Supplementary Fig. 2. As added in our result in line 171-183, “Interestingly, our translomic analysis showed that mRNAs involved in auxin response and auxin-activated signaling pathway are overrepresented in cluster 4, suggesting that SPAs might repress the expression of these mRNAs under dark condition but promote their expression upon light exposure. To test whether SPAs might also regulate TOR-RPS6 pathway through auxin signal transduction, we selected the genes that are involved in TOR-RPS6 pathway and examined their expression patterns at both transcriptional and translational levels. As shown in Supplementary Fig. 2a, genes related to auxin-activated signal transduction show enhanced expression under dark condition but repressed expression upon light exposure. Moreover, 21 selected genes involved in TOR-RPS6 pathway, including TOR, S6K, FCS-LIKE ZINC FINGER (FLZ), LETHAL WITH SEC13 PROTEIN 8 (LST8) and RHO OF PLANTS 2 (ROP2), also show similar pattern (Supplementary Fig. 2b). These results indicate that SPAs might play contrasting role in regulating translation through repressing auxin-TOR signal transduction under dark condition but promoting auxin-TOR signal transduction upon light exposure.”

My concerns were addressed.

#2: The authors need to present more convincing data for the phosphorylation of eIF2 α dependency on SPA kinases in vivo.

In Figure 5b, the semi in vivo phosphorylation of eIF2 α should include *spaQ* mutant because this figure did not provide SPA1 dependency in eIF2 α phosphorylation, and the plant extracts contain massive other kinases, which may lead to false positive results.

In Figure 5c, because the native antibodies (Please provide cat no.) can detect more than one band in the hypo-phosphorylated conditions (Dark), making it not a perfect match for the assay. Moreover, the third line, which has the slightly increased p-eIF2 α , has more protein loading than the rest. I suggest generating the *spaQ*/eIF2 α -GFP transgenes, which were proven to show a unique band in WT, to repeat the phos-tag assay. According to the multiple phosphor-sites predicted in eIF2 α , I expect strong shifted phosphorylation bands in light and weaker ones in *spaQ* mutant.

Response: We thank the reviewer for the great comment and suggestion. We have actually tried expressing eIF2 α -GFP in Col-0 and *spaQ* background by AGROBEST technique and perform the phos-tag assay. However, the protein size of eIF2 α -GFP was too large to distinguish the shifted band. We have repeated this phos-tag mobility shift experiment and provided a more significant and higher-quality result as shown in Fig. 5d. The shifted band of phosphorylated eIF2 α was detected in Col-0 but not in *spaQ*. The loading was more equal compared to the previous result. To show that SPAs are necessary for the phosphorylation of eIF2 α , we have repeated the semi-in vivo kinase assay and included *spaQ* seedling sample. As shown in Fig. 5c, in *spaQ* mutants, eIF2 α was less phosphorylated under both dark and light conditions. Together with our new semi-in vivo kinase assay using 32P detecting phosphorylated C-eIF2 α in both Col-0 and *spaQ* background and in vivo kinase assay detecting native eIF2 α in both Col-0 and *spaQ* backgrounds, we hope these results are more convincing.

The authors partially addressed my concerns. Because AGROBEST is a transient experiment, I would suggest the authors use HA or FLAG-tagged eIF2 α instead of eIF2 α -GFP to perform the band shift assay using phos-tag gel because the HA and FLAG are tiny tags, which will significantly address the big protein size problem of eIF2 α -GFP. If they insist on performing the phos-tag assay using non-specific (which can detect at least two bands) antibodies, I suggest adding another PPase-treated control in Col-0 samples to demonstrate that the shifted band is not a non-specific band in Col-0 background, which is somehow missing in *spaQ* mutant background. Anyway, this study needs PPase treatment to demonstrate that the SPAs-mediated band shift in phos-tag gel is indeed due to phosphorylation in vivo.

Response: We thank the reviewer for the comment and suggestion. We have reperformed the phostag assay using CIP treatment to confirm that the upper band is the phosphorylated eIF2 α as shown in Fig. 5d. As added in our result in line 354-356, “The *in vivo* phosphorylation status of eIF2 α was further examined by utilizing 15 μ M Phos-tag containing SDS-PAGE gels. In wild type plants treated with white light, only the inactivated boiled calf intestinal alkaline phosphatase (CIP) treated eIF2 α showed mobility shift.”

My concerns were addressed.

#3: The claim that SPA kinases regulate translation via affecting eIF2 complex formation could greatly benefit from additional experiments to check eIF2 assembly in *spaQ* mutant background. For example, check the eIF2 α .2 and eIF2 β interaction in *spaQ*/eIF2 α .2-GFP transgenes. The availability of the eIF2 β native antibody from Dr. Browning's lab, one of the authors in this work, makes the experiment easy to perform.

We thank the reviewer for the suggestion. Due to lack of eIF2 α -GFP/*spaQ* transgenic plants, we have performed a semi-*in vivo* pull-down experiment using GST-eIF2 α to pull down native eIF2 β in Col-0 and *spaQ* as shown in Supplementary Fig. 16. The results showed that almost no interaction between alpha and eIF2 β can be detected in both Col-0 and *spaQ*, whereas eIF2 α .2 interacts with eIF2 β in Col-0 after light treatment but no interaction was detected in *spaQ* background.

Ecoli-purified GST-eIF2 α may lack the necessary modification, which is essential for its function. I suggest using transient expression assay through Arabidopsis protoplasts or AGROBEST instead of semi-*in vitro* assay to check the assembly of eIF2 complex by transiently expressing different tagged eIF2 α , β , and even γ in Col-0 and *spaQ* mutant and perform Co-IP experiments.

Response: We thank the reviewer for the comment and suggestion. We have performed co-IP experiments using transiently expressed eIF2 β -GFP and eIF2 γ -GFP in both Col-0 and *spaQ* seedlings using AGROBEST transformation method (Fig. 9c, d, f). As described in our result in line 541-547: “To further verify the complex formation in planta, we also check the eIF2 α -eIF2 β and eIF2 α -eIF2 γ interaction in Col-0 and *spaQ* background by performing *in vivo* co-IP assay using AGROBEST technique. Five-day-old etiolated seedlings of Col-0 and *spaQ* were transfected

with GFP conjugated eIF2 β and eIF2 γ for transient expression. The results showed that in the light-treated Col-0 sample, eIF2 β and eIF2 γ exhibited a stronger interaction with native eIF2 α . In contrast, the interaction intensity between native eIF2 α and eIF2 β or eIF2 γ in light-treated *spaQ* significantly decreased. (Fig. 9c, d, f).”

My concerns were addressed.

#4: Because one of the main focuses in the manuscript is to elucidate how SPA kinases regulate translation efficiency (TE), I suggest calculating TE (TE = mRNApl fold change/mRNAss fold change) instead of showing both the transcriptional and translational changes to simplify the data interpretation in Figure 2, and ranking the GO analysis base on the TE values. The current data in Figure 2 is distracting because we must focus on mRNAss and mRNApl changes.

We thank the reviewer for the suggestion. We have analyzed the TE and added this figure in Fig. 2b in the revised manuscript. As shown in Fig. 2b, most DEG showed increased TE (redder) under dark condition but decreased TE (greener) under light condition, suggesting that SPAs inhibit the translation in dark, but enhance the translation of these genes under light, which is more consistent with our polysomal profile data. We have also kept the classification analysis showing the GO terms regulated at different clusters. We think this is more informative to support our hypothesis that the translational regulation light-responsive and development-related genes results in photomorphogenic phenotypes as well as other phenotypes.

My concerns were addressed.

#5: Typically, the ratio of polysome (PL)/monosome (NP) in ribosomal profiling is dynamically changing, with more polysome companies with less monosome or the other way around, like the paper (38) cited in this manuscript. However, this dynamic is hard to observe in the profiling figures (Figures 1 and 6), instead showing a similar trend in the manuscript. Could the authors comment on this?

We thank the reviewer for pointing this out. We have replaced the previous profiles with better profile results that were performed previously but in a different batch. We have indicated the peaks of 40S, 60S and 80S in the first profile in Fig. 1. The small peak next to 80S on the right represents

disome, so our polysomal fraction contains the part with ≥ 3 ribosomes. We actually do see some dynamic changes in some profiles, especially the size changes in 80S peaks. However, the monosomal fraction sometimes looks a lot bigger because of the increase in disome after light treatment. Our profiles might be different from the paper cited because of the gradient we use (12.5%-60%) might cause different resolution observed and the sensitivity set in different labs. It also might be because that the ribosome biosynthesis is affected in *spaQ* and *cop1-4* backgrounds.

My concerns were addressed.

#6: The AHA labeling assay in Figure S1, *spaQ* mutant showing more newly translated proteins than Col-0 under light conditions, indicating SPAs negatively regulate protein translation in light conditions. Does it contradict the paper's claim that SPAs positively control protein translation under light conditions?

The AHA labeling experiments were repeated more than 3 times with similar results. We have replaced the previous figures with higher-quality visuals that more accurately depict results closest to the average value as shown in Supplementary Fig. 1 in the revised version. The average value of the 3 biological repeats is provided in the table in Supplementary Fig. 1.

My concerns were addressed.

#7: Protein translation occurs in the cytoplasm, and SPA1 kinase was reported localized in the nucleus. How does SPA1 regulate protein translation? Do SPA kinases have cytosol localization in light conditions? Please check the interaction between SPA1 and eIF2 α in vivo using BiFC assay.

We thank the reviewer for the great suggestion. To first observe the light-dependent localization of SPA1, we generated the construct of GFP fused to the C-terminus of SPA1 driven by CaMV 35S promoter and transformed it into Col-0 protoplasts. As shown in Supplementary Fig. 5, SPA1-GFP localized in the nucleus in 91.3% of the transformed protoplasts incubated under dark for 2 h before observation. However, under light condition, 37.5% the transformed protoplasts have SPA1-GFP forming speckles in the cytosol, suggesting that light does induce some translocation of SPA1 into cytoplasm. To further confirm the interaction between SPA1 and eIF2 α in the cytosol,

we isolated cytosolic fraction of TAP-SPA1 from transgenic seedlings and incubated with anti-cMyc to perform co-IP (Fig. 3c). The results showed that the cytosolic TAP-SPA1 has higher interaction intensity with native eIF2 α after light treatment. We also performed BiFC to further validate the interaction between eIF2 α homologs and SPA1 in planta (Fig. 3d). The BiFC results showed that both homologs of eIF2 α can interact with SPA1 in the cytosol under light condition. Collectively, these results suggest that the interactions between SPA1 and eIF2 α homologs are light-dependent and localized in the cytosol. These results indicate that SPA1 might interact with eIF2 α under light condition, and when overexpressing SPA1 might cause them to aggregate when there are not enough eIF2 α to interact with them.

My concerns were addressed.

#8: eIF2 α -8A transgene repressed protein translation in both dark and light conditions (Figure 7); however, the transgene displayed a more extended hypocotyl elongation. Do the authors comment on this counterintuitive phenotype and how the phenotype is linked to translation?

In our hypothesis, we think that the higher translation efficiency results in photomorphogenic phenotype, whereas lower translation efficiency results in skotomorphogenic phenotype. It could be the low translation efficiency in eIF2 α -8A transgenic plants triggers a more severe skotomorphogenesis, leading to longer, extended hypocotyl length, a more closed cotyledon and less chlorophyll synthesis. The transcriptome and translatoome analyses in *spaQ* could reflect the light-responsive genes regulated at the translational level both under dark and under light condition. We will further investigate the genes regulated at the translational level in mutative eIF2 α transgenic plants. We hope to provide more evidence in our future studies.

I suggest using fresh or dry weight to evaluate the translation efficiency eIF2 α variants caused.

Response: We thank the reviewer for the suggestion. We have measured the fresh and dry weight in these transgenic lines following the reviewer's suggestion (Supplementary Fig. 15a, b). To further verify their translation efficiencies and protein synthesis, we have also performed *de novo* protein synthesis analysis using AHA labeling method (Supplementary Fig. 15d). As described in our result in line 448-453: "To further verify the protein production in these transgenic lines, we measured their fresh and dry weight, as well as relative water content. The results showed that both eIF2 α .2-WT and 8D had higher fresh and dry weight than eIF2 α .2-8A (Supplementary Fig.

15a, b), whereas the relative water content in these lines did not exhibit significant difference (Supplementary Fig. 15c). We also performed *de novo* protein synthesis analysis and obtained similar pattern compared to the result of polysome profiles (Supplementary Fig. 15d).”

My concerns were addressed.

#9: Please indicate the SPA1 protein position in Figure 4.

We have added the labels and indicate the protein sizes in the panel in Fig. 4.

My concerns were addressed.

#10: In Figure S2, is it GST or MBP pull-down? Please confirm.

We have corrected this mistake. It is GST pull-down assay as shown in Supplementary Fig. 3 in the revised manuscript.

My concerns were addressed.

Reviewer #3 (Remarks to the Author):

The authors have addressed my concerns in the revised manuscript.

1. Major points: a. The C-term-phosphorylated eIF2 α (8D) promotes translation efficiency and photomorphogenesis in the dark, whereas the C-term-unphosphorylated eIF2 α (8A) results in decreased translation efficiency. *spaQ* is defective in the C-termphosphorylation of eIF2 α and thus in its translation efficiency, but develops enhanced photomorphogenesis with short hypocotyls like 8D under light conditions. The photomorphogenic phenotypes of 8A and *spaQ* seem contradictory at the first sight. One possible explanation is that strong inhibition of the degradation of photomorphogenesis-promoting proteins in *spaQ* is a major cause of constitutive photomorphogenesis since SPAs are crucial components of COP1 E3 ligase complex. Therefore, it is not clear here concerning the contribution of SPA1-mediated eIF2 α phosphorylation in SPA regulation of photomorphogenesis. The reviewer suggests the authors provide necessary genetic

data to specifically evaluate the function of SPA1-mediated eIF2 α phosphorylation (apart from COP1-SPAs E3 activity). For example, compare the phenotypes of Col and LUC-mSPA1/*spaQ* (not shown here) and discuss. Will expressing 8D rescue LUC-mSPA1/*spaQ*?

We thank the reviewer for the comment and suggestion. It is very likely that the photomorphogenesis regulated by SPA1-mediated eIF2 α phosphorylation (as the phenotypes observed in eIF2 α -8A and 8D) and the SPA1-mediated photomorphogenic phenotypes as part of the E3 ligase complex with COP1 are very distinct and resulted from many aspects. The eIF2 α -8D overexpression results in a shorter hypocotyl length, but it is still far from the extremely short hypocotyl in *spaQ* mutant. The photomorphogenic phenotype of *spaQ* comes from the strong inhibition of the degradation of photomorphogenesis-promoting proteins in *spaQ* because of the E3 ligase activity of SPAs. This photomorphogenic phenotype did not reverse even though SPA1 actually promotes photomorphogenesis under light condition. It has been reported that SPA1, as a kinase, plays a major role in regulating photomorphogenesis through phosphorylating PIFs and HY5 under light condition (Paik et al., 2019; Lee et al., 2020; Wang et al., 2021). Thus, one cannot expect the similar effect in photomorphogenic phenotypes caused by ectopic expression of eIF2 α -8A and mutation in SPAs (*spaQ* or mSPA1). The LUC-mSPA1/*spaQ* phenotype was previously reported in Paik et al., 2019 and Wang et al., 2021. It showed similar phenotype as *spaQ* mutant, as mutation in SPA1 kinase domain was very critical for both its roles in E3 ligase component as well as in kinase activity, which result in the overall phenotype. We believe that expressing eIF2 α -8D might partially rescue the constitutive photomorphogenic phenotypes in LUC-mSPA1/*spaQ* and *spaQ*. However, we have failed many times to generate the eIF2 α variants in *spaQ* mutants. It might be because that over-expression of the variant in *spaQ* is somehow toxic. We are currently trying to generate the transgenic lines driven by eIF2 α 's native promoter. Nevertheless, to show that SPAs are necessary for the phosphorylation of eIF2 α , we have repeated this phos-tag mobility shift experiment and provided a more significant and higher-quality result as shown in Fig. 5d. The shifted band of phosphorylated eIF2 α was detected in Col-0 but not in *spaQ*. The loading was more equal compared to the previous result. We have also repeated the semi-in vivo kinase assay and included *spaQ* seedling sample. As shown in Fig. 5c, in *spaQ* mutants, eIF2 α was less phosphorylated under both dark and light conditions. To manifest that SPAs are necessary for the ternary complex formation, we have also performed semi-in vivo pull-down assay and showed that

eIF2 α could not interact with eIF2 β in *spaQ* mutants (Supplementary Fig. 16). We hope these additional experiments support our hypothesis and satisfy reviewer's concern.

b. *spaQ* exhibited enhanced translation in the dark but repressed translation upon light exposure, suggesting that SPAs play opposite roles in translation regulation under dark and light conditions. How does light signal trigger the SPA1-eIF2 α interaction and eIF2 α phosphorylation to promote translation in plants? Based on in vitro systems, the SPA1-eIF2 α interaction and eIF2 α phosphorylation is independent on light conditions. However, these two events were detected specifically under light conditions in planta. In addition, from the model figure, the in vivo association between SPA1 and eIF2 α appears to be regulated at a subcellular level by light, and the assembly of COP1-SPA complex is also affected? The reviewer suggests the authors provide necessary analyses to clarify these points which are important for understanding the molecular basis of SPA1-mediated translation regulation.

We thank the reviewer for the comment. We think that this might be related to the subcellular level of SPA1 regulated by light. To first observe the light-dependent localization of SPA1, we generated the construct of GFP fused to the C-terminus of SPA1 driven by CaMV 35S promoter and transformed it into Col-0 protoplasts. As shown in Supplementary Fig. 5, SPA1-GFP localized in the nucleus in 91.3% of the transformed protoplasts incubated under dark for 2 h before observation. However, under light condition, 37.5% the transformed protoplasts have SPA1-GFP forming speckles in the cytosol, suggesting that light induces either retention or translocation of some SPA1 into cytoplasm from nucleus. To further confirm the interaction between SPA1 and eIF2 α in the cytosol, we isolated cytosolic protein of TAP-SPA1 transgenic seedlings and incubated with anti-cMyc to perform co-IP (Fig. 3c). The results showed that the cytosolic TAP-SPA1 has higher interaction intensity with native eIF2 α after light treatment. We also performed BiFC to further validate the interaction between eIF2 α homologs and SPA1 in planta (Fig. 3d). The BiFC results showed that both homologs of eIF2 α can interact with SPA1 in the cytosol under light condition. Collectively, these results suggest that the interactions between SPA1 and eIF2 α homologs are light-dependent and localized in the cytosol.

c. Another concern of the reviewer associates with the functional redundancy and possible differentiated roles of SPA1-4 in eIF2 α phosphorylation and translation regulation. Do SPA1-4

play conserved roles in eIF2 α interaction and eIF2 α phosphorylation? Considering the functional redundancy of SPA1-4 and the predominant role of SPA1 in repressing photomorphogenesis, *spa1* single mutant is potentially a good material to test translation defects and eIF2 α phosphorylation defects. It is a little bit difficult to understand why these two experiments were performed only using *spaQ* but not including *spa1* single mutant.

We thank the reviewer for the comment. Although SPA1 plays a predominant role in repressing photomorphogenesis, it is possible that all four SPAs play roles in the translational regulation. We think that the global translational changes observed in *spaQ* result in its tiny and severe phenotype that is not observed in *spa1* single mutant and lower order mutants, reflected by the changes in light-responsive, growth-related genes revealed in our translome analyses in *spaQ*, suggesting the possible functional redundancy in the translation defects and eIF2 α phosphorylation defects. We will keep investigating the roles of SPA2-4 in this regulation in our future studies.

2. Minor points:

a. Line 158-159, Cluster 4, representing the largest group of mRNAs (4,006 genes), showed decreased translation but negligible changes in mRNASS level in *spaQ* under the light condition. Why the decreased translation in *spaQ* mRNASS was regarded as negligible? The changes are much clearer than that in the dark.

We thank the reviewer for pointing this out. We have modified this statement in line 163: “Cluster 4, representing the largest group of mRNAs (4,006 genes), showed a more decrease in translation but smaller changes in mRNASS level in *spaQ* under the light condition compared to the levels in dark condition.” Also, to manifest the changes in translation compared to the transcriptional level, we have analyzed the TE by normalizing the fold change in mRNAPL with the fold change in mRNASS (TE = mRNAPL fold change/mRNASS fold change) and performed clustering analysis as shown in Fig. 2b in the revised manuscript. As shown in Fig. 2b, most DEG showed increased TE (redder) under dark condition but decreased TE (greener) under light condition, suggesting that SPAs inhibit the translation in the dark but enhance the translation of these genes under light, which is more consistent with our polysomal profile data.

b. Fig.4, SPA1 band was very vague in Fig.4b, and its migration pattern seemed different from that in Fig.4a. Because there were many unspecific or degraded fragments, it is very difficult to tell the

full-length and truncated eIF2 α .1 and eIF2 α .2 proteins. The authors should indicate these proteins on Coomassie blue staining gels. mSPA1 was not included as a negative control in Fig.4c.

We have added the labels for the proteins and indicate the protein sizes in the panel in Fig. 4. We have also repeated the kinase assay for mSPA1 as shown in Supplementary Fig. 10 to support the SPA1-mediated phosphorylation of eIF2 α .2 in Fig. 4c. The SPA1 migration in Fig. 4a and b was resulted from the different markers used in the experiment. We have added the size in the panel in Fig. 4.

c. Line 254-255, T295 should be S295. Please check throughout the manuscript.

We have corrected these errors throughout the manuscript.

d. Line 314, “+, inactivated boiled CIP” should be “-, inactivated boiled CIP”?

We have repeated this phos-tag mobility shift experiment and provided a more significant and higher-quality result without CIP treatment as shown in Fig. 5d. The shifted band of phosphorylated eIF2 α was detected in Col-0 but not in *spaQ*. The loading was more equal compared to the previous result.

The quantification of Fig.9c and Fig.9d was exactly the same (1.00, 0.80, 1.91). Please check and make the data convincing.

We have corrected this mistake as shown in Fig. 9c and d.